# AlphaVAE: Unified End-to-End RGBA Image Reconstruction and Generation with Alpha-Aware Representation Learning

## Abstract

Recent advances in latent diffusion models have achieved remarkable results in high-fidelity RGB image synthesis by leveraging pretrained VAEs to compress and reconstruct pixel data at low computational cost. However, the generation of transparent or layered content (RGBA images) remains largely unexplored, partly due to the absence of dedicated evaluation metrics. In this work, we introduce a new evaluation metric for RGBA images that adapts standard RGB measures to four-channel data via alpha blending over canonical backgrounds. We further propose ALPHAVAE, a unified end-to-end RGBA VAE that extends a pretrained RGB VAE by incorporating a dedicated alpha channel. The model is trained with a composite objective that combines alpha-blended pixel reconstruction, patch-level fidelity, perceptual consistency, and dual KL divergence constraints to ensure latent fidelity across both RGB and alpha representations. Our RGBA VAE, trained on only 8K images in contrast to 1M used by prior methods, achieves a +4.9 dB improvement in PSNR and a +3.2% increase in SSIM over LayerDiffuse in reconstruction. It also enables superior transparent image generation when fine-tuned within a latent diffusion framework. Our code, data, and models are released on `https://anonymous.4open.science/r/AlphaVAE-0DB8` for reproducibility.

## 1 Introduction

Recent progress in conditional image synthesis has been driven by the synergistic combination of diffusion models (Ho et al., 2020; Nichol & Dhariwal, 2021; Rombach et al., 2022) and variational autoencoders (VAEs) (Kingma & Welling, 2022; Higgins et al., 2016; van den Oord et al., 2017), which together form a powerful framework for high-fidelity image generation. Specifically, latent diffusion models (Rombach et al., 2022) leverage a pretrained VAE decoder to decompress high-resolution RGB images from a significantly lower-dimensional latent representation, drastically reducing computational cost and memory footprint. This compressed-domain synthesis has enabled state-of-the-art results in generating large-scale, semantically aligned images across diverse domains. Consequently, the reconstruction fidelity of the VAE becomes a crucial determinant of overall generation quality.

Despite the maturity of RGB-centric pipelines, generation of transparent or layered content (RGBA) remains severely underexplored, even though most professional editing tools and graphic design workflows rely on layer-based composition with transparency. The scarcity of large-scale RGBA datasets and the sensitivity of pretrained latent diffusion models to shifts in latent space statistics pose significant obstacles. Moreover, existing approaches such as Text2Layer (Zhang et al., 2023) and LayerDiffuse (Zhang & Agrawala, 2024) typically model RGB and alpha channels separately, incurring extra encoder and decoder branches that increase parameter count and inference cost, and often reduce to matting-like post-processing rather than true reconstruction as shown in Figure 4.

To address these challenges, we propose a unified, end-to-end RGBA VAE called ALPHAVAE that extends a pretrained three-channel VAE with an additional alpha channel via zero-initialization and channel-specific weight splits. We carefully design a composite training objective, combining pixel-level alpha-blended reconstruction loss, patch-level fidelity, perceptual consistency, and dual

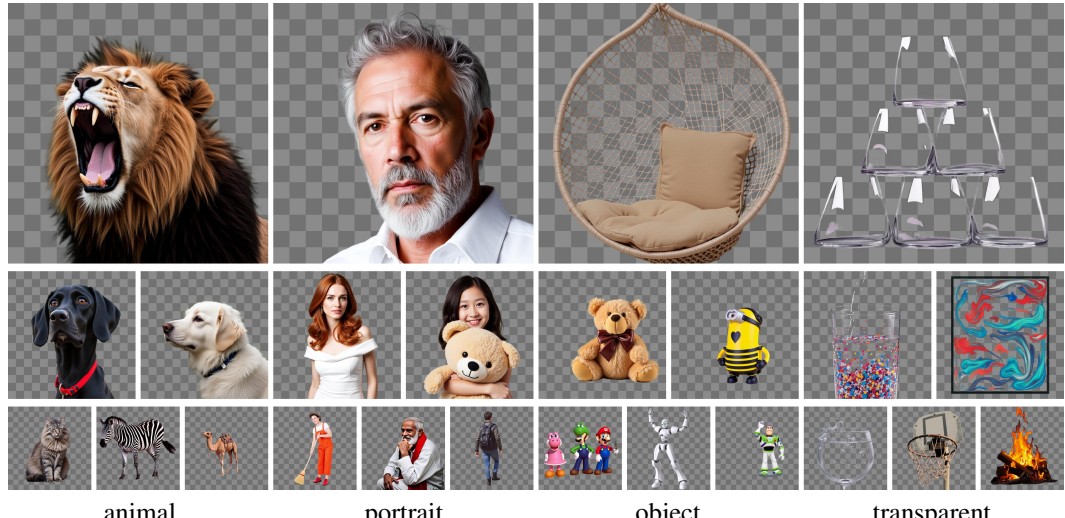

Figure 1: Qualitative results of image generation. Texts shown in each column correspond to category labels from the AIM-500 dataset Li et al. (2021b) and our test dataset. We sample images from different categories in the AIM-500 dataset and feed them into a captioning model Bai et al. (2025) to generate prompts. These prompts are then used to guide the generation of transparent images.

KL constraints, to ensure the reconstruction quality and preserve the original latent distribution while learning high-fidelity transparency representations.

Traditional image quality metrics such as PSNR, SSIM (Wang et al., 2004), and LPIPS (Zhang et al., 2018) are defined for three-channel RGB images and cannot be directly applied to four-channel RGBA data. Moreover, existing reconstruction and generation benchmarks exclusively target RGB content, resulting in the absence of a unified evaluation protocol for transparency-aware synthesis. To address this, we introduce a new evaluation metric for RGBA image reconstruction and generation, which leverages alpha blending over a fixed set of canonical backgrounds to seamlessly extend standard RGB metrics to four-channel images.

Our experiments demonstrate that, with only 8K training images, our method outperforms LayerDiffuse trained on 1M samples across multiple quantitative metrics (e.g. +4.9 dB for PSNR, +3.2% for SSIM). Furthermore, when fine-tuning the corresponding latent diffusion model with ALPHAVAE, we achieve superior transparent image generation compared to baseline pipelines.

In summary, our contributions are threefold:

- We propose ALPHAVAE, a unified end-to-end RGBA VAE architecture that augments a pre-trained three-channel VAE with a dedicated alpha channel, accompanied by specialized reconstruction and regularization losses tailored for RGBA data.

- We introduce a new evaluation metric for RGBA images, which extends standard RGB protocols through alpha blending and enables fair assessment of transparency-aware synthesis.

- We validate our framework with extensive experiments on multiple datasets, demonstrating state-of-the-art VAE reconstruction and superior latent-diffusion–based transparent image generation using much less training data.

## 2 RELATED WORK

**Alpha channel processing in RGBA images.** Alpha channel processing is closely related to image decomposition (Akimoto et al., 2020; Du et al., 2023; Yang et al., 2024), layer extraction (Koyama & Goto, 2018) and image matting (Xu et al., 2017; Xia et al., 2024; Kim et al., 2024; Ma et al., 2023; Li et al., 2023). Traditional approaches in transparent image processing often build on image decomposition, color segmentation, and geometric reasoning in RGB space (Aksoy et al., 2018). Recent diffusion-based methods such as DreamLayer (Huang et al., 2025) extend layered generation

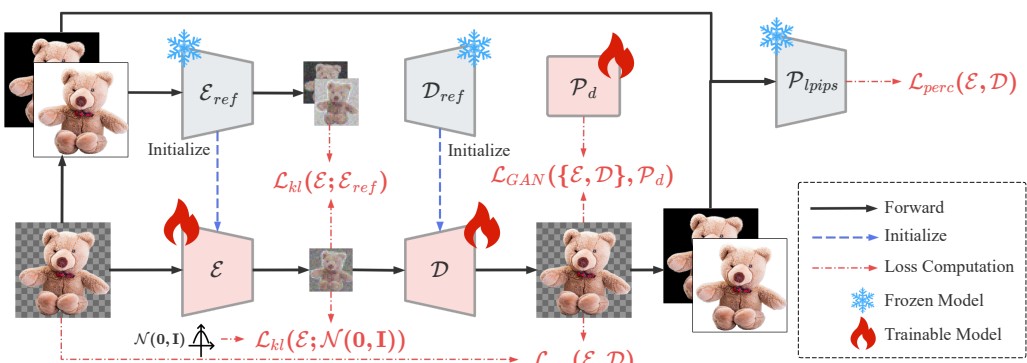

Figure 2: Training pipeline of ALPHAVAE.

by modeling inter-layer relationships through attention mechanisms, enabling more coherent multi-layer compositions. In parallel, image matting focuses on estimating alpha matte from natural images, representing fractional foreground occupancy at the pixel level. Semantic Image Matting (Sun et al., 2021) introduces class-aware matting constraints via semantic trimaps and a multi-class discriminator. Glance and Focus Matting (Li et al., 2022) decomposes the task into high-level segmentation and low-level detail refinement, enabling end-to-end matting with improved generalization to real-world images. Referring Image Matting (Li et al., 2023) extends the problem to natural language-guided alpha extraction and introduces the RefMatte dataset for instruction-based transparency modeling.

**Transparency-aware representation learning in RGBA images.** Modeling the alpha channel is essential for both reconstructing and generating RGBA images (Fontanella et al., 2024; Quattrini et al., 2024; Zhang & Agrawala, 2024; Huang et al., 2025). Recent diffusion-based generation methods have also attempted to produce RGBA images, such as Text2Layer (Zhang et al., 2023), which introduces layered architectures for foreground-background separation, but treats transparency implicitly through compositing attention, without explicit alpha modeling objectives. LayerDiffuse (Zhang & Agrawala, 2024) attempts to address this via a bypass VAE that encodes alpha as a latent offset. However, the learned representation resembles coarse matting masks rather than capturing continuous and structured alpha distributions. More recently, the Anonymous Region Transformer (ART) (Pu et al., 2025) builds upon the Diffusion Transformer (DiT) (Peebles & Xie, 2023) architecture to enable efficient generation of transparency-aware images with region-level control and variable opacity using an anonymous layout design. However, these methods still fall short in generating high-quality RGBA images with an accurate and coherent alpha channel.

## 3 ALPHAVAE: TEACHING VAE TO RECONSTRUCT RGBA IMAGES

In the following, we introduce three key aspects: the evaluation protocol, which adapts standard metrics through alpha blending to account for transparency; the model architecture and initialization, where a pre-trained RGB VAE is extended to four channels; and the training objective, which integrates reconstruction, perceptual, regularization, and adversarial losses.

### 3.1 EVALUATION PROTOCOL

**Motivation.** Standard image–to–image (i2i) metrics—*e.g.*, PSNR, SSIM (Wang et al., 2004), FID (Heusel et al., 2017), LPIPS (Zhang et al., 2018), and LAION-AES (Schuhmann et al., 2022)—are defined for three-channel RGB inputs. Directly extending them to four-channel RGBA images is problematic: pixels with zero opacity ($\alpha = 0$) become visually irrelevant, yet naïve 4-channel formulations still penalize the corresponding RGB values. We therefore evaluate each image after compositing it onto a set of canonical backgrounds, converting the task back to the well-studied RGB setting while fully respecting transparency.

**Alpha blending.** Given an RGBA image $x \in \mathbb{R}^{4 \times H \times W}$ with RGB part $x_{\mathrm{rgb}}$ and alpha matte $x_\alpha \in [0, 1]$, and an RGB background $b \in \mathbb{R}^{3 \times H \times W}$, alpha blending is defined as

$$\mathcal{A}(x, b) = x_{\mathrm{rgb}} \odot x_\alpha + b \odot (1 - x_\alpha), \tag{1}$$

where $\odot$ denotes element-wise multiplication[1]. This operation produces a standard RGB image that faithfully reflects the visual appearance of $x$ composited over background $b$.

**Metric extension.** Let $\mathcal{M}_3$ be any RGB metric, $\mathcal{M}_3 : \mathbb{R}^{N \times 3 \times H \times W} \times \mathbb{R}^{N \times 3 \times H \times W} \to \mathbb{R}$. Define its RGBA counterpart $\mathcal{M}_4$ by averaging $\mathcal{M}_3$ over a predefined background set $\mathcal{B}$:

$$\mathcal{M}_4(\mathcal{X}, \hat{\mathcal{X}}) = \frac{1}{|\mathcal{B}|} \sum_{b \in \mathcal{B}} \mathop{\mathbb{E}}_{(x, \hat{x}) \in \mathcal{X} \times \hat{\mathcal{X}}} \Big[ \mathcal{M}_3 \Big( \mathcal{A}(x, b), \mathcal{A}(\hat{x}, b) \Big) \Big] \tag{2}$$

where $\mathcal{X}$ and $\hat{\mathcal{X}}$ denote the ground-truth and reconstructed sets, respectively.

**Background set.** We adopt nine solid colours that span the pixel $\{0, 1\}^3 \cup \{0.5\}^3$: `black`, `white`, `red`, `green`, `blue`, `yellow`, `cyan`, `magenta` and `gray`. Using low-frequency, texture-free backgrounds eliminates semantic bias and yields numerically stable values for all metrics.

**Reported scores.** Final scores are reported as the average across all background colors. This pipeline allows robust evaluation of both RGB fidelity and transparency-aware perceptual quality, while remaining compatible with existing metrics.

## 3.2 MODEL ARCHITECTURE AND INITIALIZATION

Variational Autoencoders (VAEs) encode high-dimensional inputs into a structured latent space and then decode this representation to reconstruct the original data. The encoder

$$\mathcal{E} : \mathbb{R}^{C \times H \times W} \longrightarrow \mathbb{R}^{D \times h \times w} \tag{3}$$

maps an image $x \in \mathbb{R}^{C \times H \times W}$ to a distribution in latent space, where the dimensionality reduction $D\, h\, w < C\, H\, W$ encourages the model to capture salient, abstract features in compact form. The decoder

$$\mathcal{D} : \mathbb{R}^{D \times h \times w} \longrightarrow \mathbb{R}^{C \times H \times W} \tag{4}$$

reconstructs the input from its latent code, effectively up-sampling and re-rendering the encoded information.

In the RGBA setting, $x \in \mathbb{R}^{4 \times H \times W}$ comprises an RGB part $x_{\mathrm{rgb}}$ and an alpha (transparency) channel $x_\alpha$. To reuse a pre-trained RGB (three-channel) VAE while supporting four channels, we extend the first convolution of $\mathcal{E}$ and the final convolution of $\mathcal{D}$ from three to four channels.

Let $W_0^{\mathcal{E}} \in \mathbb{R}^{k \times k \times 4 \times D_0}$ and $b_0^{\mathcal{E}} \in \mathbb{R}^{D_0}$ denote the weight and bias of the first convolution in $\mathcal{E}$, and let $W_L^{\mathcal{D}} \in \mathbb{R}^{k \times k \times D_L \times 4}$ and $b_L^{\mathcal{D}} \in \mathbb{R}^4$ denote the weight and bias of the last convolution in $\mathcal{D}$. We copy the pre-trained RGB parameters into the first three channel slices and initialize the newly added alpha-channel slices as

$$W_0^{\mathcal{E}}[:, :, 4, :] = \mathbf{0}, \qquad W_L^{\mathcal{D}}[:, :, :, 4] = \mathbf{0}, \qquad b_L^{\mathcal{D}}[4] = \mathbf{1},$$

while keeping $b_0^{\mathcal{E}}$ and the first three components of $b_L^{\mathcal{D}}$ unchanged. This simple yet effective initialization preserves the rich feature extraction learned from RGB images and lets the network gradually learn to exploit the transparency channel during fine-tuning. The pipeline is illustrated in Figure 2.

## 3.3 TRAINING OBJECTIVE

We train an RGBA VAE with a composite loss function that includes the following terms.

### 3.3.1 RECONSTRUCTION LOSS

In image reconstruction tasks, a common approach is to penalize the difference between the original image $x$ and its reconstruction $\hat{x} = \mathcal{D}(\mathcal{E}(x))$, where $\mathcal{E}$ is an encoder and $\mathcal{D}$ is a decoder. A

---

[1]For simplicity, element-wise multiplication is implied when the operator is omitted, unless stated otherwise.

straightforward reconstruction loss would be the squared L2 norm, which calculates the mean squared error (MSE) between the reconstructed and original image tensors:

$$\mathcal{L}_{rec}(\mathcal{E}, \mathcal{D}) = \|\hat{x} - x\|_2^2 \tag{5}$$

As described in Section 3.1, the ideal difference should be the RGB difference after blending a background. Thus, our proposed reconstruction loss, $\mathcal{L}_{rec}$, is the expected L2 difference between the alpha-blended reconstruction and the alpha-blended original image, which we call Alpha-Blending Mean Square Error (ABMSE), taken over a distribution of backgrounds $b$:

$$\mathcal{L}_{rec}(\mathcal{E}, \mathcal{D}) = \mathbb{E}_b \|\mathcal{A}(\hat{x}, b) - \mathcal{A}(x, b)\|_2^2 \tag{6}$$

The expression inside the expectation becomes:

$$\mathcal{A}(\hat{x}, b) - \mathcal{A}(x, b) = (\hat{x}_{rgb}\hat{x}_\alpha - x_{rgb}x_\alpha) - b(\hat{x}_\alpha - x_\alpha) \tag{7}$$

To simplify the calculations, each pixel in $b$ is considered to be independent and identically distributed (i.i.d.). Let $P = \hat{x}_{rgb}\hat{x}_\alpha - x_{rgb}x_\alpha$ (the difference in premultiplied RGB values) and $\Delta_\alpha = \hat{x}_\alpha - x_\alpha$ (the difference in alpha values). Then, $\|(\hat{x}_{rgb}\hat{x}_\alpha - x_{rgb}x_\alpha) - b(\hat{x}_\alpha - x_\alpha)\|_2^2 = \|P - b\Delta_\alpha\|_2^2$. Expanding[2] this squared norm and taking the expectation with respect to $b$:

$$\mathcal{L}_{rec}(\mathcal{E}, \mathcal{D}) = \mathbb{E}_b \left[ \|P - b\Delta_\alpha\|_2^2 \right] = \|P\|_2^2 - 2\Delta_\alpha \langle \mathbb{E}[b], P \rangle + \Delta_\alpha^2 \|\mathbb{E}[b^2]\|_1 \tag{8}$$

Thus, $\mathbb{E}[b]$ and $\mathbb{E}[b^2]$ can be pre-calculated. In practice, the distribution of $b$ is estimated from the ImageNet training split. Detailed statistics of $b$ are provided in Appendix B.

### 3.3.2 PERCEPTUAL LOSS

In VQVAE and SD, perceptual loss is calculated by a pretrained model $\mathcal{P}_{lpips}$:

$$\mathcal{L}_{perc}(\mathcal{E}, \mathcal{D}) = \mathcal{P}_{lpips}(\hat{x}, x) \tag{9}$$

However, this model receives three-channel images. Thus, before feeding a image into $\mathcal{P}_{lpips}$, we also perform alpha blending on it, i.e., $\mathcal{P}_{lpips}(\mathcal{A}(\hat{x}, b), \mathcal{A}(x, b))$. Note that the model $\mathcal{P}_{lpips}$ can not be simply represented by a polynomial which can simplifies the function a composition of $x$, $\hat{x}$, and $\mathbb{E}\left[b^k\right]$. Finding that the distribution of channel value is concentrated at 0 and 1, we apply white-colored and black-colored images as the background, i.e.,

$$\mathcal{L}_{perc}(\mathcal{E}, \mathcal{D}) = \mathbb{E}_{b \in \{\mathbf{0}, \mathbf{1}\}} [\mathcal{P}_{lpips}(\mathcal{A}(\hat{x}, b), \mathcal{A}(x, b))] \tag{10}$$

$$= \frac{1}{2} (\mathcal{P}_{lpips}(\mathcal{A}(\hat{x}, \mathbf{0}), \mathcal{A}(x, \mathbf{0})) + \mathcal{P}_{lpips}(\mathcal{A}(\hat{x}, \mathbf{1}), \mathcal{A}(x, \mathbf{1}))) \tag{11}$$

### 3.3.3 REGULARIZATION LOSS

In our Variational Autoencoder (VAE) framework, we incorporate a regularization loss to govern the characteristics of the latent space distribution generated by the encoder, denoted as $\mathcal{E}(\cdot)$. The regularization is achieved through two distinct Kullback-Leibler (KL) divergence terms.

The first KL divergence term, $\mathcal{L}_{kl}(\mathcal{N}(\mathbf{0}, \mathbf{I}))$, enforces a standard regularization on the encoder. It encourages the posterior distribution of the latent variables $z$ given an input $x$, $q(z|x)$ (as approximated by the encoder $\mathcal{E}(z|x)$), to conform to a prior, which is a standard multivariate Gaussian distribution $\mathcal{N}(z; \mathbf{0}, \mathbf{I})$. This is a common practice in VAEs to ensure a well-structured and continuous latent space. The formal definition is:

$$\mathcal{L}_{kl}(\mathcal{E}; \mathcal{N}(0, \mathbf{I})) = \mathbb{E}_x [KL(\mathcal{E}(z|x)\|\mathcal{N}(z; \mathbf{0}, \mathbf{I}))] \tag{12}$$

The second KL divergence term, $\mathcal{L}_{kl}(\mathcal{E}_{ref})$, is specifically introduced due to our fine-tuning strategy. Our current four-channel VAE encoder, $\mathcal{E}$, is adapted from a pre-trained three-channel reference VAE encoder, $\mathcal{E}_{ref}$. To ensure that the latent distribution of our fine-tuned encoder $\mathcal{E}$ remains compatible with that of $\mathcal{E}_{ref}$, thereby facilitating its integration into a diffusion model with minimal subsequent

---

[2]The detailed deduction is displayed in Appendix A.

Table 1: VAE reconstruction results on AIM-500 Li et al. (2021b) and our test dataset. ↑ indicates higher values are better, and ↓ indicates lower values are better. **Best** and second-best values per metric are indicated in bold and by underlining, respectively. Differences relative to the LayerDiffuse baseline are colored green for improvement. These formatting conventions are maintained for all subsequent result tables.

| Method | Base Model | AIM | | | | | AlphaVAE Test | | | | |
|---|---|---|---|---|---|---|---|---|---|---|---|
| | | PSNR ↑ | SSIM ↑ | LAION AES ↑ | rFID ↓ | LPIPS ↓ | PSNR ↑ | SSIM ↑ | LAION AES ↑ | rFID ↓ | LPIPS ↓ |
| LayerDiffuse | SDXL | 32.0879 | 0.9436 | 4.9208 | 17.7023 | 0.0418 | 32.4531 | 0.9473 | 5.0306 | 6.4832 | 0.0324 |
| AlphaVAE (Ours) | SDXL | 35.7446 
 (+3.6567) | 0.9576 
 (+0.0140) | 4.9456 
 (+0.0248) | **10.9178** 
 (-6.7845) | 0.0495 
 (+0.0077) | 35.5597 
 (+3.1066) | 0.9605 
 (+0.0132) | 5.0500 
 (+0.0194) | 3.7305 
 (-2.7527) | 0.0402 
 (+0.0078) |
| | FLUX | **36.9439** 
 (+4.8560) | **0.9737** 
 (+0.0301) | **4.9511** 
 (+0.0303) | 11.7884 
 (-5.9139) | **0.0283** 
 (-0.0135) | **38.1987** 
 (+5.7456) | **0.9792** 
 (+0.0319) | **5.0616** 
 (+0.0310) | **1.6600** 
 (-4.8232) | **0.0145** 
 (-0.0179) |

fine-tuning, we introduce this additional constraint. This term minimizes the KL divergence between the latent distribution produced by our encoder $\mathcal{E}$ and that produced by the reference encoder $\mathcal{E}_{ref}$.

A critical consideration is the input channel mismatch: $\mathcal{E}$ processes four-channel images, while $\mathcal{E}_{ref}$ expects three-channel images. To address this, we also employ the blending function $\mathcal{A}(x, b)$. For the input to our fine-tuned encoder $\mathcal{E}$ in this context, the blended three-channel image $\mathcal{A}(x, b)$ is concatenated with a fourth channel consistently set to unity (i.e., $[\mathcal{A}(x, b); 1]$). The loss is formulated as:

$$\mathcal{L}_{kl}(\mathcal{E}; \mathcal{E}_{ref}) = \mathbb{E}_{x}\left[ \mathbb{E}_{b \in \{\mathbf{0}, \mathbf{1}\}} \left[ KL\left( \mathcal{E}\left( z \Big| [\mathcal{A}(x, b); 1]\right) \Big\| \mathcal{E}_{ref}\left( z \Big| \mathcal{A}(x, b)\right)\right)\right]\right] \tag{13}$$

### 3.3.4 GAN Loss

In (Esser et al., 2021; Rombach et al., 2022), a patch-based discriminator $\mathcal{P}_d$ is usually applied to learn a rich representation.

$$\mathcal{L}_{GAN}(\{\mathcal{E}, \mathcal{D}\}, \mathcal{P}_d) = \log \mathcal{P}_d(x) + \log(1 - \mathcal{P}_d(\hat{x})) \tag{14}$$

Following VQGAN (Esser et al., 2021), we also compute the adaptive weight $\lambda_{adapt}$ according to

$$\lambda_{adapt} = \frac{\nabla_{\mathcal{D}_L}[\mathcal{L}_{rec}]}{\nabla_{\mathcal{D}_L}[\mathcal{L}_{GAN}] + \epsilon} \tag{15}$$

where $\mathcal{D}_L$ represents the last layer of $\mathcal{D}$, and $\epsilon = 10^{-4}$ ensures numerical stability.

Thus, the final objective is:

$$\mathcal{E}^*, \mathcal{D}^* = \arg\min_{\mathcal{E}, \mathcal{D}} \max_{\mathcal{P}_d} \Big[ \mathcal{L}_{rec}(\mathcal{E}, \mathcal{D}) + \lambda_{perc}\mathcal{L}_{perc}(\mathcal{E}, \mathcal{D}) + \lambda_{norm}\mathcal{L}_{kl}(\mathcal{E}; \mathcal{N}(0, \mathbf{I}))$$
$$+ \lambda_{ref}\mathcal{L}_{kl}(\mathcal{E}; \mathcal{E}_{ref}) + \lambda_{GAN}\lambda_{adapt}\mathcal{L}_{GAN}(\{\mathcal{E}, \mathcal{D}\}, \mathcal{P}_d)\Big] \tag{16}$$

## 4 EXPERIMENTS

### 4.1 IMPLEMENTATION DETAILS

We train both the VAE and the diffusion model on the same dataset, where image–caption pairs are constructed using Qwen-VL-2.5 (Bai et al., 2025) as the captioning model. The dataset consists of 7,722 training images and 402 testing images, with further details provided in Appendix D. The detailed configurations of the VAE and the diffusion model are described in the following subsections. All experiments are conducted on 8 NVIDIA H800 GPUs.

**VAE.** We fine-tune the VAE components of both SDXL (Podell et al., 2023) and FLUX.1[dev] (Labs, 2024) models. As a form of data augmentation, each input transparent image is blended with a randomly generated solid-color background with a probability of 0.3 during training. Full-parameter

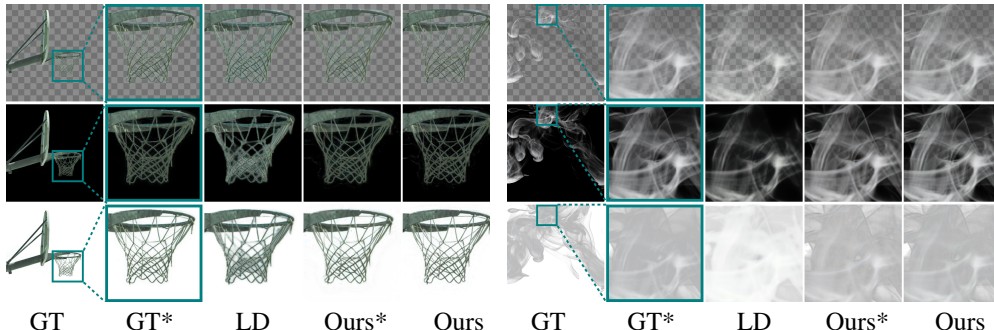

GT    GT*    LD    Ours*    Ours    GT    GT*    LD    Ours*    Ours

Figure 3: Qualitative Results of image reconstruction on transparent images. Each row shows reconstruction results (with 4 channels) composited over three backgrounds: checkerboard, black, and white (from top to bottom) for better visualization. LD represents layerdiffuse Zhang & Agrawala (2024), Ours* denotes ALPHAVAE based on SDXL, and Ours is ALPHAVAE based on FLUX. The above images have rich alpha channel information from the AIM-500 dataset.

fine-tuning is performed for 30,000 iterations using a global batch size of 8. We employed the AdamW optimizer with a learning rate of $1 \times 10^{-5}$.

**Diffusion.** After training the VAE, we fine-tune the corresponding latent diffusion module via Low-Rank Adaptation (LoRA) with a rank of 64. The model is optimized using the Prodigy optimizer at a learning rate of 1.0, and trained for 20,000 iterations with a global batch size of 8.

## 4.2 QUANTITATIVE RESULTS

**VAE.** We conduct quantitative evaluations of ALPHAVAE using the proposed RGBA metrics. Specifically, we evaluate the VAEs in both SDXL and FLUX models on our test dataset and the AIM-500 dataset. As shown in Table 1, our method significantly outperforms LayerDiffuse (Zhang & Agrawala, 2024) in terms of PSNR and rFID across both datasets. On AIM-500, ALPHAVAE improves PSNR from 32.09 to 35.74 (SDXL) and 36.94 (FLUX), yielding absolute gains of +3.66 and +4.86 dB, respectively. The rFID also drops sharply from 17.70 to 10.92 (SDXL) and 11.79 (FLUX), indicating improved perceptual realism. Similar trends are observed on the Alpha benchmark, where PSNR increases by +3.11 and +5.75 dB, and rFID decreases from 6.48 to as low as 1.66. These improvements demonstrate the robustness and generalizability of our proposed VAE framework, with consistently superior reconstruction quality under multiple metrics.

**Diffusion.** We evaluate the FID metric of our diffusion model on our test dataset and the AIM-500 dataset to measure the distributional discrepancy between generated and real images. As shown in Table 2, the results indicate that our generation model exhibits strong generalization capability. Specifically, on AIM-500, our method achieves an FID of 155.66, improving by 4.61 points compared to LayerDiffuse (160.27). On our test dataset, we attain a FID of 74.12, an improvement (reduction) of 6.15 points from the 80.27 baseline.

Table 2: Generation results on AIM-500 and our test dataset.

| Setting | AIM | | | AlphaVAE Test | | |
|---|---|---|---|---|---|---|
| | LayerDiffuse | Ours | w/o Ref KL | LayerDiffuse | Ours | w/o Ref KL |
| **FID** ↓ | 160.2664 | **155.6606** (-4.6058) | 161.4522 (+1.1858) | 80.2703 | **74.1224** (-6.1479) | 74.5844 (-5.6859) |

## 4.3 QUALITATIVE RESULTS

**VAE.** We present qualitative reconstruction results for transparent images in Figure 3. Each row displays composited outputs on three different backgrounds: checkerboard, black, and white, providing a clear view of both the RGB and the alpha channel fidelity. All samples are drawn from the AIM-500 test dataset, which contains images with rich transparency details. Note that this dataset was not used during training. Compared to layerdiffuse (Zhang & Agrawala, 2024), our model

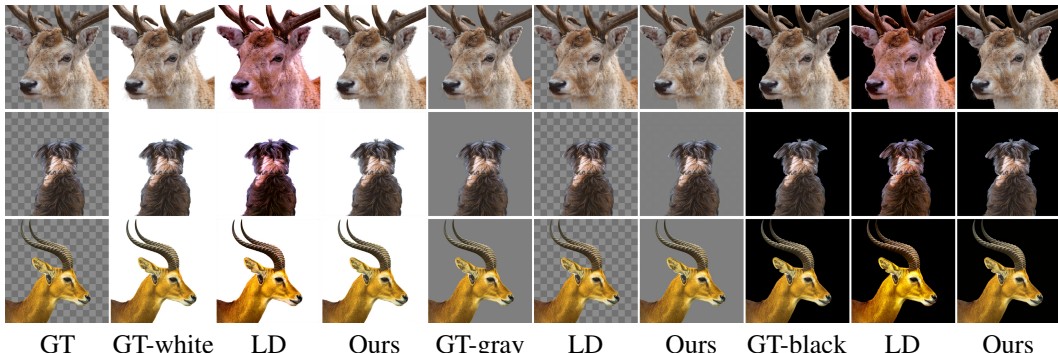

GT    GT-white    LD    Ours    GT-gray    LD    Ours    GT-black    LD    Ours

Figure 4: Qualitative results of image reconstruction on transparent images blended over solid color backgrounds. GT-white, GT-gray, and GT-black denote the ground-truth transparent image blended with white, gray, and black backgrounds, respectively.

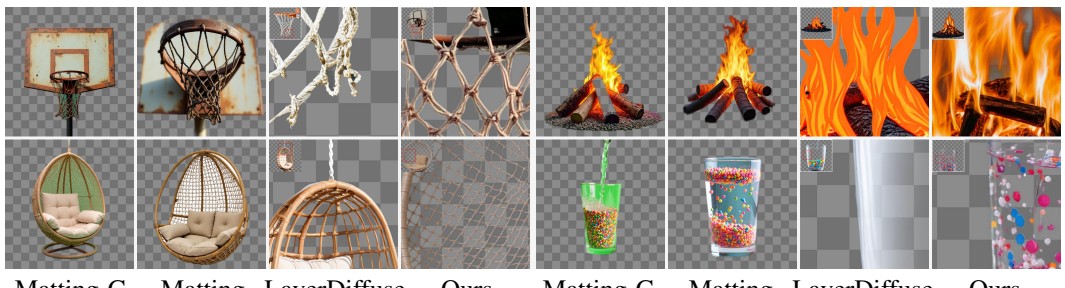

Matting-G    Matting    LayerDiffuse    Ours    Matting-G    Matting    LayerDiffuse    Ours

Figure 5: Qualitative comparison of RGBA object generation. All methods were given the same prompts, except that Matting-G (matting with a green screen background) was provided an additional background constraint, while Matting (matting with an unconstrained background) used the identical prompt as the generative baselines. For LayerDiffuse and Ours, the top-left inset shows the corresponding ground-truth image, with the red-boxed region magnified as the main panel to highlight fine-grained details for clearer comparison.

(Ours and Ours*) better preserves the original structure and transparency of the input images. In particular, LayerDiffuse (Zhang & Agrawala, 2024) fails to reconstruct the alpha channel faithfully: it tends to shrink the transparency regions or reduce opacity, leading to results that diverge from the original image. This indicates a degradation in the representational capacity of its VAE component, as it cannot maintain consistent alpha values during decoding. Furthermore, when the transparent input image is composited with a solid-color background, the VAE in LayerDiffuse not only fails to preserve alpha consistency, but also exhibits noticeable degradation in RGB fidelity — for instance, as shown in Figure 4, reconstructed images often display a reddish color cast, diverging from the original appearance.

**Diffusion.** We generate images using our fine-tuned diffusion model, as shown in Figure 1. The prompts used for image generation are derived by applying a captioning model (Bai et al., 2025) to the original images in the AIM-500 dataset and our test dataset. Notably, the training set contains only around 8K images, yet the model is able to generate semantically aligned images from the out-of-distribution AIM-500 dataset. The plausible generation highlights the strong generative prior learned by the diffusion model.

## 4.4 QUALITATIVE COMPARISON ON TRANSPARENT IMAGE GENERATION PIPELINES

In general, there are two ways to obtain the RGBA image of a specific object. Before introducing each category, we describe the unified experimental setup. To ensure fairness, we used identical prompts to generate four sets of images. The only difference is that for the matting (green-screen) variant, we explicitly added a green-screen background constraint, while the matting (unconstrained) and all generation-based methods used identical prompts.

**Matting-based approaches.**These methods first generate an RGB image containing the object and then apply matting to extract the object.As illustrated in Figure 5, both matting variants suffer from severe boundary artifacts, background leakage, and poor handling of transparent or mesh-like objects.In addition, matting pipelines typically require manual inputs such as bounding boxes, foreground points, or background points, making them unsuitable for large-scale RGBA generation that would be valuable for dataset construction and downstream tasks.

**Generation-based approaches.**These methods directly synthesize RGBA images without requiring a separate matting step. LayerDiffuse represents the current state of the art. As shown in Figure 5, although LayerDiffuse avoids manual annotations, it still struggles to preserve fine structural details and to faithfully reproduce transparency. In particular, objects with mesh-like patterns or structures containing holes are especially challenging, often resulting in blurry edges, background leakage, or incomplete reconstructions. In contrast, our diffusion-based method consistently produces RGBA images with clean boundaries and accurate transparency, demonstrating robustness across diverse cases and scalability for large-scale generation.

## 4.5 ABLATIVE STUDY

Table 3: Ablation study results.

| Method | AIM | | | | | Alpha | | | | |
|---|---|---|---|---|---|---|---|---|---|---|
| | PSNR ↑ | SSIM ↑ | LAION AES ↑ | rFID ↓ | LPIPS ↓ | PSNR ↑ | SSIM ↑ | LAION AES ↑ | rFID ↓ | LPIPS ↓ |
| AlphaVAE + FLUX | 36.9439 | 0.9737 | 4.9511 | **11.7884** | **0.0283** | 38.1987 | 0.9792 | 5.0616 | 1.6600 | **0.0145** |
| w/o Norm KL | 35.1629 (-1.7810) | 0.9691 (-0.0046) | 4.9538 (+0.0027) | 16.3582 (+4.5698) | 0.0358 (+0.0075) | 37.3880 (-0.8107) | 0.9776 (-0.0016) | **5.0639** (+0.0023) | 2.4898 (+0.8298) | 0.0174 (+0.0029) |
| w/o Ref KL | **36.9473** (+0.0034) | **0.9738** (+0.0001) | **4.9555** (+0.0044) | 12.0928 (+0.3044) | 0.0287 (+0.0004) | 38.1428 (-0.0559) | 0.9790 (-0.0002) | 5.0635 (+0.0019) | 1.7945 (+0.1345) | 0.0155 (+0.0010) |
| w/o GAN | 36.8578 (-0.0861) | 0.9734 (-0.0003) | 4.9484 (-0.0027) | 12.8674 (+1.0790) | 0.0299 (+0.0016) | **38.4839** (+0.2852) | **0.9802** (+0.0010) | 5.0556 (-0.0060) | 1.7091 (+0.0491) | 0.0154 (+0.0009) |
| w/o LPIPS | 32.9796 (-3.9643) | 0.9636 (-0.0101) | 4.9547 (+0.0036) | 22.3132 (+10.5248) | 0.0457 (+0.0174) | 36.5583 (-1.6404) | 0.9763 (-0.0029) | 5.0538 (-0.0078) | 3.1332 (+1.4732) | 0.0196 (+0.0051) |
| w/o ABMSE | 36.8036 (-0.1403) | 0.9709 (-0.0028) | 4.9460 (-0.0051) | 13.4679 (+1.6795) | 0.0312 (+0.0029) | 38.0847 (-0.1140) | 0.9784 (-0.0008) | 5.0589 (-0.0027) | **1.6599** (-0.0001) | 0.0150 (+0.0005) |
| AlphaVAE + SDXL | 35.7446 | **0.9576** | 4.9456 | **10.9178** | **0.0495** | 35.5597 | 0.9605 | 5.0500 | 3.7305 | 0.0402 |
| w/o Norm KL | 33.8361 (-1.9085) | 0.9504 (-0.0072) | 4.9277 (-0.0179) | 18.9242 (+8.0064) | 0.0607 (+0.0112) | 35.4985 (-0.0612) | 0.9604 (-0.0001) | 5.0475 (-0.0025) | 3.8617 (+0.1312) | 0.0401 (-0.0001) |
| w/o Ref KL | 33.9751 (-1.7695) | 0.9509 (-0.0067) | 4.9383 (-0.0073) | 18.2684 (+7.3506) | 0.0587 (+0.0092) | 35.4917 (-0.0680) | 0.9602 (-0.0003) | 5.0484 (-0.0016) | 3.7902 (+0.0597) | **0.0393** (-0.0009) |
| w/o GAN | 34.0428 (-1.7018) | 0.9507 (-0.0069) | 4.9338 (-0.0118) | 18.5222 (+7.6044) | 0.0598 (+0.0103) | **35.6887** (+0.1290) | **0.9610** (+0.0005) | 5.0467 (-0.0033) | **3.6883** (-0.0422) | 0.0405 (+0.0003) |
| w/o LPIPS | 33.9153 (-1.8293) | 0.9511 (-0.0065) | 4.9348 (-0.0108) | 18.6036 (+7.6858) | 0.0592 (+0.0097) | 35.4946 (-0.0651) | 0.9604 (-0.0001) | 5.0494 (-0.0006) | 3.7675 (+0.0370) | 0.0396 (-0.0006) |
| w/o ABMSE | 34.5586 (-1.1860) | 0.9512 (-0.0064) | **4.9471** (+0.0015) | 16.0156 (+5.0978) | 0.0559 (+0.0064) | 35.3997 (-0.1600) | 0.9587 (-0.0018) | **5.0603** (+0.0103) | 3.7250 (-0.0055) | 0.0393 (-0.0009) |

Following the evaluation protocols described in Section 3.1, we systematically assess the performance of various VAE configurations using PSNR, SSIM, LAION, rFID, and LPIPS metrics. Overall, our proposed method consistently achieves the best results across all evaluation criteria.

Notably, within the FLUX-VAE framework, the configuration without the reference KL term ("w/o ref KL") performs particularly well on the AIM-500 dataset Li et al. (2021b). The reference KL term is specifically designed to incorporate information from the alpha channel into the latent representation without significantly disrupting the original latent distribution. As shown in Table 3, this design helps to better preserve generation quality during subsequent training in the diffusion model.

## 5 DISCUSSION: EFFECT OF REFERENCE KL ON DIFFUSION TRAINING

As discussed in Sec. 3, the reference KL term $L_{\mathrm{KL}}(E; E_{\mathrm{ref}})$ is introduced to keep the adapted RGBA encoder close to the latent distribution of the pretrained RGB VAE, so that the resulting latents remain compatible with the diffusion prior. While its impact on reconstruction metrics is numerically small (cf. Tab. 3), it plays a more important role during diffusion training.

To better understand this effect, we compare the training dynamics of two text-to-RGBA diffusion models that are *identical* in architecture and optimization schedule, but are initialized with VAEs trained (i) with the reference KL (*ours*) and (ii) without it (*w/o Ref KL*). Fig. 6 shows the smoothed diffusion training loss as a function of training steps on the same dataset. Across the entire training trajectory, the model using our reference-KL VAE consistently attains a lower diffusion loss and

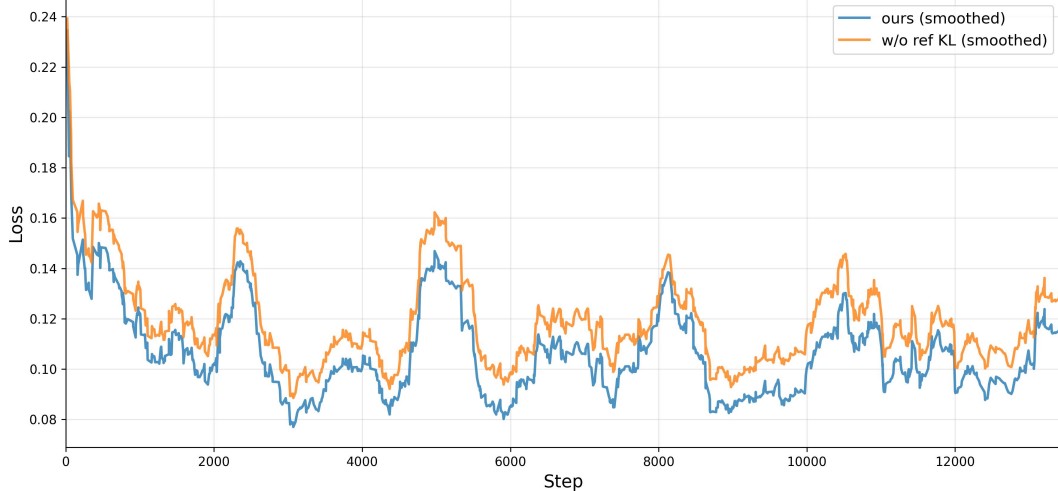

Figure 6: Training loss comparison of diffusion models using VAEs trained with and without the reference KL term.

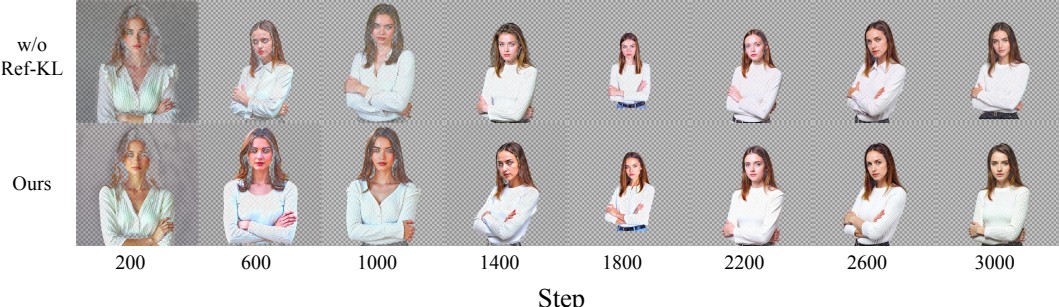

Figure 7: Qualitative comparison of diffusion models using VAEs trained with and without the reference KL term across training steps.

converges faster: the blue curve (*ours*) lies below the orange curve (*w/o Ref KL*) for almost all steps, especially in the early and mid stages of training.

In addition, Fig. 7 visualizes the evolution of generated RGBA samples over the course of training. We decode text-to-RGBA outputs every 400 steps starting from step 200, with the top row showing the model *without* the reference KL and the bottom row showing *ours*. The model without the reference KL fails to learn a correct alpha mask for the human subject even after many steps: the interior of the body remains partially transparent and exhibits ghosting artifacts, instead of having an opaque foreground (alpha $\approx 1$) with clean boundaries. In contrast, the model using our reference-KL VAE quickly learns a sharp, nearly opaque foreground alpha and stable edges, and its visual quality improves much faster across columns.

## 6   CONCLUSION

In this paper, we address the long-standing challenge of transparent image evaluation. We first introduce a new evaluation metric for RGBA images that seamlessly extends standard RGB measures, such as PSNR, SSIM, and FID, via alpha blending. And we present ALPHAVAE, a unified end-to-end RGBA VAE architecture that augments a pretrained three-channel VAE with a dedicated alpha channel and specialized losses to preserve latent fidelity and achieve high-quality reconstruction. Through extensive experiments on reconstruction and image generation tasks, we demonstrate that, with only 8K training images, our method outperforms state-of-the-art approaches trained on orders of magnitude more data in both VAE reconstruction and transparent image generation when fine-tuned within a latent diffusion framework. Our work not only fills a critical gap in evaluation protocols and model design for RGBA image reconstruction and synthesis but also lays the groundwork for future extensions to dynamic, multi-layer video generation and interactive editing applications.

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

# Appendix

## A  FORMULA DERIVATION

In this section, we derive Equation equation 8. Assuming the three channels of the background color $b \in \mathbb{R}^3$ are independent and identically distributed, we can work on a single pixel without lack of generality. Define

$$
\begin{cases}
\mathcal{L}_{\text{rec}}(\mathcal{E}, \mathcal{D}) = \mathbb{E}_b\Big[\big\|\mathcal{A}(\hat{x}, b) - \mathcal{A}(x, b)\big\|_2^2\Big], \\
\mathcal{A}(\hat{x}, b) - \mathcal{A}(x, b) = P - \Delta_\alpha b, \\
P = \hat{x}_{\text{rgb}} \odot \hat{x}_\alpha - x_{\text{rgb}} \odot x_\alpha \in \mathbb{R}^3, \\
\Delta_\alpha = \hat{x}_\alpha - x_\alpha \in \mathbb{R}.
\end{cases}
\tag{17}
$$

Substituting the second line of equation 17 into the first gives

$$
\begin{aligned}
\mathcal{L}_{\text{rec}}(\mathcal{E}, \mathcal{D}) &= \mathbb{E}_b\Big[\big\|P - \Delta_\alpha b\big\|_2^2\Big] \\
&= \mathbb{E}_b\Big[(P - \Delta_\alpha b)(P - \Delta_\alpha b)^\top\Big] \\
&= \mathbb{E}_b\Big[PP^\top - 2\Delta_\alpha\, bP^\top - \Delta_\alpha^2 bb^\top\Big] \\
&= \|P\|_2^2 - 2\Delta_\alpha \langle \mathbb{E}[b], P\rangle - \Delta_\alpha^2\, \mathbb{E}[bb^\top].
\end{aligned}
\tag{18}
$$

Because $\mathbb{E}[bb^\top] = \mathbb{E}[b_1^2 + b_2^2 + b_3^2] = \|\mathbb{E}[b^2]\|_1$, Equation equation 8 follows immediately.

For implementation, we use the equivalent channel-wise form

$$
\begin{aligned}
\mathcal{L}_{\text{rec}}(\mathcal{E}, \mathcal{D}) &= \|P\|_2^2 - 2\Delta_\alpha \langle \mathbb{E}[b], P\rangle - \Delta_\alpha^2 \|\mathbb{E}[b^2]\|_1 \\
&= \|P^2\|_1 - \mathbb{E}_{\text{ch}}\big[2\Delta_\alpha\, \mathbb{E}[b] \odot P\big] - \Delta_\alpha^2 \|\mathbb{E}[b^2]\|_1 \\
&= \mathbb{E}_{\text{ch}}\Big[P^2 - 2\Delta_\alpha\, \mathbb{E}[b] \odot P - \Delta_\alpha^2\, \mathbb{E}[b^2]\Big],
\end{aligned}
\tag{19}
$$

where $\mathbb{E}_{\text{ch}}$ denotes the expectation over the RGB channels.

## B  DETAILS OF HYPERPARAMETERS.

In our reconstruction loss, the background pixel distribution $b$ contributes two pre-computable terms: its first raw moment, $\mathbb{E}[b]$, and its second raw moment, $\mathbb{E}[b^2]$. Figure 8 visualizes the empirical distribution of $b$ on the ImageNet training split. Throughout training we rescale pixel intensities to the range $[-1, 1]$, yielding

$$
\mathbb{E}[b] = (-0.0357, -0.0811, -0.1797) \quad \text{and} \quad \mathbb{E}[b^2] = (0.3163, 0.3060, 0.3634).
\tag{20}
$$

In addition to the background pixel statistics, we detail the weighting of various loss terms introduced in Section 3. Specifically, we use a reconstruction loss with a weight of 1.0, a perceptual loss (computed using LPIPS) with a weight of 0.5, and a composite regularization loss comprising two KL divergence terms: the primary latent KL loss is weighted by $10^{-6}$, while the reference KL loss is assigned a much smaller weight of $10^{-16}$. For adversarial training, we enable the GAN loss after 4000 steps and set the generator loss weight to 1.0.

## C  VALIDATION OF LPIPS ON ALPHA-BLENDED IMAGES

When evaluating RGBA image reconstruction quality, the transparency channel requires special consideration for perceptual metrics. Since LPIPS operates on RGB images, we must first composite the RGBA image onto a background before computing the metric. This section validates our choice of using black and white background averaging (BW-Avg) as an efficient and accurate evaluation strategy.

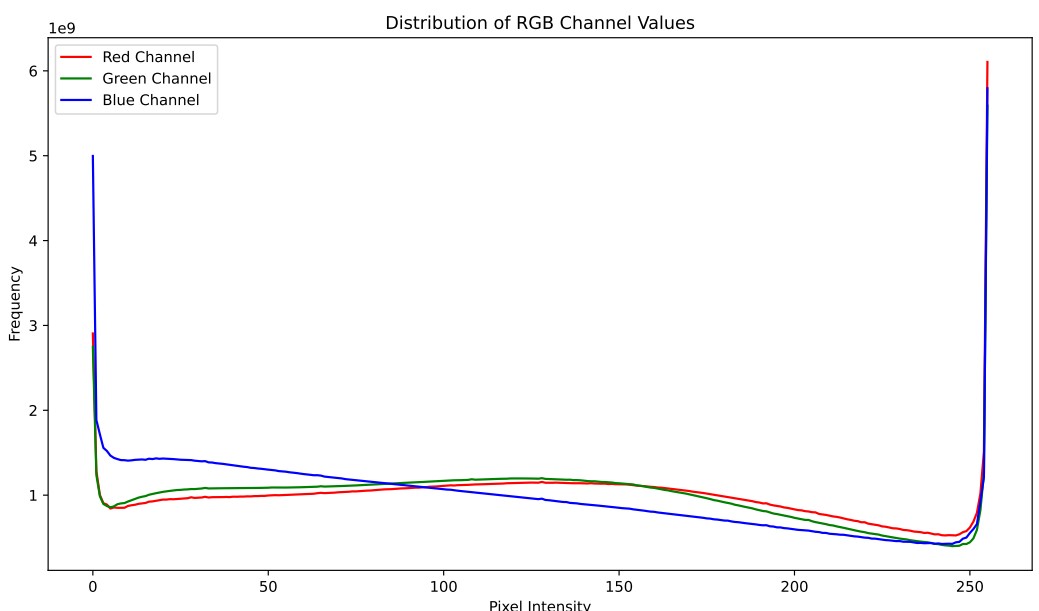

Figure 8: Color distribution of pixels in ImageNet train split. The pixel values are scaled to $[0, 255]$.

## C.1 EVALUATION METHODS

We compare three approaches for computing LPIPS on RGBA images:

- **RGB-only**: Using only the RGB channels without considering transparency.
- **BW-Avg (Ours)**: Computing LPIPS on both black and white background composites, then averaging:

$$\mathcal{L}_{\text{BW-Avg}} = \frac{1}{2} \left[ \text{LPIPS}(I_{\text{black}}, \hat{I}_{\text{black}}) + \text{LPIPS}(I_{\text{white}}, \hat{I}_{\text{white}}) \right] \tag{21}$$

where $I_{\text{black}} = \text{RGB} \cdot \alpha$ and $I_{\text{white}} = \text{RGB} \cdot \alpha + (1 - \alpha)$.

- **Real-BG**: Compositing with a realistic, contextually appropriate background image.

## C.2 EXPERIMENTAL SETUP

**Real Background Generation.** To obtain realistic backgrounds for comparison, we use Nano Banana Pro, which takes an RGBA image with transparency as input and generates a contextually appropriate background harmonizing with the foreground object.

**Test Configuration.** We use a representative RGBA image containing a foreground object with varying transparency levels, including semi-transparent regions and sharp edges. For the sensitivity experiment, we apply Gaussian noise with standard deviations ranging from 0.001 to 0.1. For the background consistency experiment, we apply Gaussian noise with std = 0.02 and evaluate across 13 different backgrounds (solid colors, random colors, and a realistic generated background).

## C.3 RESULTS

Figure 9 presents our experimental results.

## C.4 ANALYSIS

**Sensitivity.** LPIPS reliably detects minimal distortions, with measurable differences even at noise std = 0.001. This confirms its effectiveness for quality assessment on composited images.

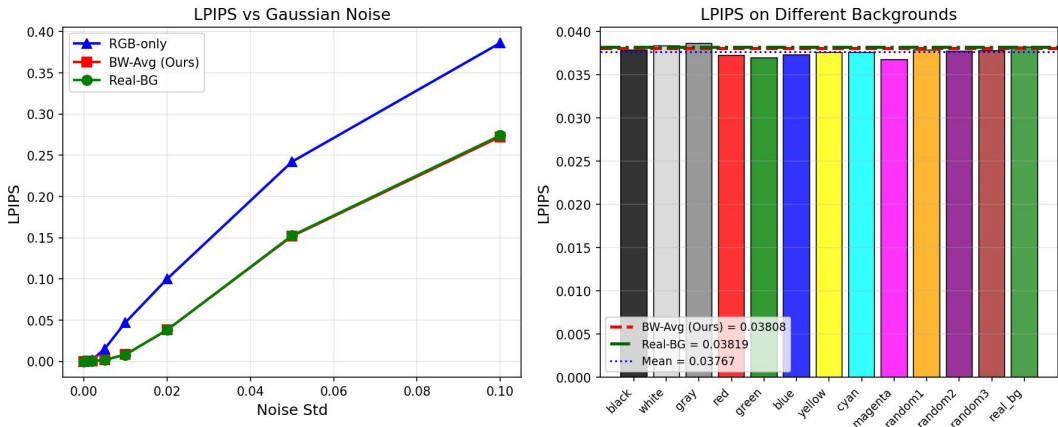

Figure 9: Validation of LPIPS on alpha-blended RGBA images. **Left**: LPIPS as a function of Gaussian noise standard deviation. All three methods (RGB-only, BW-Avg, and Real-BG) exhibit strict monotonic increase, with BW-Avg and Real-BG curves nearly overlapping. Even extremely small perturbations (std = 0.001) are detectable. **Right**: LPIPS evaluated on 13 different backgrounds with fixed noise (std = 0.02). Our BW-Avg method (0.03808) differs from real background evaluation (0.03819) by only 0.27%.

**Monotonicity.** All three evaluation methods exhibit strictly monotonic behavior—larger distortions consistently yield higher LPIPS values, as expected from a reliable perceptual metric.

**Method Equivalence.** The BW-Avg method (0.03808) differs from realistic background evaluation (0.03819) by only 0.27%. This negligible difference validates that our simpler approach accurately approximates perceptual quality assessment with realistic backgrounds.

**Background Consistency.** Across 13 different backgrounds, LPIPS exhibits a coefficient of variation of only 1.35%, demonstrating high consistency regardless of background choice.

### C.5 DESIGN RATIONALE

We adopt Black-White averaging for the following reasons:

- **Computational Efficiency**: Realistic background generation requires additional model inference, introducing significant overhead during training and evaluation. Solid-color compositing requires only element-wise operations.

- **Empirical Equivalence**: BW-Avg yields results within 0.27% of real background evaluation, making the additional computational cost unjustified.

- **Reproducibility**: Solid colors provide deterministic evaluation without dependence on external models or datasets.

- **Extreme Case Coverage**: Black and white represent luminance extremes. Averaging these effectively covers the range of possible background conditions.

These experiments confirm that LPIPS computed via Black-White averaging is both computationally efficient and empirically equivalent to evaluation with realistic backgrounds.

## D DETAILS OF DATASET.

**Datasets collection.** High-quality RGBA training data is scarce due to the difficulty of acquiring accurate alpha matte and consistent foreground transparency in the wild. However, we observe that existing high-quality image matting datasets, originally designed for alpha prediction, can be effectively repurposed for RGBA generation. These datasets typically contain paired foreground and alpha-mask

Table 4: Quantitative comparison between the original dataset and the refined dataset.

| Metric | Original Dataset | Refined Dataset |
|---|---|---|
| Total Images | 9,236 | 8,124 |
| Defective Alpha Mask Rate ($\downarrow$ better) | 6.62% (612/9236) | 0% |
| Aesthetic Quality (LAION-AES, $\uparrow$ better) | 4.9062 | 5.0636 |
| Pre-processing (Decontamination, Blurring) | $\times$ | $\checkmark$ |

Table 5: Statistics of each dataset in our dataset. Resolution indicates the average image dimensions in each dataset, computed as the mean height multiplied by the mean width of all images.

| Dataset | # Train Images | # Test Images | Resolution |
|---|---|---|---|
| Adobe Image Matting dataset | 449 | 23 | 1292×1082 |
| AM-2K | 1,900 | 100 | 1471×1195 |
| Distinctions-646 | 607 | 31 | 1569×1732 |
| HHM-2K | 1,900 | 100 | 3570×4041 |
| Human-1K | 953 | 50 | 2060×2094 |
| P3M-500-NP | 475 | 25 | 1374×1313 |
| PhotoMatte85 | 81 | 4 | 2304×3456 |
| realWorldPortrait-636 | 605 | 31 | 1038×1327 |
| SIMD | 318 | 16 | 2346×2079 |
| Transparent-460 | 434 | 22 | 3799×3767 |
| **Total** | 7,722 | 402 | 2176×2240 |

images extracted from real-world compositions. To convert such data into RGBA format, we simply combine the RGB foreground $I_{\text{fg}}$ and the corresponding alpha matte $\alpha$ to form a four-channel image $x = \text{concat}(I_{\text{fg}}, \alpha)$. This approach ensures that the resulting RGBA images are aligned, photorealistic, and contain diverse transparency patterns. We choose **8,124** high-quality images, from ten image matting datasets, including Adobe Image Matting dataset (Xu et al., 2017), AM-2K (Li et al., 2022), Distinctions-646 (Qiao et al., 2020), HHM-2K (Sun et al., 2023), Human-1K (Liu et al., 2021), P3M-500-NP (Li et al., 2021a), PhotoMatte85 (Lin et al., 2021), realWorldPortrait-636 (Yu et al., 2020), SIMD (Sun et al., 2021) and Transparent-460 (Cai et al., 2022), covering a broad range of object categories and transparency patterns. All images are preprocessed into four-channel RGBA format by compositing foregrounds with their corresponding alpha mattes.

**Dataset Refinement**. We perform a series of refinement steps to enhance data quality and ensure supervision consistency. First, we apply color decontamination and background blurring to all 9,236 original images, improving boundary sharpness and edge fidelity. Next, we discard 612 samples with defective alpha masks (e.g., subjects occluded or severed by fences). Finally, we conduct aesthetic filtering by computing LAION-AES scores, identifying the 1,000 lowest-scoring images, and manually excluding 500 of them. After these refinements, the curated dataset comprises 8,124 RGBA images out of approximately 9,200 initial samples. The statistics are summarized in Table 4, which highlights improvements in edge quality, mask validity, and aesthetic quality.

**Train/Test split.** For each dataset, we follow a standardized protocol to reserve a small portion for testing during training. Specifically, we select the test subset as 5% of the dataset size. This ensures sufficient coverage for testing even in smaller datasets, while preserving training diversity. The final split yields **7,722** training images and **402** testing images.

**Overview.** We present the dataset statistics for training and evaluation in Table 5. The training set, comprises 7,722 RGBA images spanning diverse domains, with an emphasis on challenging transparency effects such as hair, glass, and soft shadows. The evaluation set, includes 402 RGBA images exhibiting similar complexities. This comprehensive setup allows us to assess the adaptability of AlphaGen across varied transparent image generation scenarios.

## E    LIMITATIONS

Our study focuses exclusively on parameter-efficient fine-tuning with LoRA. Other fine-tuning methods, such as full-parameter fine-tuning and task-specific adaptation modules like ControlNet, are not evaluated because they require significantly more computation. Investigating these approaches is a valuable avenue for future work.

## F    MORE QUALITATIVE ANALYSIS

We present additional qualitative results from Figure 10 to Figure 29, with all images rendered at a resolution of $1024 \times 1024$.

## G    DETAILS OF EXPERIMENTAL RESULTS

We present details of the experimental results from Table 6 to Table 31.

## H    LLM USAGE STATEMENT

A large language model (LLM) is utilized solely to improve grammar and conciseness in certain sections of the text. The LLM does not participate in the development of research ideas, method design, experiment execution, result analysis, or conclusion formulation. All scientific contributions, insights, and conclusions presented in this paper are solely ours.

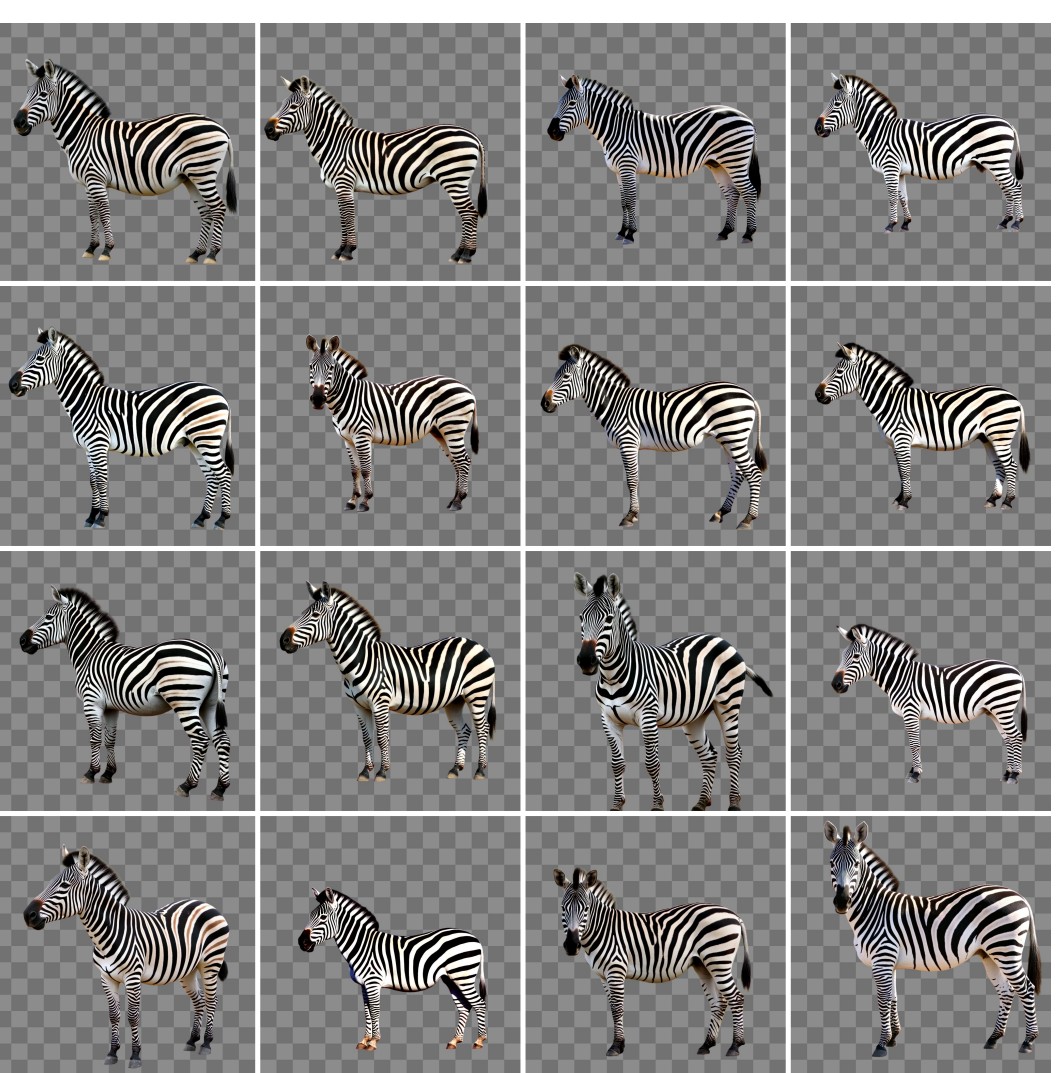

Figure 10: A zebra stands with its head turned to the left, showcasing its distinctive black and white stripes.

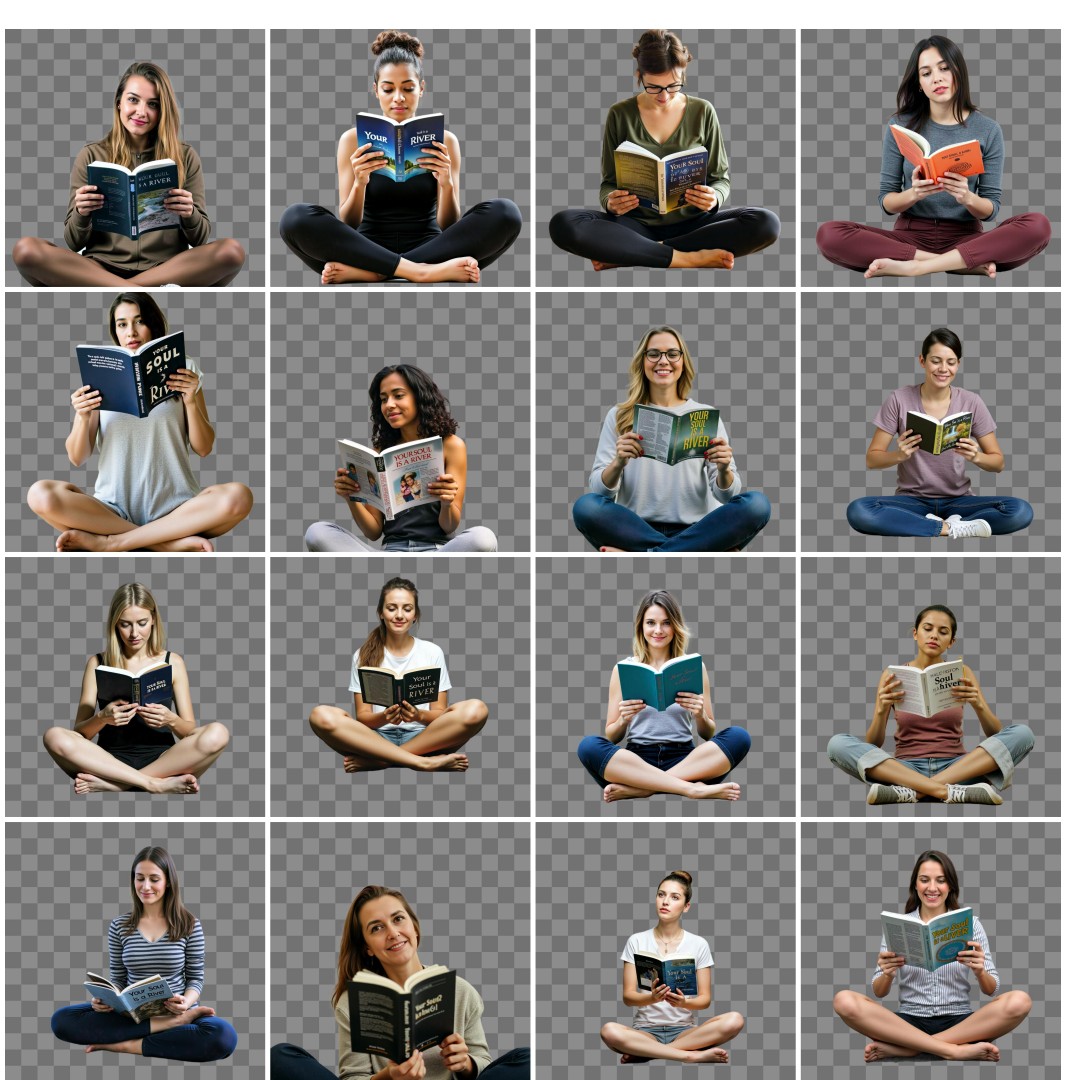

Figure 11: A woman sits cross-legged, reading a book titled "Your Soul is a River."

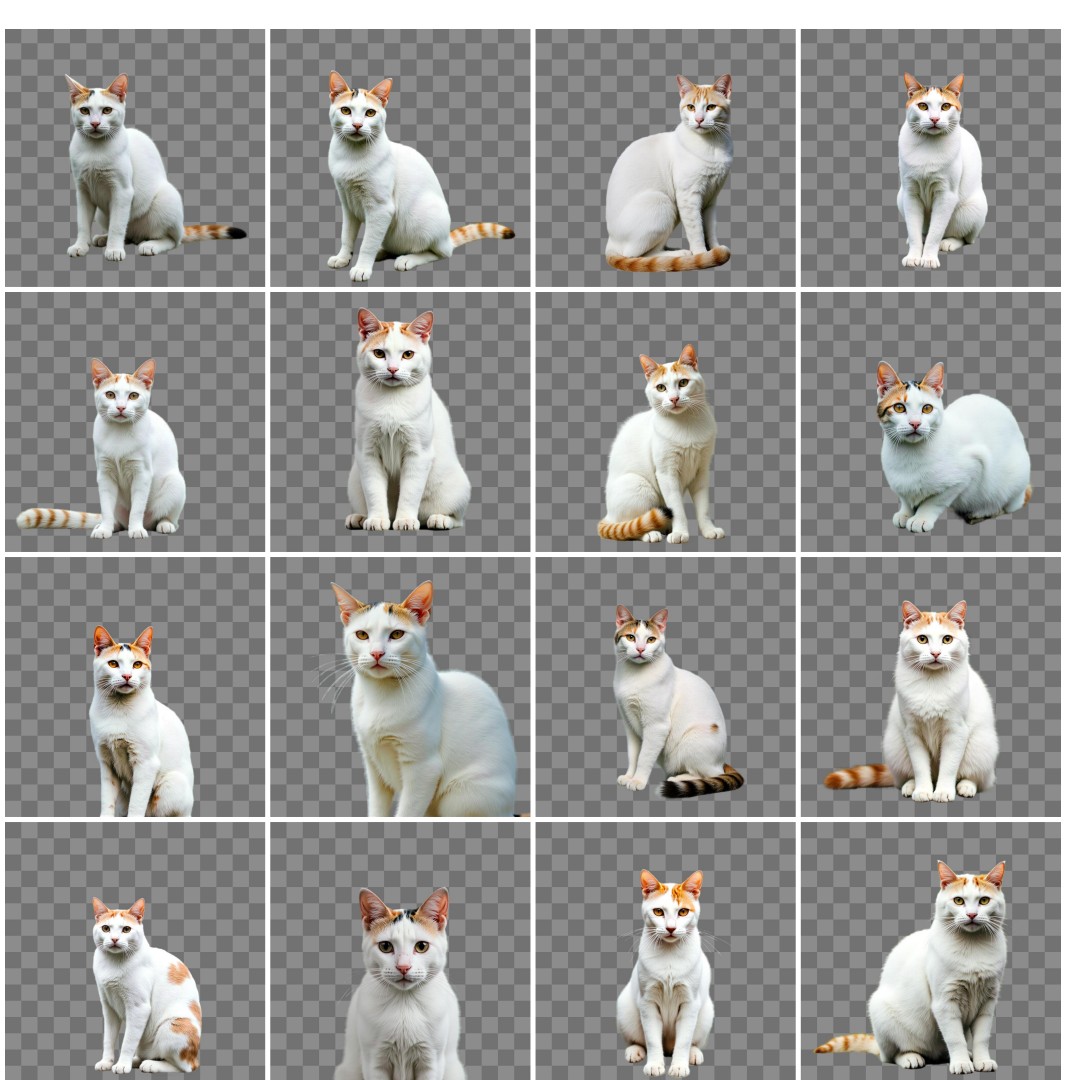

Figure 12: A white cat with orange and black markings sits calmly, gazing forward.

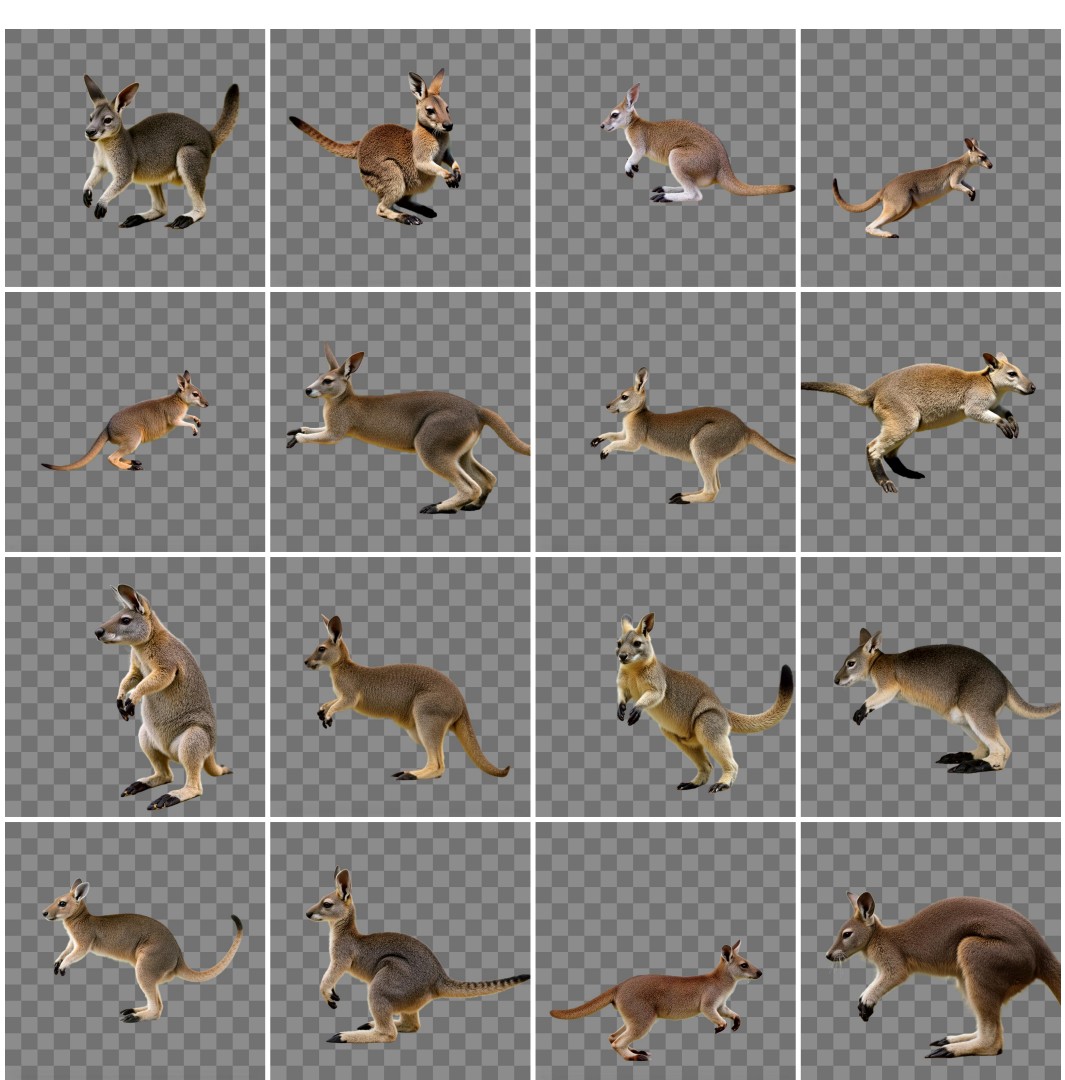

Figure 13: A wallaby is captured mid-hop, showcasing its agile movement and distinctive features.

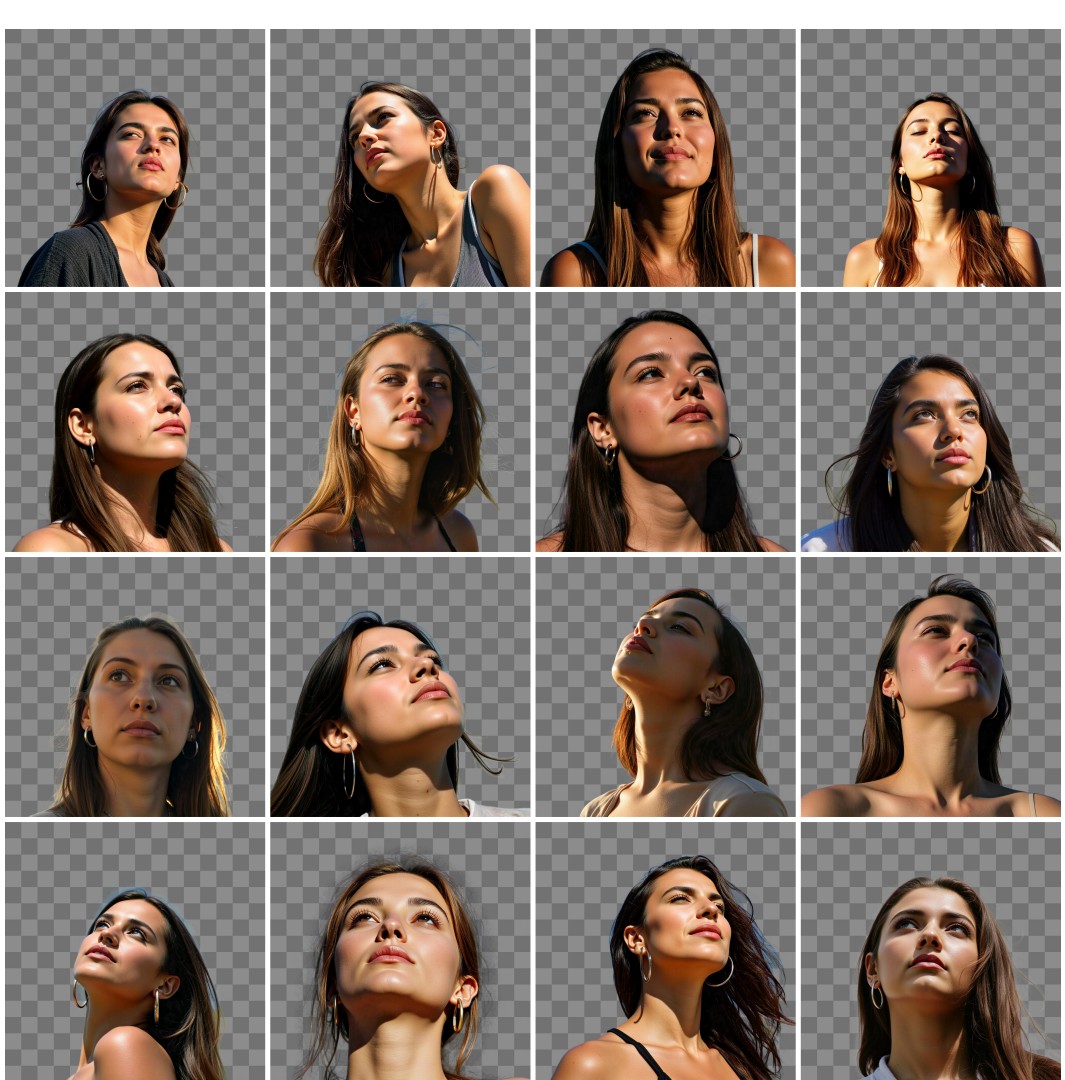

Figure 14: A person with long hair and hoop earrings looks upward, bathed in sunlight.

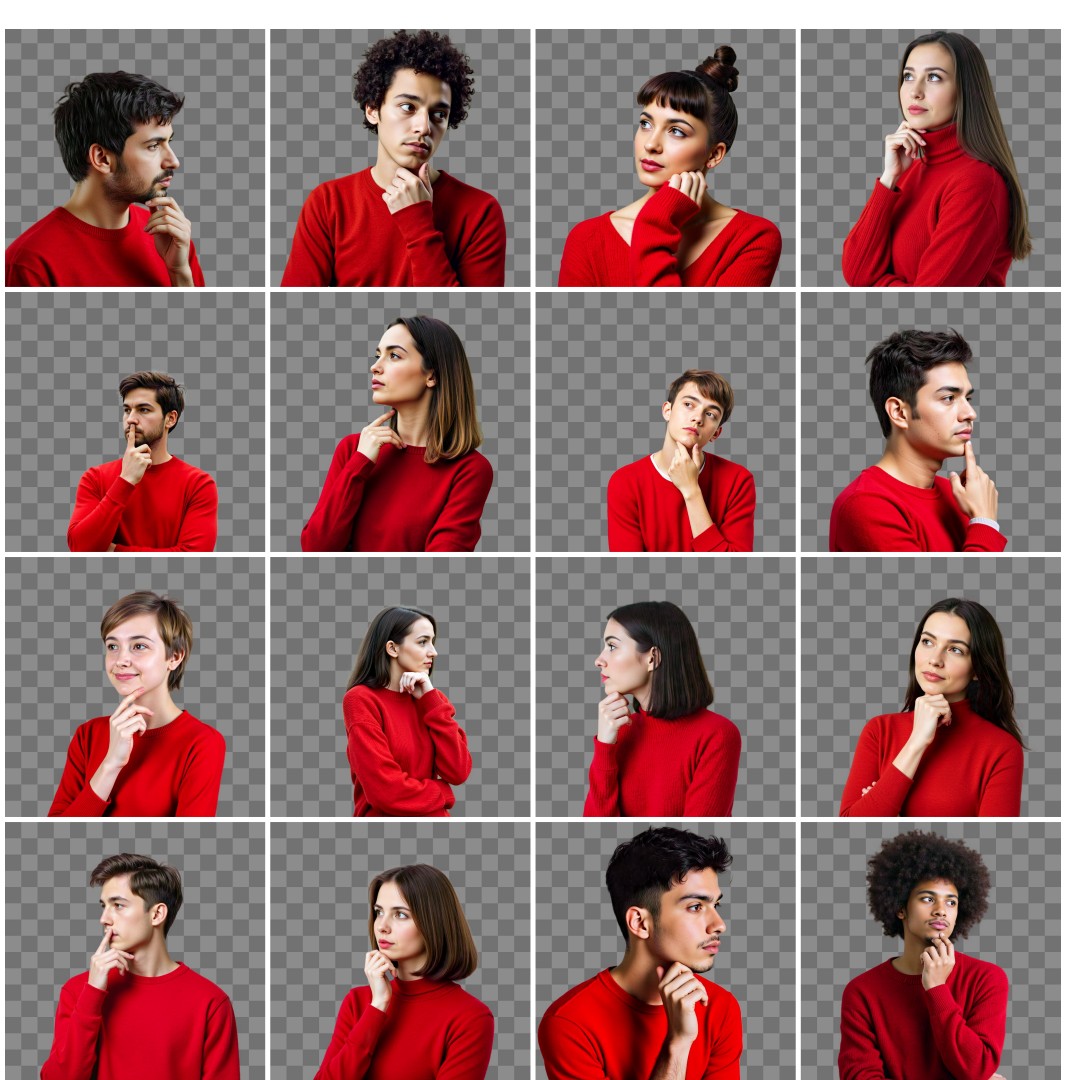

Figure 15: A person wearing a red sweater is looking to the side with their hand on their chin.

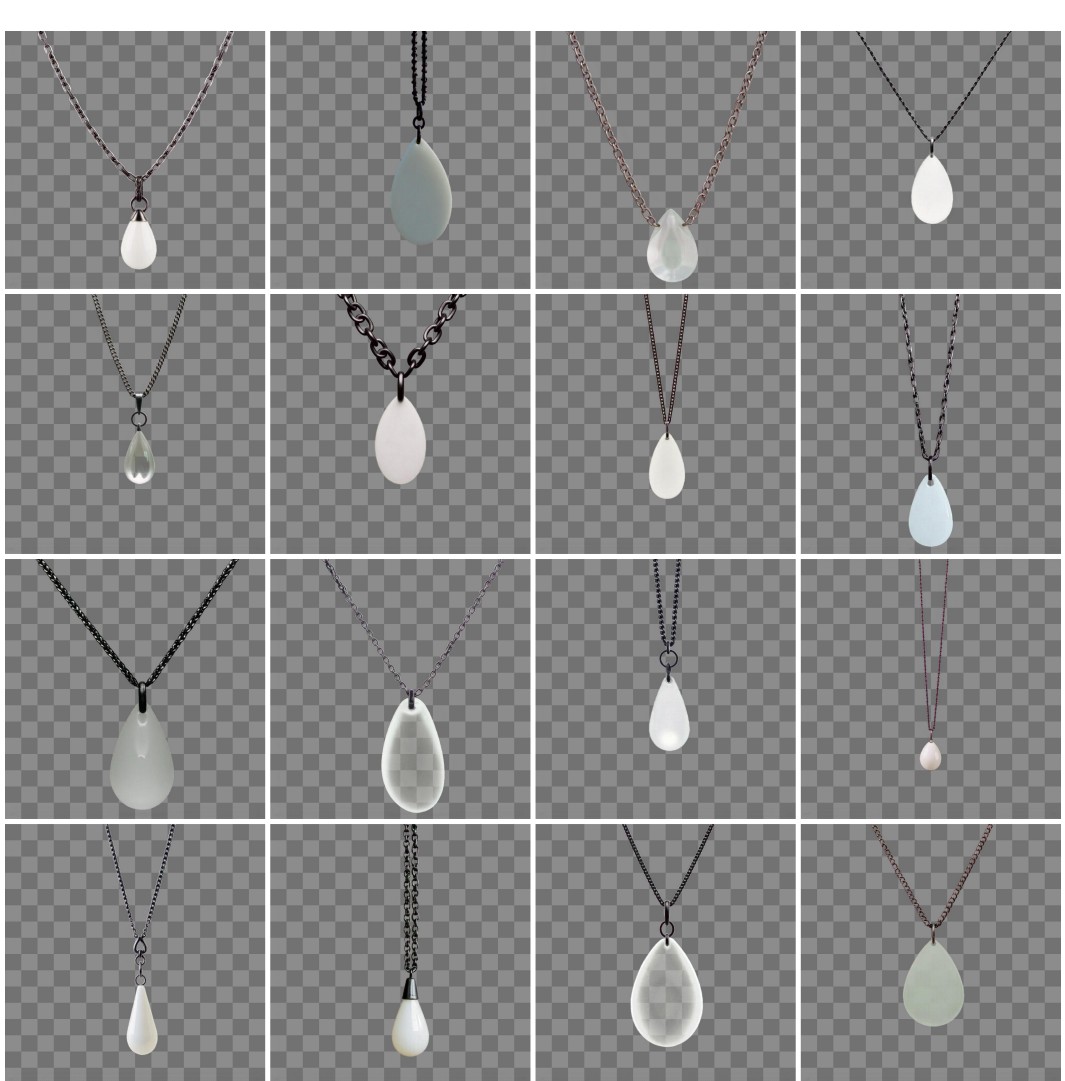

Figure 16: A necklace with a black chain and a white, teardrop-shaped pendant hangs against a white backdrop.

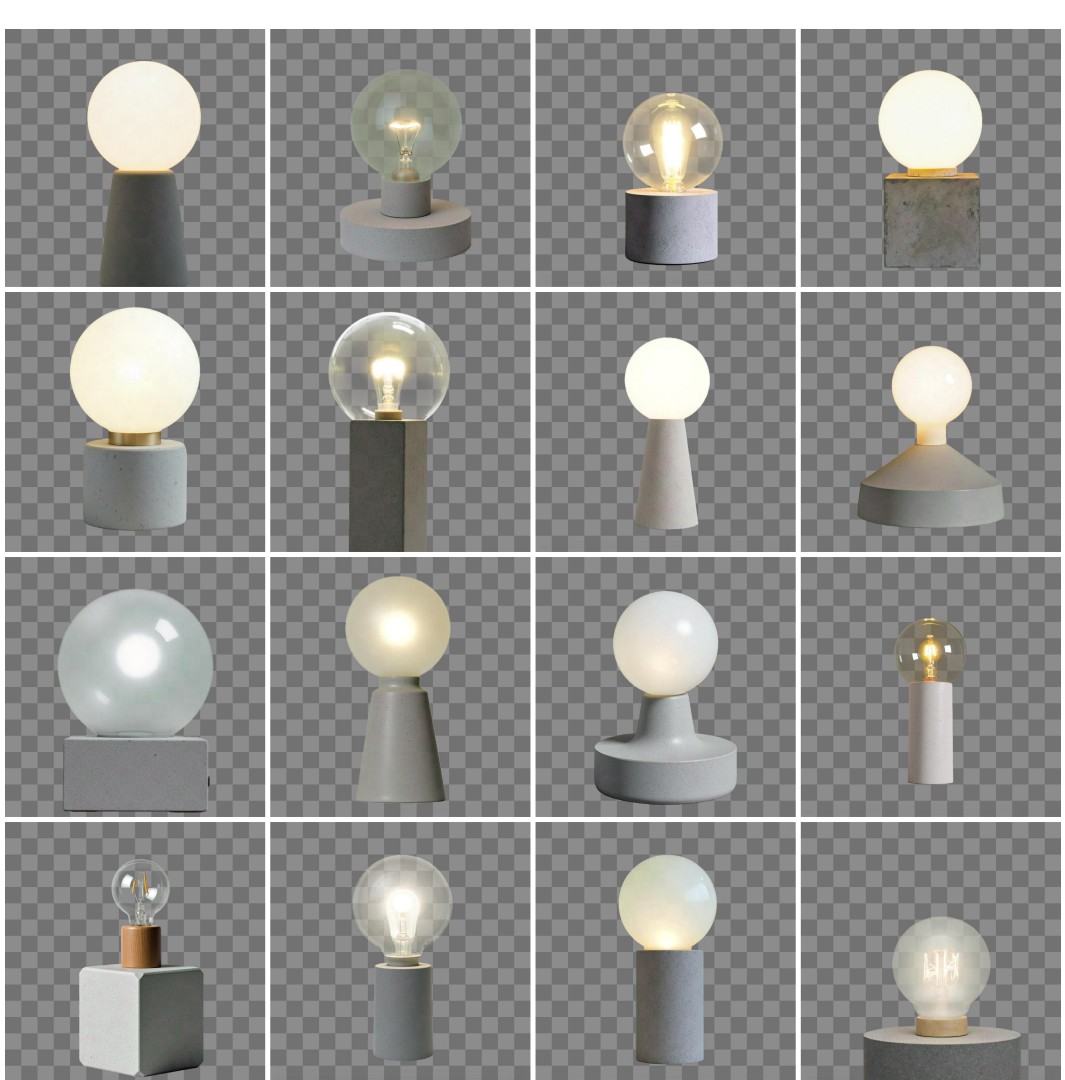

Figure 17: A modern table lamp with a concrete base and a round, frosted glass bulb.

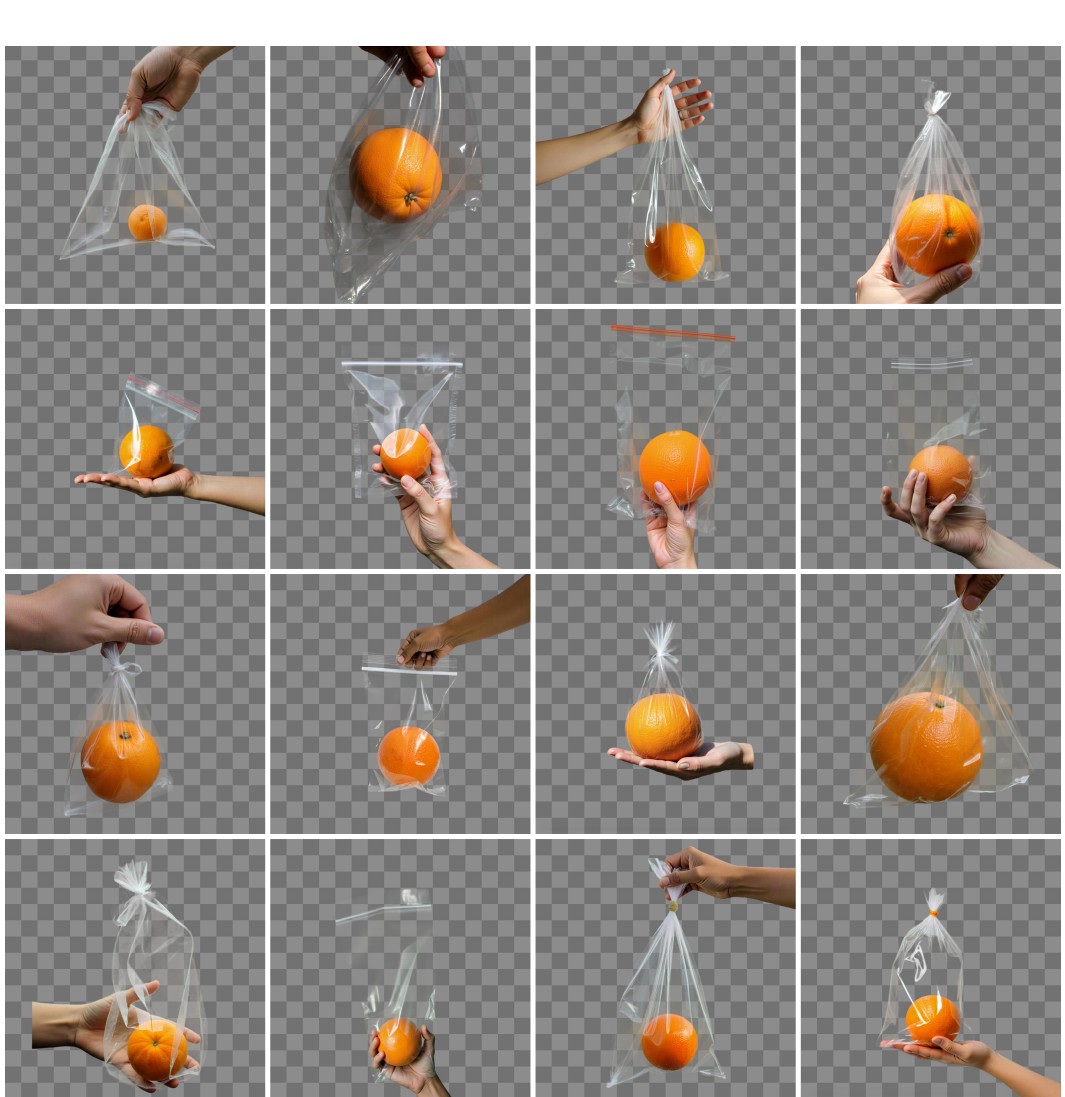

Figure 18: A hand holds a plastic bag containing an orange.

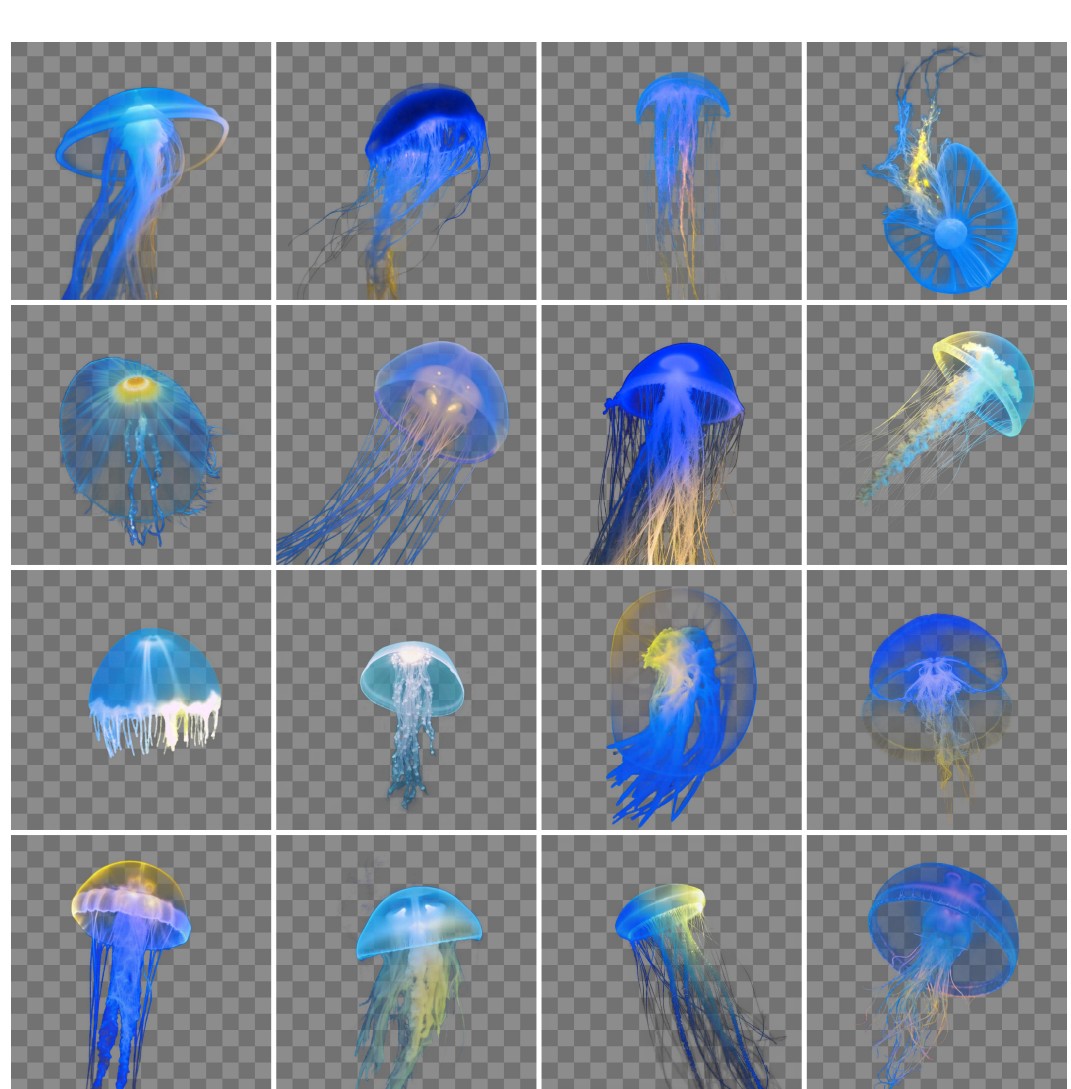

Figure 19: A glowing jellyfish with translucent blue and yellow hues is shown.

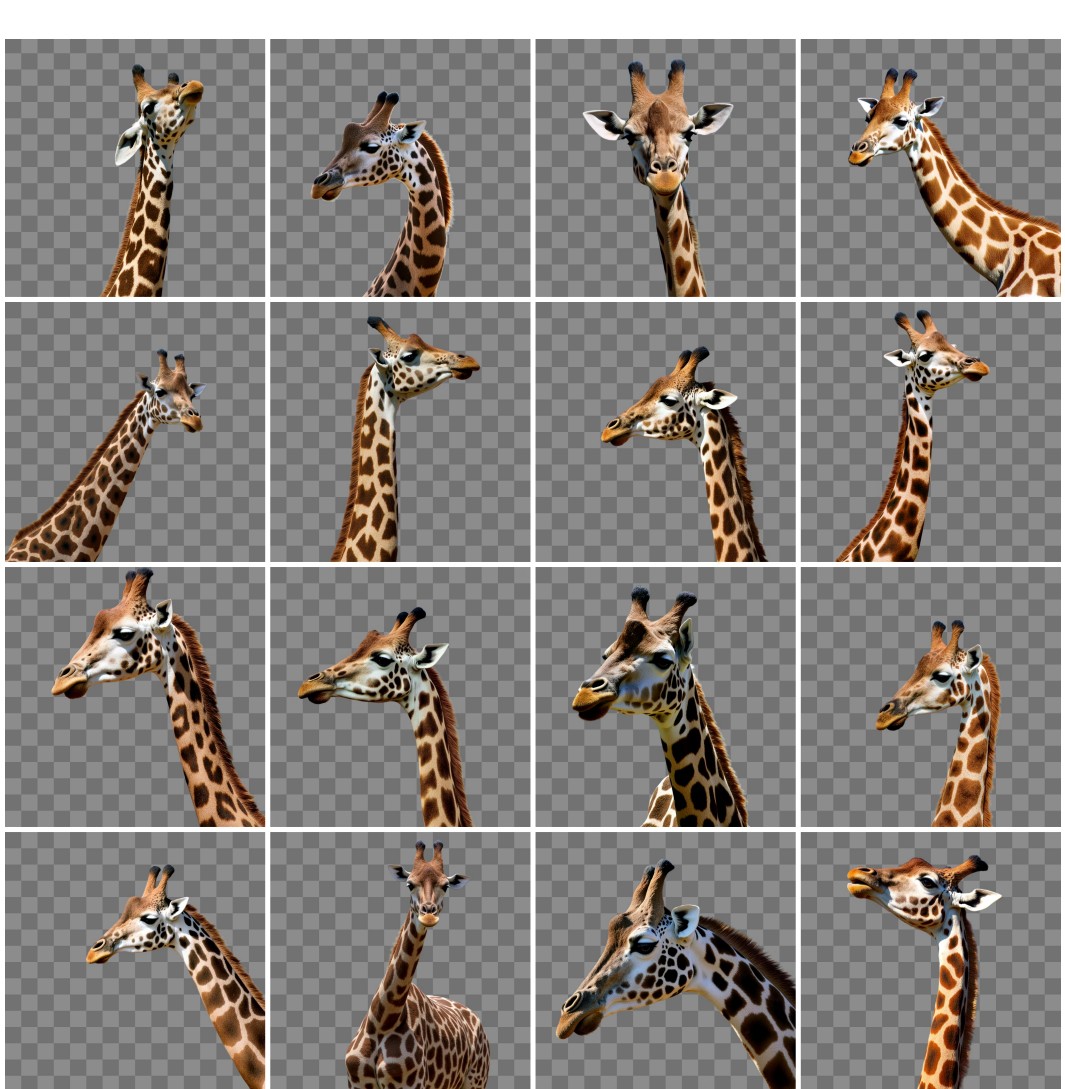

Figure 20: A giraffe with its head turned to the side, showcasing its long neck and distinctive coat pattern.

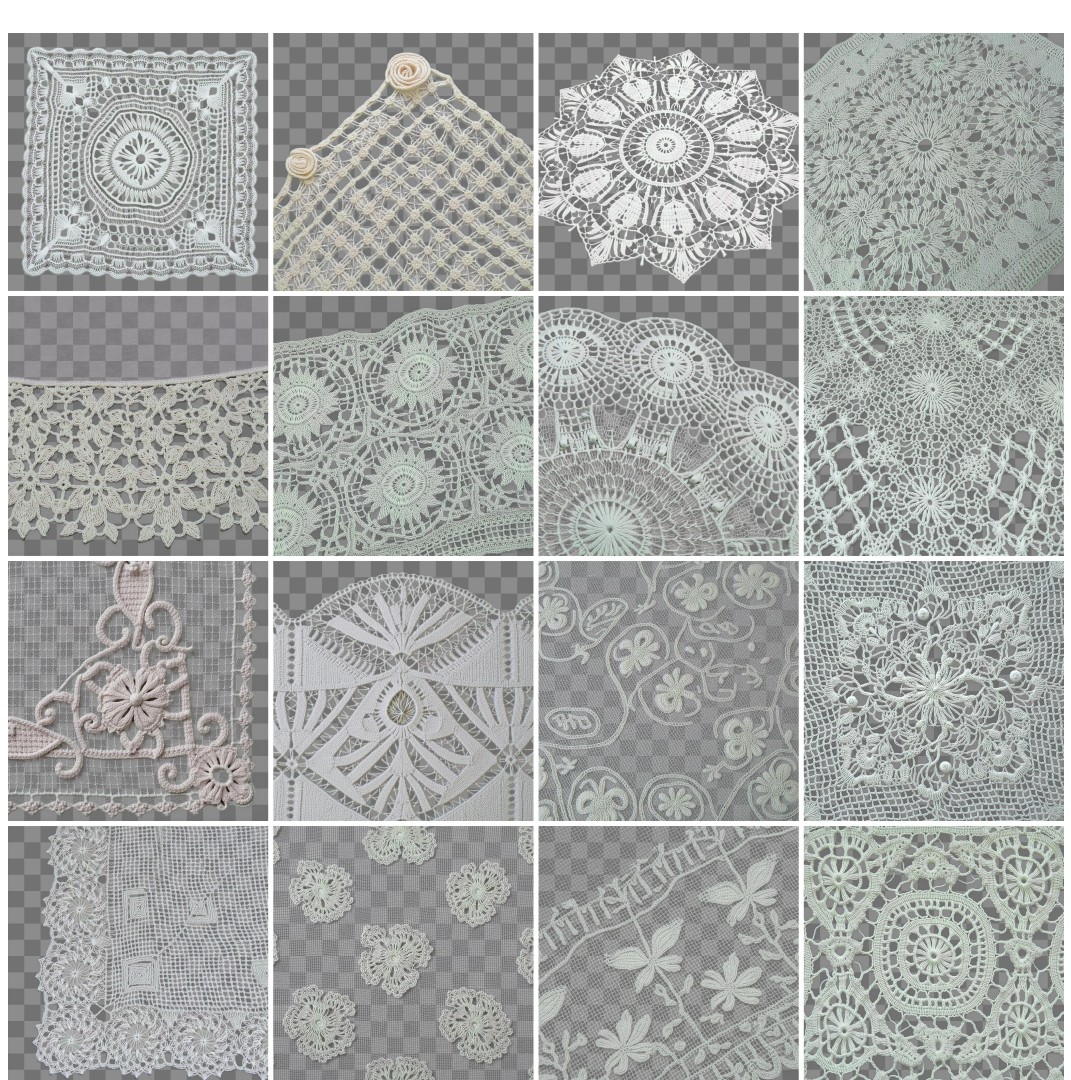

Figure 21: A detailed, intricate crocheted pattern with floral motifs and openwork design.

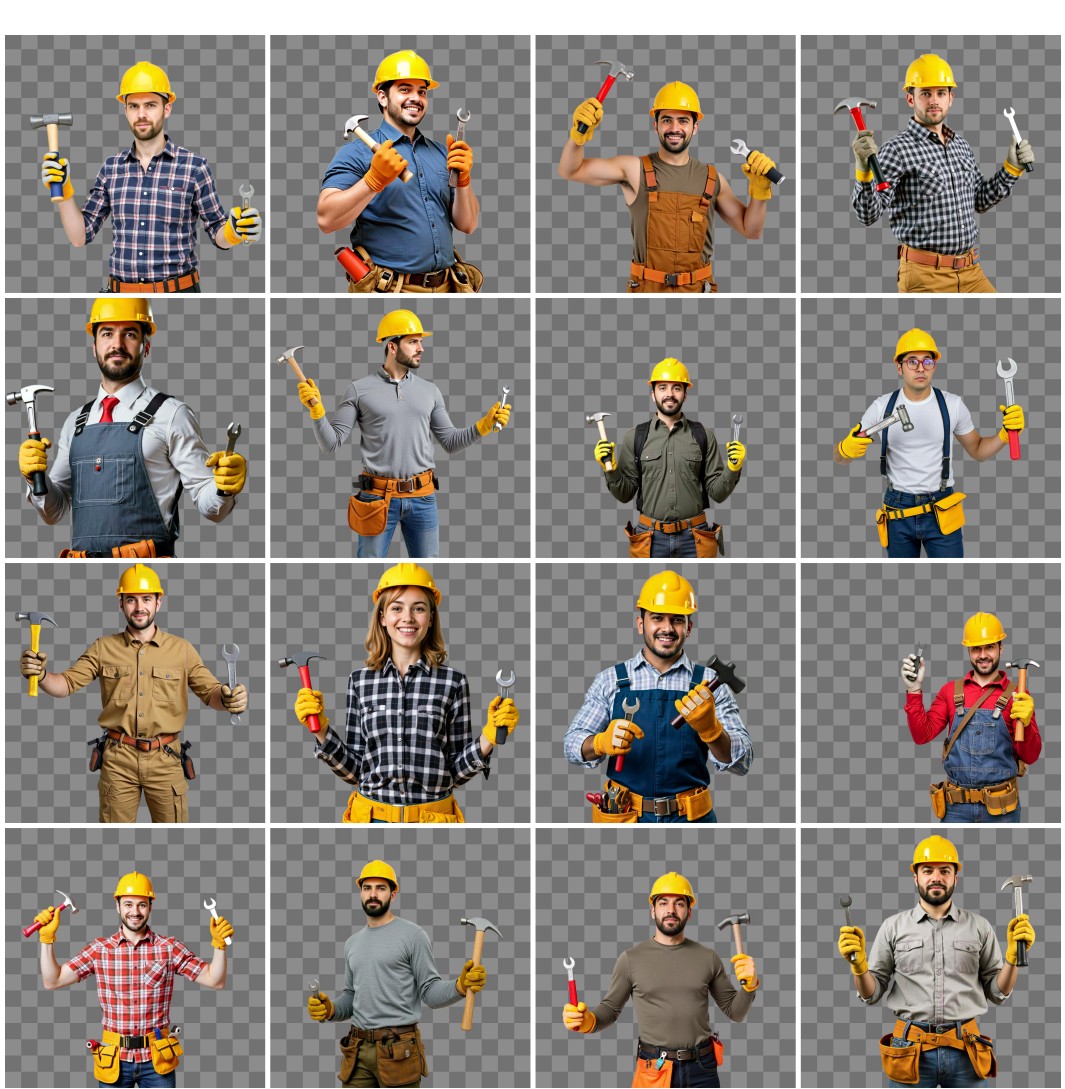

Figure 22: A construction worker wearing a yellow hard hat, gloves, and tool belt holds a hammer and wrench.

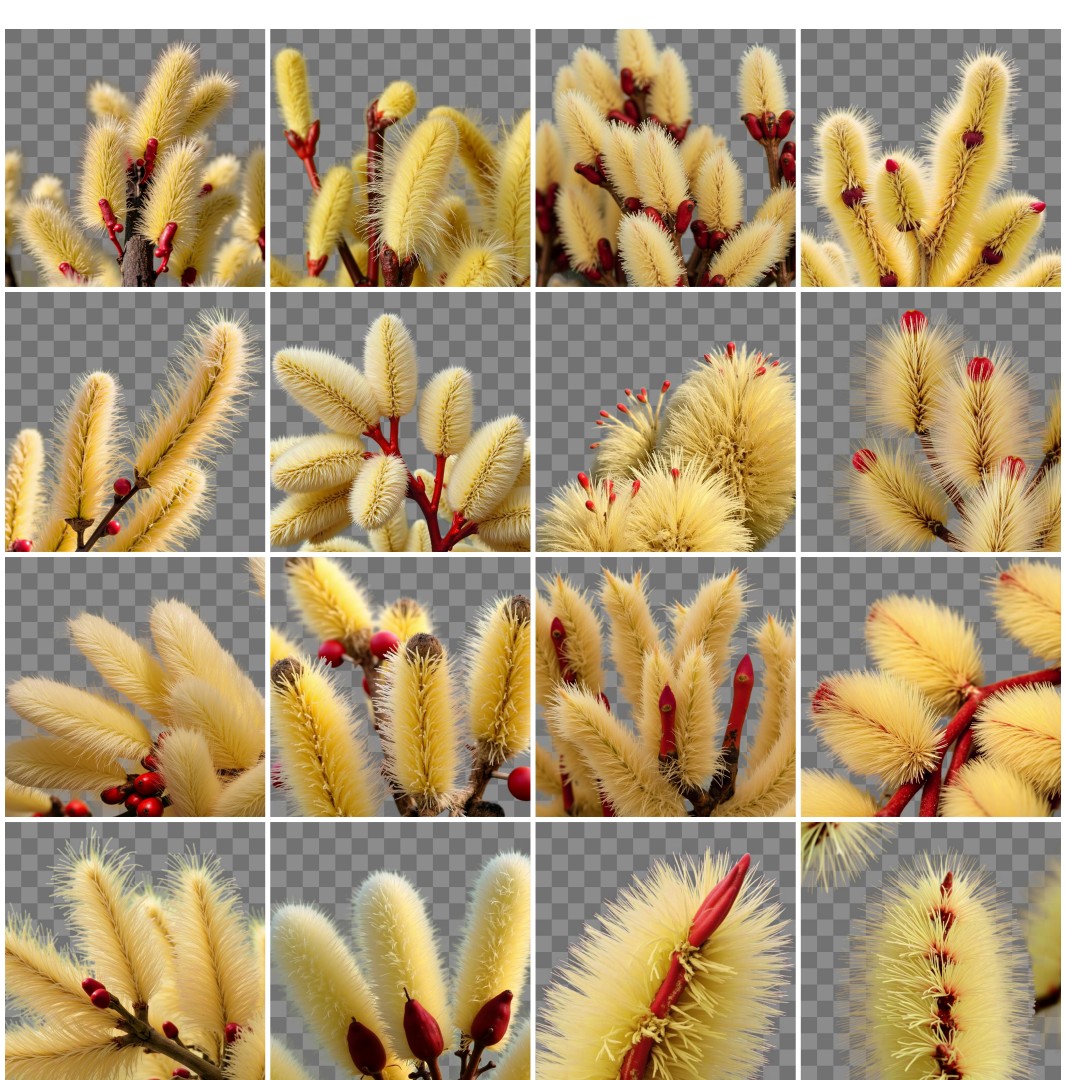

Figure 23: A close-up of fluffy, yellowish-white catkins with red buds.

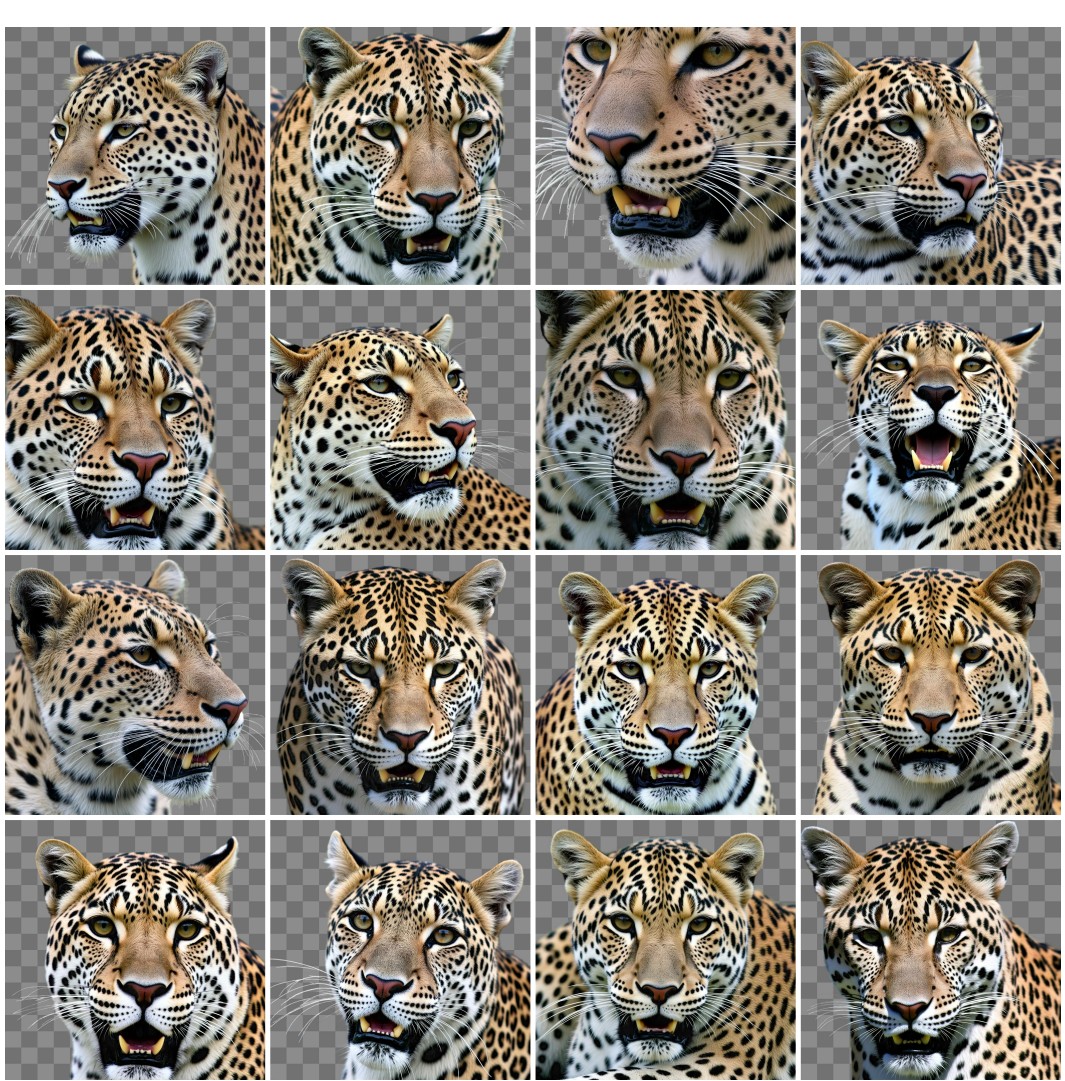

Figure 24: A close-up of a leopard's face with its mouth slightly open, showcasing its sharp teeth and distinctive spotted fur.

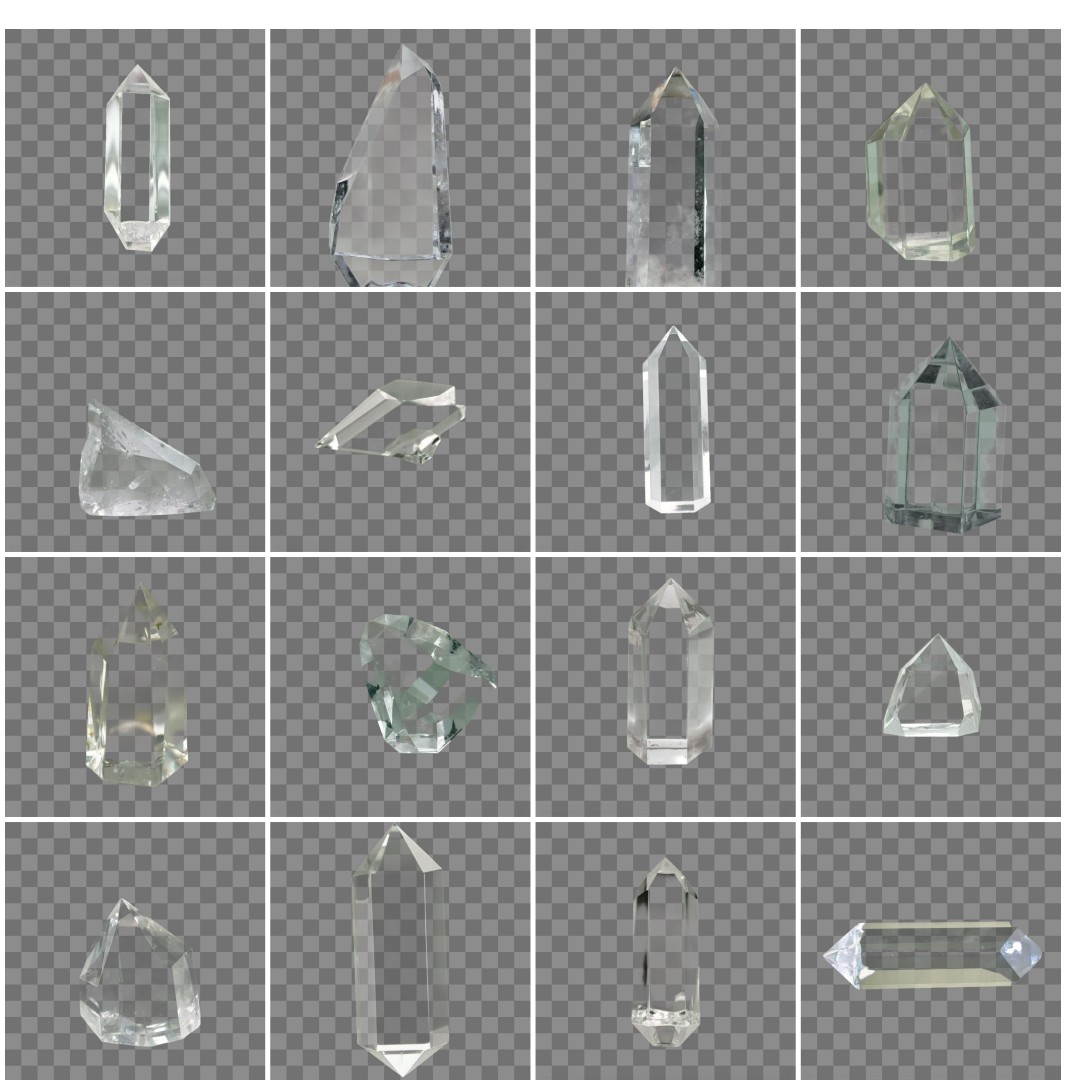

Figure 25: A clear, faceted crystal with smooth edges and a pointed tip is shown.

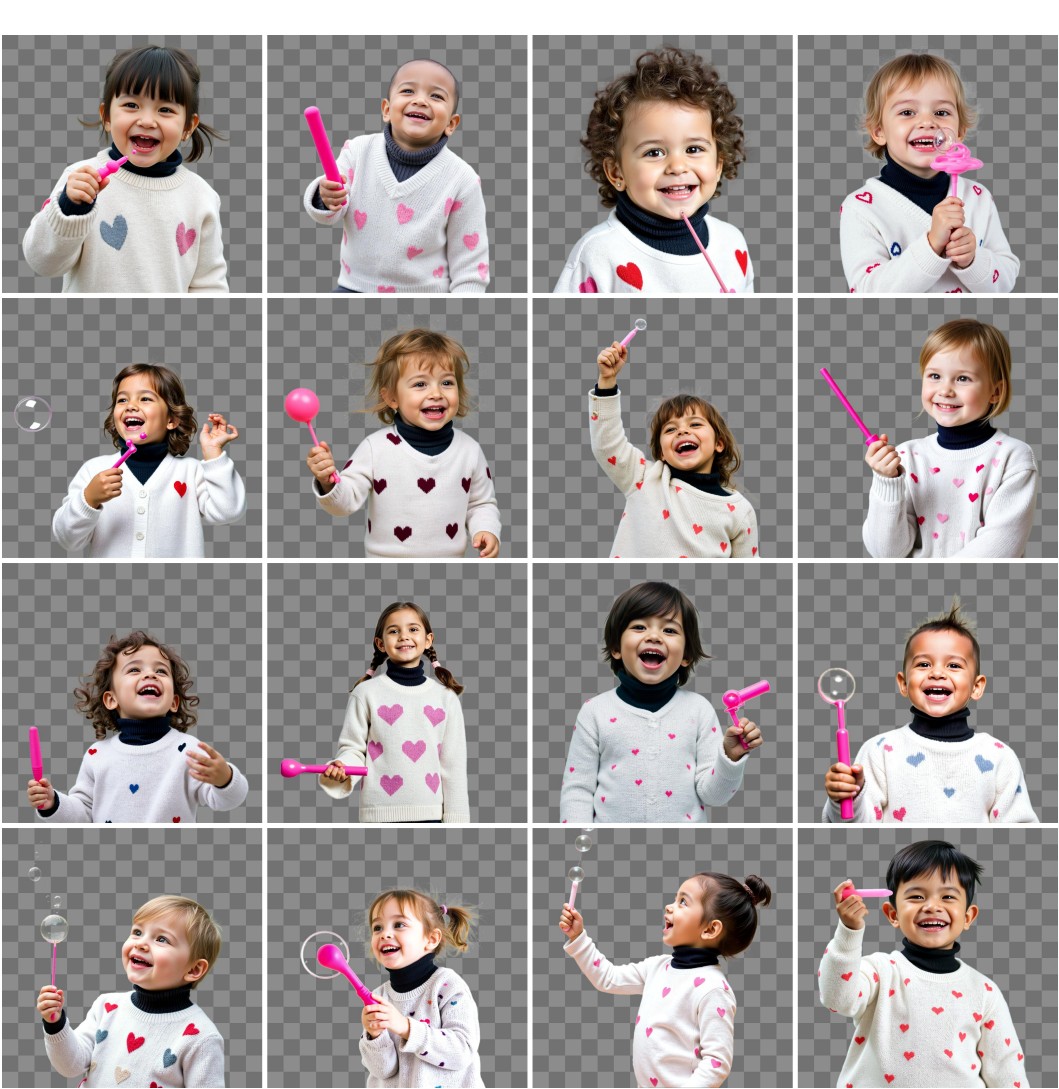

Figure 26: A child is joyfully playing with a pink bubble wand, wearing a white sweater with heart patterns and a dark turtleneck.

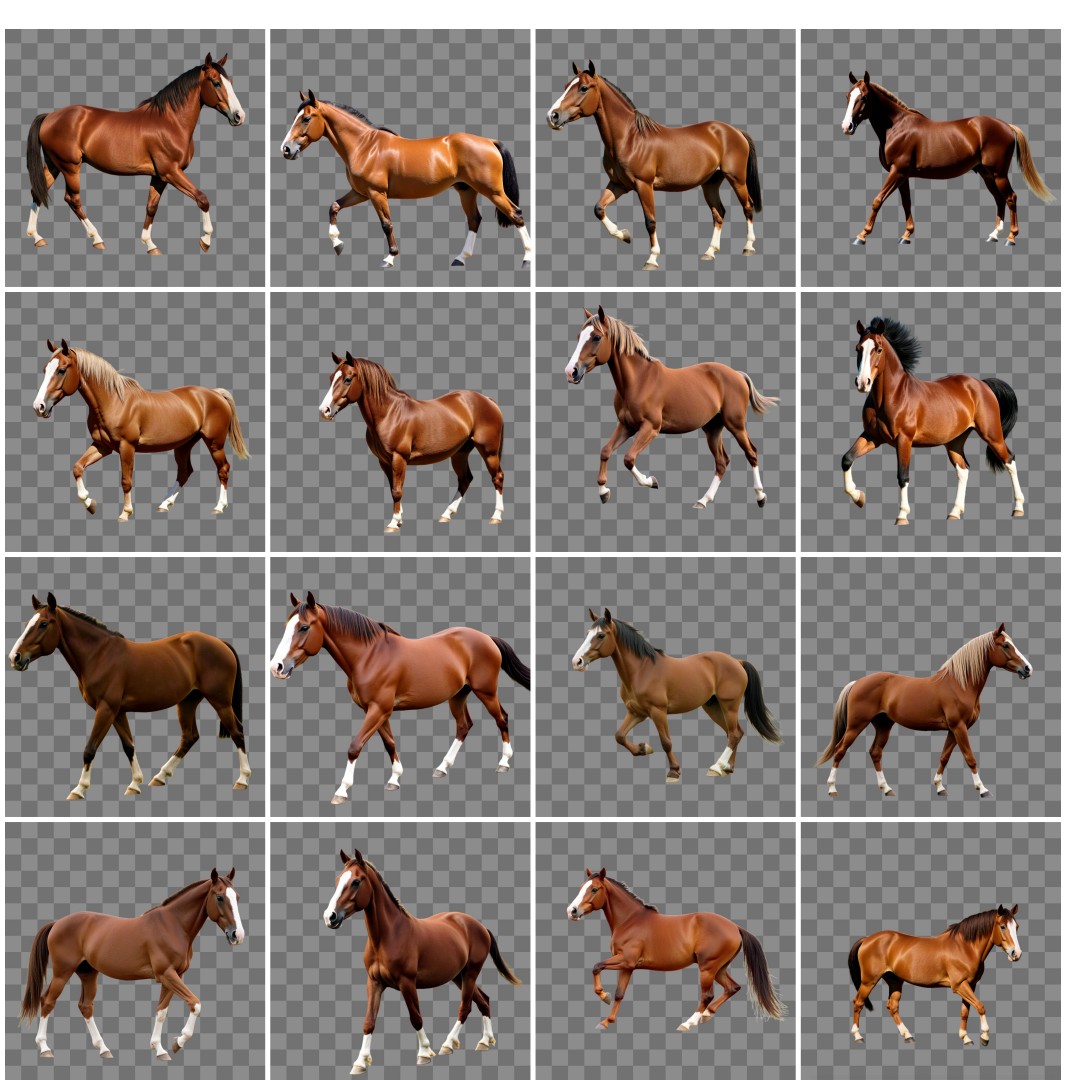

Figure 27: A brown horse with white markings on its face and legs is captured mid-stride.

1944
1945
1946
1947
1948
1949
1950
1951
1952
1953
1954
1955
1956
1957
1958
1959
1960
1961
1962
1963
1964
1965
1966
1967
1968
1969
1970
1971
1972
1973
1974
1975
1976
1977
1978
1979
1980
1981
1982
1983
1984
1985
1986
1987
1988
1989
1990
1991
1992
1993
1994
1995
1996
1997

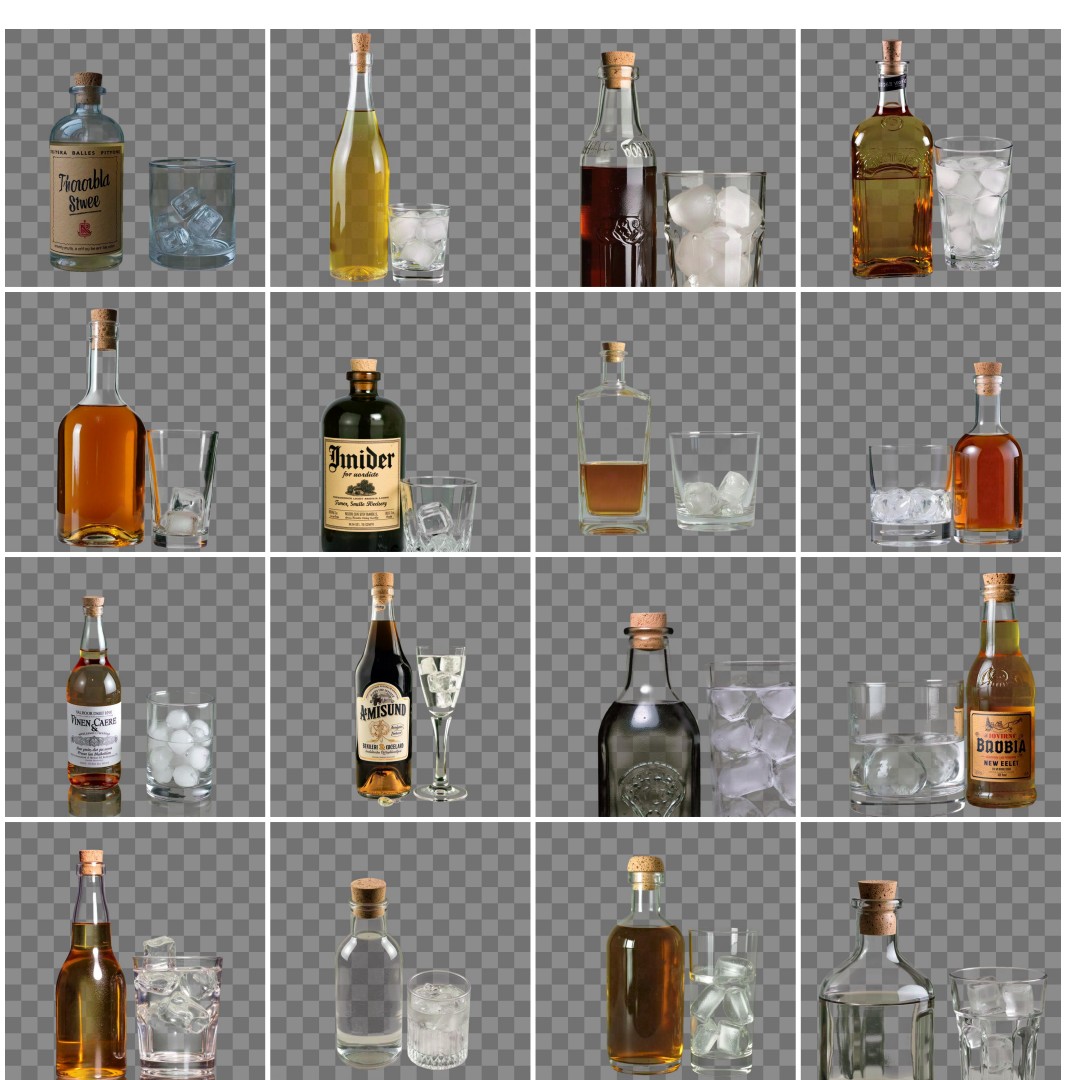

Figure 28: A bottle with a cork and a glass with ice cubes are placed together.

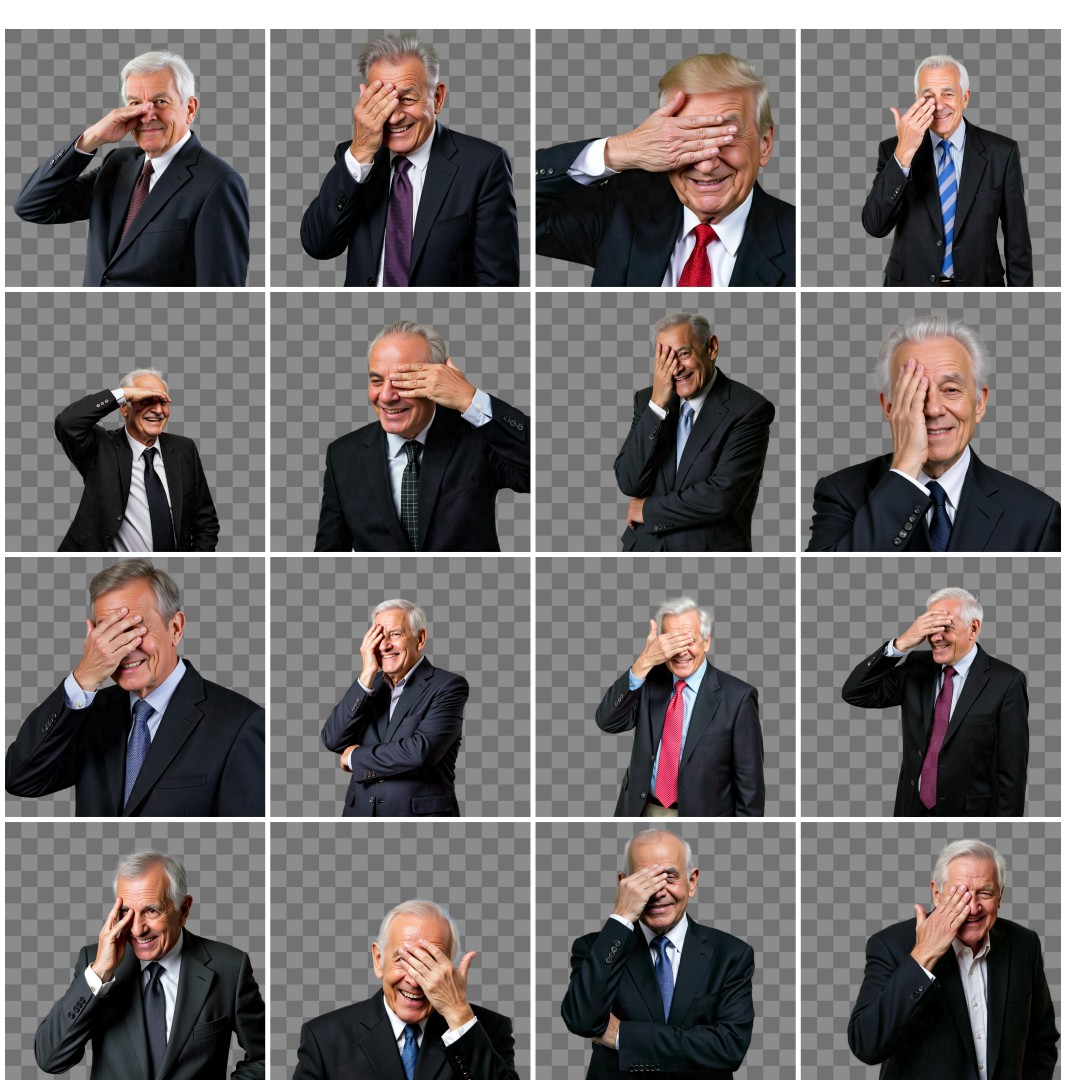

Figure 29: An elderly man in a suit covers his eyes with his hand, smiling.

Table 6: Metric scores of Alpha test split across color categories on VAE of LayerDiffuse.

| Metric | black | gray | white | red | yellow | green | cyan | blue | magenta | overall |
|---|---|---|---|---|---|---|---|---|---|---|
| PSNR | 32.2987 | 33.4605 | 32.2383 | 32.4273 | 32.3152 | 32.4010 | 32.3259 | 32.2476 | 32.3637 | 32.4531 |
| SSIM | 0.9424 | 0.9497 | 0.9453 | 0.9481 | 0.9464 | 0.9498 | 0.9486 | 0.9457 | 0.9496 | 0.9473 |
| LAION AES | 5.3193 | 5.2509 | 5.1818 | 5.0469 | 4.8619 | 4.9056 | 4.9785 | 4.9216 | 4.8089 | 5.0306 |
| rFID | 7.5630 | 5.6691 | 7.3526 | 6.3528 | 6.7486 | 6.2935 | 6.1922 | 6.1664 | 6.0109 | 6.4832 |
| LPIPS | 0.0368 | 0.0305 | 0.0355 | 0.0322 | 0.0330 | 0.0307 | 0.0314 | 0.0308 | 0.0308 | 0.0324 |

Table 7: Metric scores of Alpha test split across color categories on VAE of FLUX (ours).

| Metric | black | gray | white | red | yellow | green | cyan | blue | magenta | overall |
|---|---|---|---|---|---|---|---|---|---|---|
| PSNR | 38.1086 | 39.3765 | 37.8794 | 38.2636 | 37.9722 | 38.2767 | 38.0581 | 37.8197 | 38.0331 | 38.1987 |
| SSIM | 0.9763 | 0.9802 | 0.9784 | 0.9796 | 0.9792 | 0.9804 | 0.9801 | 0.9785 | 0.9803 | 0.9792 |
| LAION AES | 5.3554 | 5.2547 | 5.2045 | 5.0825 | 4.9082 | 4.9341 | 5.0104 | 4.9599 | 4.8444 | 5.0616 |
| rFID | 1.8807 | 1.9358 | 2.6166 | 1.3276 | 1.9337 | 1.2855 | 1.6091 | 1.2121 | 1.1391 | 1.6600 |
| LPIPS | 0.0187 | 0.0161 | 0.0181 | 0.0139 | 0.0139 | 0.0116 | 0.0128 | 0.0135 | 0.0122 | 0.0145 |

Table 8: Metric scores of Alpha test split across color categories on VAE of FLUX (w/o Norm KL).

| Metric | black | gray | white | red | yellow | green | cyan | blue | magenta | overall |
|---|---|---|---|---|---|---|---|---|---|---|
| PSNR | 37.3068 | 38.8858 | 36.9265 | 37.4930 | 37.1216 | 37.4400 | 37.1503 | 36.9786 | 37.1892 | 37.3880 |
| SSIM | 0.9732 | 0.9789 | 0.9761 | 0.9785 | 0.9772 | 0.9792 | 0.9787 | 0.9770 | 0.9792 | 0.9776 |
| LAION AES | 5.3538 | 5.2504 | 5.1904 | 5.0873 | 4.9319 | 4.9275 | 5.0210 | 4.9620 | 4.8512 | 5.0639 |
| rFID | 2.7818 | 2.8000 | 3.9974 | 1.9073 | 3.0712 | 1.9013 | 2.5008 | 1.7790 | 1.6696 | 2.4898 |
| LPIPS | 0.0216 | 0.0191 | 0.0217 | 0.0167 | 0.0169 | 0.0141 | 0.0155 | 0.0160 | 0.0146 | 0.0174 |

Table 9: Metric scores of Alpha test split across color categories on VAE of FLUX (w/o Ref KL).

| Metric | black | gray | white | red | yellow | green | cyan | blue | magenta | overall |
|---|---|---|---|---|---|---|---|---|---|---|
| PSNR | 38.0446 | 39.3336 | 37.8078 | 38.1920 | 37.8873 | 38.2212 | 38.0218 | 37.7860 | 37.9905 | 38.1428 |
| SSIM | 0.9758 | 0.9800 | 0.9781 | 0.9794 | 0.9789 | 0.9802 | 0.9799 | 0.9782 | 0.9801 | 0.9790 |
| LAION AES | 5.3596 | 5.2505 | 5.2010 | 5.0858 | 4.9156 | 4.9281 | 5.0128 | 4.9661 | 4.8519 | 5.0635 |
| rFID | 2.0329 | 2.0735 | 2.9343 | 1.3408 | 2.2037 | 1.3283 | 1.7552 | 1.3025 | 1.1790 | 1.7945 |
| LPIPS | 0.0199 | 0.0170 | 0.0191 | 0.0149 | 0.0148 | 0.0124 | 0.0135 | 0.0145 | 0.0130 | 0.0155 |

Table 10: Metric scores of Alpha test split across color categories on VAE of FLUX (w/o GAN).

| Metric | black | gray | white | red | yellow | green | cyan | blue | magenta | overall |
|---|---|---|---|---|---|---|---|---|---|---|
| PSNR | 38.3982 | 39.7655 | 38.1507 | 38.5581 | 38.2452 | 38.5394 | 38.3143 | 38.0783 | 38.3053 | 38.4839 |
| SSIM | 0.9773 | 0.9811 | 0.9793 | 0.9807 | 0.9801 | 0.9814 | 0.9811 | 0.9795 | 0.9813 | 0.9802 |
| LAION AES | 5.3435 | 5.2575 | 5.1959 | 5.0754 | 4.9003 | 4.9262 | 5.0056 | 4.9532 | 4.8428 | 5.0556 |
| rFID | 1.8812 | 2.1725 | 2.6998 | 1.2819 | 1.9951 | 1.3422 | 1.6618 | 1.1954 | 1.1517 | 1.7091 |
| LPIPS | 0.0195 | 0.0169 | 0.0193 | 0.0148 | 0.0150 | 0.0125 | 0.0138 | 0.0143 | 0.0129 | 0.0154 |

Table 11: Metric scores of Alpha test split across color categories on VAE of FLUX (w/o LPIPS).

| Metric | black | gray | white | red | yellow | green | cyan | blue | magenta | overall |
|---|---|---|---|---|---|---|---|---|---|---|
| PSNR | 36.5276 | 38.4143 | 36.0067 | 36.6541 | 36.2457 | 36.5500 | 36.2184 | 36.1114 | 36.2966 | 36.5583 |
| SSIM | 0.9710 | 0.9779 | 0.9747 | 0.9772 | 0.9761 | 0.9782 | 0.9777 | 0.9754 | 0.9781 | 0.9763 |
| LAION AES | 5.3500 | 5.2426 | 5.1774 | 5.0790 | 4.9108 | 4.9194 | 5.0064 | 4.9554 | 4.8430 | 5.0538 |
| rFID | 3.6630 | 3.1768 | 4.4897 | 2.6961 | 3.6317 | 2.6668 | 3.0846 | 2.4477 | 2.3424 | 3.1332 |
| LPIPS | 0.0236 | 0.0201 | 0.0231 | 0.0193 | 0.0194 | 0.0167 | 0.0182 | 0.0184 | 0.0172 | 0.0196 |

Table 12: Metric scores of Alpha test split across color categories on VAE of FLUX (w/o ABMSE).

| Metric | black | gray | white | red | yellow | green | cyan | blue | magenta | overall |
|---|---|---|---|---|---|---|---|---|---|---|
| PSNR | 38.0177 | 39.1793 | 37.7778 | 38.1497 | 37.8376 | 38.1371 | 37.9670 | 37.7438 | 37.9519 | 38.0847 |
| SSIM | 0.9758 | 0.9791 | 0.9775 | 0.9789 | 0.9783 | 0.9796 | 0.9793 | 0.9779 | 0.9795 | 0.9784 |
| LAION AES | 5.3491 | 5.2584 | 5.2022 | 5.0804 | 4.9057 | 4.9268 | 5.0043 | 4.9578 | 4.8455 | 5.0589 |
| rFID | 1.8418 | 2.0215 | 2.6620 | 1.2887 | 1.8918 | 1.3379 | 1.5880 | 1.1761 | 1.1315 | 1.6599 |
| LPIPS | 0.0189 | 0.0172 | 0.0188 | 0.0142 | 0.0145 | 0.0120 | 0.0131 | 0.0136 | 0.0124 | 0.0150 |

Table 13: Metric scores of Alpha test split across color categories on VAE of SDXL (ours).

| Metric | black | gray | white | red | yellow | green | cyan | blue | magenta | overall |
|---|---|---|---|---|---|---|---|---|---|---|
| PSNR | 35.5273 | 36.5843 | 35.1748 | 35.6110 | 35.3228 | 35.6430 | 35.4329 | 35.3176 | 35.4235 | 35.5597 |
| SSIM | 0.9560 | 0.9618 | 0.9591 | 0.9613 | 0.9602 | 0.9620 | 0.9617 | 0.9600 | 0.9620 | 0.9605 |
| LAION AES | 5.3394 | 5.2481 | 5.1973 | 5.0617 | 4.9121 | 4.9081 | 5.0042 | 4.9470 | 4.8323 | 5.0500 |
| rFID | 4.2172 | 4.0979 | 4.9661 | 3.2091 | 4.2911 | 3.2912 | 3.6115 | 3.0034 | 2.8873 | 3.7305 |
| LPIPS | 0.0464 | 0.0446 | 0.0469 | 0.0390 | 0.0396 | 0.0352 | 0.0374 | 0.0371 | 0.0355 | 0.0402 |

Table 14: Metric scores of Alpha test split across color categories on VAE of SDXL (w/o Norm KL).

| Metric | black | gray | white | red | yellow | green | cyan | blue | magenta | overall |
|---|---|---|---|---|---|---|---|---|---|---|
| PSNR | 35.4681 | 36.5361 | 35.1141 | 35.5533 | 35.2700 | 35.5798 | 35.3618 | 35.2497 | 35.3532 | 35.4985 |
| SSIM | 0.9564 | 0.9617 | 0.9591 | 0.9611 | 0.9602 | 0.9620 | 0.9619 | 0.9598 | 0.9619 | 0.9604 |
| LAION AES | 5.3420 | 5.2439 | 5.1978 | 5.0637 | 4.9034 | 4.9057 | 4.9980 | 4.9451 | 4.8281 | 5.0475 |
| rFID | 4.2193 | 4.3088 | 5.3154 | 3.3007 | 4.4657 | 3.3526 | 3.7425 | 3.1065 | 2.9439 | 3.8617 |
| LPIPS | 0.0463 | 0.0446 | 0.0466 | 0.0390 | 0.0395 | 0.0352 | 0.0374 | 0.0371 | 0.0355 | 0.0401 |

Table 15: Metric scores of Alpha test split across color categories on VAE of SDXL (w/o Ref KL).

| Metric | black | gray | white | red | yellow | green | cyan | blue | magenta | overall |
|---|---|---|---|---|---|---|---|---|---|---|
| PSNR | 35.4490 | 36.4958 | 35.1270 | 35.5450 | 35.2833 | 35.5766 | 35.3628 | 35.2377 | 35.3482 | 35.4917 |
| SSIM | 0.9557 | 0.9615 | 0.9590 | 0.9610 | 0.9602 | 0.9618 | 0.9615 | 0.9596 | 0.9617 | 0.9602 |
| LAION AES | 5.3399 | 5.2404 | 5.1905 | 5.0652 | 4.9079 | 4.9075 | 5.0016 | 4.9487 | 4.8343 | 5.0484 |
| rFID | 4.1823 | 4.3080 | 5.1787 | 3.2270 | 4.2665 | 3.2916 | 3.6771 | 3.0593 | 2.9209 | 3.7902 |
| LPIPS | 0.0452 | 0.0439 | 0.0456 | 0.0381 | 0.0386 | 0.0344 | 0.0366 | 0.0362 | 0.0348 | 0.0393 |

Table 16: Metric scores of Alpha test split across color categories on VAE of SDXL (w/o GAN).

| Metric | black | gray | white | red | yellow | green | cyan | blue | magenta | overall |
|---|---|---|---|---|---|---|---|---|---|---|
| PSNR | 35.6458 | 36.7020 | 35.3358 | 35.7219 | 35.4739 | 35.7754 | 35.5738 | 35.4336 | 35.5357 | 35.6887 |
| SSIM | 0.9571 | 0.9622 | 0.9598 | 0.9617 | 0.9609 | 0.9625 | 0.9622 | 0.9604 | 0.9624 | 0.9610 |
| LAION AES | 5.3344 | 5.2463 | 5.1910 | 5.0589 | 4.8991 | 4.9111 | 5.0043 | 4.9456 | 4.8293 | 5.0467 |
| rFID | 4.0130 | 4.3954 | 5.0296 | 3.1645 | 4.0869 | 3.2718 | 3.4708 | 2.9374 | 2.8250 | 3.6883 |
| LPIPS | 0.0468 | 0.0448 | 0.0468 | 0.0394 | 0.0398 | 0.0355 | 0.0377 | 0.0374 | 0.0359 | 0.0405 |

Table 17: Metric scores of Alpha test split across color categories on VAE of SDXL (w/o LPIPS).

| Metric | black | gray | white | red | yellow | green | cyan | blue | magenta | overall |
|---|---|---|---|---|---|---|---|---|---|---|
| PSNR | 35.4575 | 36.5327 | 35.1170 | 35.6070 | 35.2386 | 35.4740 | 35.3096 | 35.2734 | 35.4419 | 35.4946 |
| SSIM | 0.9561 | 0.9616 | 0.9591 | 0.9612 | 0.9602 | 0.9619 | 0.9615 | 0.9598 | 0.9619 | 0.9604 |
| LAION AES | 5.3446 | 5.2464 | 5.1980 | 5.0667 | 4.9047 | 4.9006 | 4.9996 | 4.9498 | 4.8342 | 5.0494 |
| rFID | 4.1360 | 4.2773 | 5.1766 | 3.1705 | 4.2642 | 3.4594 | 3.6515 | 2.9238 | 2.8485 | 3.7675 |
| LPIPS | 0.0455 | 0.0441 | 0.0461 | 0.0382 | 0.0392 | 0.0350 | 0.0372 | 0.0364 | 0.0348 | 0.0396 |

Table 18: Metric scores of Alpha test split across color categories on VAE of SDXL (w/o ABMSE).

| Metric | black | gray | white | red | yellow | green | cyan | blue | magenta | overall |
|---|---|---|---|---|---|---|---|---|---|---|
| PSNR | 35.3931 | 36.1835 | 35.0748 | 35.4523 | 35.1950 | 35.4707 | 35.3176 | 35.2164 | 35.2939 | 35.3997 |
| SSIM | 0.9562 | 0.9594 | 0.9566 | 0.9596 | 0.9581 | 0.9600 | 0.9596 | 0.9585 | 0.9602 | 0.9587 |
| LAION AES | 5.3486 | 5.2545 | 5.1987 | 5.0779 | 4.9230 | 4.9194 | 5.0173 | 4.9556 | 4.8473 | 5.0603 |
| rFID | 3.8295 | 4.4907 | 5.2114 | 3.1326 | 4.2212 | 3.3066 | 3.6059 | 2.8809 | 2.8459 | 3.7250 |
| LPIPS | 0.0442 | 0.0458 | 0.0477 | 0.0376 | 0.0388 | 0.0339 | 0.0361 | 0.0355 | 0.0344 | 0.0393 |

Table 19: Metric scores across subtypes of AIM and color categories on VAE of LayerDiffuse.

| Subtype | Metric | Colors | | | | | | | | | |
|---|---|---|---|---|---|---|---|---|---|---|---|
| | | black | gray | white | red | yellow | green | cyan | blue | magenta | overall |
| portrait | PSNR | 33.0302 | 35.1074 | 32.9291 | 33.3363 | 33.1247 | 33.1945 | 33.0419 | 32.9910 | 33.1967 | 33.3280 |
| | SSIM | 0.9484 | 0.9582 | 0.9535 | 0.9563 | 0.9544 | 0.9581 | 0.9567 | 0.9532 | 0.9579 | 0.9552 |
| | LAION AES | 5.0925 | 5.0978 | 5.0091 | 5.0252 | 4.9609 | 4.9370 | 4.9686 | 4.8550 | 4.8635 | 4.9609 |
| | rFID | 22.1295 | 16.8871 | 17.0631 | 15.9141 | 17.0627 | 13.7088 | 14.4707 | 15.4059 | 14.2083 | 16.3167 |
| | LPIPS | 0.0431 | 0.0348 | 0.0412 | 0.0374 | 0.0382 | 0.0345 | 0.0356 | 0.0351 | 0.0348 | 0.0372 |
| transparent | PSNR | 28.7389 | 37.2007 | 28.2495 | 29.2511 | 28.7266 | 29.0668 | 28.7697 | 28.7344 | 28.9971 | 29.7483 |
| | SSIM | 0.8729 | 0.9699 | 0.9346 | 0.9584 | 0.9440 | 0.9686 | 0.9604 | 0.9365 | 0.9682 | 0.9459 |
| | LAION AES | 5.1693 | 4.8399 | 4.9708 | 4.8342 | 4.6140 | 4.6275 | 4.7342 | 4.6617 | 4.4989 | 4.7723 |
| | rFID | 53.3914 | 31.0126 | 63.7540 | 44.0127 | 58.8518 | 49.7999 | 55.5840 | 41.7431 | 43.2281 | 49.0420 |
| | LPIPS | 0.1002 | 0.0538 | 0.0948 | 0.0752 | 0.0819 | 0.0523 | 0.0709 | 0.0631 | 0.0523 | 0.0716 |
| toy | PSNR | 32.2552 | 34.9006 | 32.2383 | 32.5595 | 32.4096 | 32.4781 | 32.3491 | 32.2201 | 32.4445 | 32.6506 |
| | SSIM | 0.9426 | 0.9561 | 0.9487 | 0.9537 | 0.9501 | 0.9558 | 0.9537 | 0.9498 | 0.9558 | 0.9518 |
| | LAION AES | 5.0666 | 5.0380 | 4.9279 | 4.7793 | 4.7135 | 4.6888 | 4.8381 | 4.7222 | 4.6490 | 4.8248 |
| | rFID | 15.1318 | 10.4505 | 13.3246 | 12.4294 | 12.7679 | 11.8679 | 12.3385 | 11.8714 | 12.1212 | 12.4781 |
| | LPIPS | 0.0412 | 0.0317 | 0.0413 | 0.0355 | 0.0366 | 0.0316 | 0.0332 | 0.0333 | 0.0319 | 0.0351 |
| furniture | PSNR | 32.6766 | 35.7837 | 32.8652 | 33.2892 | 33.0543 | 33.1803 | 33.0694 | 32.7468 | 33.2000 | 33.3184 |
| | SSIM | 0.9471 | 0.9562 | 0.9540 | 0.9542 | 0.9542 | 0.9564 | 0.9552 | 0.9519 | 0.9562 | 0.9539 |
| | LAION AES | 5.2373 | 5.1513 | 5.0425 | 4.9424 | 4.7517 | 4.8308 | 4.8190 | 4.7604 | 4.7124 | 4.9164 |
| | rFID | 13.7183 | 8.8866 | 8.2809 | 8.9743 | 8.9639 | 7.7821 | 8.5635 | 9.5422 | 8.4230 | 9.2372 |
| | LPIPS | 0.0317 | 0.0238 | 0.0277 | 0.0231 | 0.0233 | 0.0198 | 0.0204 | 0.0215 | 0.0205 | 0.0235 |
| plant | PSNR | 31.5083 | 34.6841 | 31.4140 | 31.9633 | 31.7109 | 31.9028 | 31.6495 | 31.4401 | 31.7531 | 32.0029 |
| | SSIM | 0.9260 | 0.9548 | 0.9430 | 0.9491 | 0.9462 | 0.9551 | 0.9524 | 0.9416 | 0.9535 | 0.9469 |
| | LAION AES | 5.3481 | 5.1512 | 5.1763 | 4.9990 | 4.8413 | 4.8468 | 4.9416 | 4.8548 | 4.8394 | 4.9998 |
| | rFID | 24.9541 | 15.9974 | 20.2765 | 21.2488 | 22.8327 | 19.7409 | 18.2817 | 19.2567 | 19.2368 | 20.2028 |
| | LPIPS | 0.0497 | 0.0331 | 0.0493 | 0.0407 | 0.0417 | 0.0332 | 0.0365 | 0.0373 | 0.0350 | 0.0396 |
| fruit | PSNR | 33.0276 | 35.4943 | 33.0492 | 33.3731 | 33.2545 | 33.2247 | 33.0046 | 32.8570 | 33.1379 | 33.3803 |
| | SSIM | 0.9643 | 0.9700 | 0.9678 | 0.9687 | 0.9680 | 0.9702 | 0.9692 | 0.9671 | 0.9698 | 0.9683 |
| | LAION AES | 5.2405 | 5.0279 | 5.0647 | 4.8337 | 4.7842 | 4.7140 | 4.8225 | 4.7354 | 4.6256 | 4.8721 |
| | rFID | 9.5676 | 7.6870 | 8.1118 | 11.4766 | 11.2327 | 9.2085 | 9.3322 | 8.7917 | 10.2548 | 9.5181 |
| | LPIPS | 0.0323 | 0.0288 | 0.0314 | 0.0294 | 0.0283 | 0.0256 | 0.0262 | 0.0264 | 0.0259 | 0.0283 |
| animal | PSNR | 29.9683 | 31.2897 | 29.9186 | 30.1610 | 30.0505 | 30.1545 | 30.0615 | 29.9790 | 30.1008 | 30.1871 |
| | SSIM | 0.8764 | 0.8863 | 0.8804 | 0.8844 | 0.8816 | 0.8862 | 0.8846 | 0.8816 | 0.8861 | 0.8831 |
| | LAION AES | 5.4407 | 5.3790 | 5.2565 | 5.1032 | 4.9143 | 4.9353 | 5.0284 | 4.9831 | 4.8549 | 5.0995 |
| | rFID | 8.6218 | 6.5729 | 7.6093 | 6.9553 | 7.8976 | 6.5928 | 6.4681 | 6.6253 | 6.7486 | 7.1213 |
| | LPIPS | 0.0665 | 0.0570 | 0.0658 | 0.0573 | 0.0584 | 0.0518 | 0.0547 | 0.0547 | 0.0524 | 0.0576 |
| overall | PSNR | 31.6007 | 34.9229 | 31.5234 | 31.9905 | 31.7616 | 31.8860 | 31.7065 | 31.5669 | 31.8329 | 32.0879 |
| | SSIM | 0.9254 | 0.9502 | 0.9403 | 0.9464 | 0.9426 | 0.9501 | 0.9475 | 0.9402 | 0.9496 | 0.9436 |
| | LAION AES | 5.2279 | 5.0979 | 5.0640 | 4.9310 | 4.7751 | 4.8006 | 4.8744 | 4.7961 | 4.7205 | 4.9208 |
| | rFID | 21.0735 | 13.9277 | 19.7743 | 17.2873 | 19.9442 | 16.9573 | 17.8627 | 16.1766 | 16.3173 | 17.7023 |
| | LPIPS | 0.0521 | 0.0376 | 0.0502 | 0.0425 | 0.0442 | 0.0355 | 0.0396 | 0.0388 | 0.0361 | 0.0418 |

Table 20: Metric scores across subtypes of AIM and color categories on VAE of FLUX (ours).

| Subtype | Metric | Colors | | | | | | | | | |
|---|---|---|---|---|---|---|---|---|---|---|---|
| | | black | gray | white | red | yellow | green | cyan | blue | magenta | overall |
| portrait | PSNR | 40.2862 | 42.8302 | 40.0708 | 40.5722 | 40.2461 | 40.4436 | 40.0881 | 39.7575 | 40.1068 | 40.4891 |
| | SSIM | 0.9846 | 0.9891 | 0.9876 | 0.9883 | 0.9881 | 0.9893 | 0.9890 | 0.9868 | 0.9890 | 0.9880 |
| | LAION AES | 5.1643 | 5.1615 | 5.0640 | 5.1189 | 4.9139 | 5.0389 | 5.0244 | 4.9664 | 4.9457 | 5.0442 |
| | rFID | 4.7230 | 3.1646 | 3.5066 | 2.4490 | 3.1405 | 2.0564 | 2.3312 | 2.7373 | 2.0353 | 2.9049 |
| | LPIPS | 0.0161 | 0.0119 | 0.0139 | 0.0111 | 0.0107 | 0.0087 | 0.0097 | 0.0105 | 0.0092 | 0.0113 |
| transparent | PSNR | 32.7464 | 35.9282 | 30.5379 | 31.9773 | 31.1065 | 31.9161 | 30.9990 | 31.0960 | 30.9451 | 31.9169 |
| | SSIM | 0.8964 | 0.9495 | 0.9114 | 0.9569 | 0.9231 | 0.9488 | 0.9405 | 0.9475 | 0.9565 | 0.9367 |
| | LAION AES | 5.1757 | 4.7798 | 4.9589 | 4.9120 | 4.6946 | 4.6770 | 4.8460 | 4.7376 | 4.5938 | 4.8195 |
| | rFID | 62.4205 | 63.1103 | 63.8062 | 51.5562 | 53.9683 | 48.4641 | 42.7990 | 48.7593 | 47.4126 | 53.5885 |
| | LPIPS | 0.1062 | 0.1131 | 0.1279 | 0.0933 | 0.0975 | 0.0572 | 0.0923 | 0.0619 | 0.0606 | 0.0900 |
| toy | PSNR | 38.6346 | 41.8514 | 39.1643 | 39.1149 | 39.0971 | 39.1770 | 39.0296 | 38.3729 | 38.8742 | 39.2573 |
| | SSIM | 0.9805 | 0.9875 | 0.9845 | 0.9869 | 0.9856 | 0.9878 | 0.9871 | 0.9849 | 0.9878 | 0.9858 |
| | LAION AES | 5.0728 | 5.0132 | 4.9063 | 4.7833 | 4.7323 | 4.6776 | 4.8323 | 4.7227 | 4.6574 | 4.8220 |
| | rFID | 5.0960 | 3.2402 | 4.9327 | 2.6098 | 3.3534 | 2.2915 | 3.1292 | 2.9656 | 2.2567 | 3.3195 |
| | LPIPS | 0.0194 | 0.0145 | 0.0175 | 0.0127 | 0.0125 | 0.0086 | 0.0105 | 0.0115 | 0.0093 | 0.0129 |
| furniture | PSNR | 36.0337 | 39.3700 | 35.8151 | 36.7891 | 36.2390 | 36.5989 | 36.1344 | 35.6588 | 36.3051 | 36.5493 |
| | SSIM | 0.9688 | 0.9772 | 0.9764 | 0.9766 | 0.9772 | 0.9785 | 0.9784 | 0.9744 | 0.9778 | 0.9761 |
| | LAION AES | 5.2181 | 5.1342 | 5.0837 | 4.9339 | 4.7930 | 4.8431 | 4.8709 | 4.7638 | 4.7292 | 4.9300 |
| | rFID | 9.0251 | 6.2099 | 5.1666 | 4.3327 | 4.2806 | 2.9276 | 3.8278 | 4.8363 | 3.4676 | 4.8971 |
| | LPIPS | 0.0274 | 0.0217 | 0.0229 | 0.0163 | 0.0160 | 0.0134 | 0.0146 | 0.0119 | 0.0119 | 0.0173 |
| plant | PSNR | 35.4190 | 39.4948 | 35.0313 | 35.9945 | 35.5857 | 35.9707 | 35.3247 | 34.8846 | 35.3331 | 35.8932 |
| | SSIM | 0.9552 | 0.9810 | 0.9709 | 0.9769 | 0.9751 | 0.9822 | 0.9807 | 0.9689 | 0.9805 | 0.9746 |
| | LAION AES | 5.3597 | 5.1651 | 5.1537 | 5.0302 | 4.9190 | 4.8932 | 4.9896 | 4.8907 | 4.8750 | 5.0307 |
| | rFID | 23.5336 | 13.1029 | 15.4566 | 12.1525 | 12.6983 | 9.9689 | 8.9950 | 10.9350 | 8.9842 | 12.8697 |
| | LPIPS | 0.0456 | 0.0307 | 0.0398 | 0.0315 | 0.0282 | 0.0177 | 0.0236 | 0.0278 | 0.0222 | 0.0297 |
| fruit | PSNR | 37.1331 | 41.2947 | 37.4723 | 38.3896 | 38.3911 | 38.0195 | 37.0902 | 36.5205 | 37.4724 | 37.9759 |
| | SSIM | 0.9831 | 0.9893 | 0.9878 | 0.9884 | 0.9885 | 0.9900 | 0.9896 | 0.9865 | 0.9895 | 0.9881 |
| | LAION AES | 5.2458 | 5.0305 | 5.0561 | 4.8632 | 4.8268 | 4.7323 | 4.8490 | 4.7066 | 4.6479 | 4.8842 |
| | rFID | 3.6791 | 3.7793 | 4.1170 | 3.4607 | 3.2702 | 2.9133 | 3.4767 | 3.2026 | 3.3178 | 3.4685 |
| | LPIPS | 0.0164 | 0.0130 | 0.0141 | 0.0117 | 0.0112 | 0.0089 | 0.0107 | 0.0114 | 0.0094 | 0.0119 |
| animal | PSNR | 36.3231 | 37.7808 | 36.1940 | 36.5966 | 36.3776 | 36.6106 | 36.3719 | 36.1123 | 36.3624 | 36.5255 |
| | SSIM | 0.9641 | 0.9677 | 0.9652 | 0.9673 | 0.9659 | 0.9678 | 0.9671 | 0.9660 | 0.9679 | 0.9666 |
| | LAION AES | 5.4932 | 5.3990 | 5.2695 | 5.1383 | 4.9549 | 4.9503 | 5.0560 | 5.0122 | 4.8685 | 5.1269 |
| | rFID | 1.8697 | 1.7021 | 2.0694 | 1.0978 | 1.7038 | 1.1534 | 1.3980 | 1.1085 | 1.1317 | 1.4705 |
| | LPIPS | 0.0310 | 0.0271 | 0.0314 | 0.0242 | 0.0249 | 0.0211 | 0.0230 | 0.0233 | 0.0214 | 0.0253 |
| overall | PSNR | 36.6537 | 39.7929 | 36.3265 | 37.0620 | 36.7204 | 36.9623 | 36.4340 | 36.0575 | 36.4856 | 36.9439 |
| | SSIM | 0.9618 | 0.9773 | 0.9691 | 0.9773 | 0.9719 | 0.9778 | 0.9761 | 0.9736 | 0.9784 | 0.9737 |
| | LAION AES | 5.2471 | 5.0976 | 5.0703 | 4.9685 | 4.8335 | 4.8303 | 4.9240 | 4.8286 | 4.7596 | 4.9511 |
| | rFID | 15.7639 | 13.4728 | 14.1507 | 11.0941 | 11.7736 | 9.9679 | 9.4224 | 10.6492 | 9.8008 | 11.7884 |
| | LPIPS | 0.0374 | 0.0331 | 0.0382 | 0.0287 | 0.0287 | 0.0191 | 0.0262 | 0.0230 | 0.0206 | 0.0283 |

Table 21: Metric scores across subtypes of AIM and color categories on VAE of FLUX (w/o Norm KL).

| Subtype | Metric | Colors | | | | | | | | | |
|---|---|---|---|---|---|---|---|---|---|---|---|
| | | black | gray | white | red | yellow | green | cyan | blue | magenta | overall |
| portrait | PSNR | 38.5509 | 41.8724 | 38.2719 | 39.0076 | 38.6528 | 38.7745 | 38.3025 | 38.0592 | 38.4824 | 38.8860 |
| | SSIM | 0.9818 | 0.9876 | 0.9849 | 0.9869 | 0.9859 | 0.9880 | 0.9874 | 0.9850 | 0.9878 | 0.9861 |
| | LAION AES | 5.1598 | 5.1631 | 5.0611 | 5.1102 | 4.9317 | 5.0489 | 5.0314 | 4.9646 | 4.9550 | 5.0473 |
| | rFID | 6.2641 | 6.0452 | 6.3839 | 3.5284 | 5.4736 | 3.2585 | 4.1544 | 4.2184 | 3.1191 | 4.7162 |
| | LPIPS | 0.0191 | 0.0152 | 0.0177 | 0.0135 | 0.0135 | 0.0108 | 0.0123 | 0.0127 | 0.0112 | 0.0140 |
| transparent | PSNR | 29.7288 | 33.5282 | 27.7907 | 29.1354 | 28.3555 | 28.8551 | 28.0053 | 28.1877 | 28.0979 | 29.0761 |
| | SSIM | 0.8749 | 0.9393 | 0.8923 | 0.9472 | 0.9071 | 0.9378 | 0.9281 | 0.9342 | 0.9471 | 0.9231 |
| | LAION AES | 5.1523 | 4.8208 | 4.9559 | 4.9720 | 4.7268 | 4.7199 | 4.8436 | 4.7613 | 4.6108 | 4.8404 |
| | rFID | 73.1509 | 80.6974 | 78.5752 | 67.6762 | 65.4955 | 67.5740 | 60.6353 | 64.3469 | 64.9532 | 69.2338 |
| | LPIPS | 0.1240 | 0.1294 | 0.1463 | 0.1169 | 0.1183 | 0.0769 | 0.1149 | 0.0847 | 0.0800 | 0.1102 |
| toy | PSNR | 36.4243 | 40.6189 | 37.1455 | 37.1201 | 37.1562 | 37.0230 | 36.8876 | 36.2758 | 36.8984 | 37.2833 |
| | SSIM | 0.9758 | 0.9854 | 0.9807 | 0.9847 | 0.9825 | 0.9858 | 0.9848 | 0.9819 | 0.9859 | 0.9831 |
| | LAION AES | 5.0591 | 4.9747 | 4.8811 | 4.7791 | 4.7609 | 4.6959 | 4.8444 | 4.7240 | 4.6478 | 4.8186 |
| | rFID | 7.2048 | 5.1722 | 7.4363 | 4.4601 | 5.6144 | 4.4943 | 5.2599 | 4.5691 | 3.8543 | 5.3406 |
| | LPIPS | 0.0244 | 0.0191 | 0.0228 | 0.0171 | 0.0165 | 0.0118 | 0.0144 | 0.0154 | 0.0126 | 0.0171 |
| furniture | PSNR | 34.1329 | 38.0555 | 34.0710 | 34.9505 | 34.5558 | 34.7404 | 34.2669 | 33.8363 | 34.5155 | 34.7916 |
| | SSIM | 0.9630 | 0.9739 | 0.9714 | 0.9734 | 0.9728 | 0.9752 | 0.9748 | 0.9703 | 0.9747 | 0.9722 |
| | LAION AES | 5.2046 | 5.1255 | 5.0855 | 4.9401 | 4.8407 | 4.8606 | 4.9150 | 4.7701 | 4.7370 | 4.9421 |
| | rFID | 13.5650 | 9.4165 | 8.0235 | 6.2938 | 7.0481 | 4.8476 | 6.0175 | 7.5274 | 5.7671 | 7.6118 |
| | LPIPS | 0.0352 | 0.0273 | 0.0293 | 0.0222 | 0.0213 | 0.0155 | 0.0185 | 0.0205 | 0.0168 | 0.0230 |
| plant | PSNR | 33.1319 | 38.0106 | 32.7570 | 33.7800 | 33.4167 | 33.6463 | 32.9825 | 32.6424 | 33.1046 | 33.7191 |
| | SSIM | 0.9439 | 0.9770 | 0.9612 | 0.9719 | 0.9674 | 0.9782 | 0.9757 | 0.9614 | 0.9765 | 0.9681 |
| | LAION AES | 5.3367 | 5.1472 | 5.1222 | 5.0281 | 4.9179 | 4.8927 | 4.9802 | 4.8983 | 4.8828 | 5.0229 |
| | rFID | 31.9773 | 19.0634 | 23.8921 | 19.4423 | 20.7304 | 17.1835 | 15.9800 | 17.1081 | 15.3490 | 20.0807 |
| | LPIPS | 0.0560 | 0.0402 | 0.0527 | 0.0449 | 0.0412 | 0.0281 | 0.0355 | 0.0392 | 0.0334 | 0.0412 |
| fruit | PSNR | 36.1185 | 40.4148 | 36.1384 | 37.3119 | 37.1387 | 36.6897 | 35.7235 | 35.4306 | 36.2971 | 36.8070 |
| | SSIM | 0.9821 | 0.9885 | 0.9852 | 0.9880 | 0.9863 | 0.9889 | 0.9881 | 0.9859 | 0.9890 | 0.9869 |
| | LAION AES | 5.2366 | 4.9962 | 5.0339 | 4.8680 | 4.8818 | 4.7466 | 4.8758 | 4.7061 | 4.6696 | 4.8905 |
| | rFID | 4.8869 | 5.4323 | 6.9355 | 5.1886 | 5.9763 | 4.8071 | 4.7394 | 4.7370 | 5.2208 | 5.3717 |
| | LPIPS | 0.0197 | 0.0153 | 0.0180 | 0.0145 | 0.0140 | 0.0113 | 0.0137 | 0.0141 | 0.0118 | 0.0147 |
| animal | PSNR | 35.2974 | 37.4096 | 35.0582 | 35.6776 | 35.4220 | 35.6363 | 35.2772 | 35.0626 | 35.3525 | 35.5770 |
| | SSIM | 0.9613 | 0.9658 | 0.9618 | 0.9656 | 0.9629 | 0.9659 | 0.9649 | 0.9639 | 0.9664 | 0.9643 |
| | LAION AES | 5.4894 | 5.3688 | 5.2273 | 5.1314 | 4.9542 | 4.9444 | 5.0530 | 5.0023 | 4.8640 | 5.1150 |
| | rFID | 2.3897 | 2.5880 | 3.2575 | 1.5369 | 2.6013 | 1.6721 | 2.1457 | 1.5764 | 1.6024 | 2.1522 |
| | LPIPS | 0.0370 | 0.0323 | 0.0371 | 0.0292 | 0.0297 | 0.0254 | 0.0278 | 0.0283 | 0.0258 | 0.0303 |
| overall | PSNR | 34.7692 | 38.5586 | 34.4618 | 35.2833 | 34.9568 | 35.0522 | 34.4922 | 34.2135 | 34.6783 | 35.1629 |
| | SSIM | 0.9547 | 0.9739 | 0.9625 | 0.9740 | 0.9664 | 0.9743 | 0.9720 | 0.9689 | 0.9753 | 0.9691 |
| | LAION AES | 5.2341 | 5.0852 | 5.0524 | 4.9756 | 4.8591 | 4.8441 | 4.9348 | 4.8324 | 4.7667 | 4.9538 |
| | rFID | 19.9198 | 18.3450 | 19.2149 | 15.4466 | 16.1342 | 14.8339 | 14.1332 | 14.9292 | 14.2666 | 16.3582 |
| | LPIPS | 0.0451 | 0.0398 | 0.0463 | 0.0369 | 0.0364 | 0.0257 | 0.0339 | 0.0307 | 0.0274 | 0.0358 |

Table 22: Metric scores across subtypes of AIM and color categories on VAE of FLUX (w/o Ref KL).

| Subtype | Metric | Colors | | | | | | | | | |
|---|---|---|---|---|---|---|---|---|---|---|---|
| | | black | gray | white | red | yellow | green | cyan | blue | magenta | overall |
| portrait | PSNR | 40.1312 | 42.6648 | 39.9200 | 40.3855 | 40.0629 | 40.3297 | 40.0098 | 39.6624 | 39.9857 | 40.3502 |
| | SSIM | 0.9840 | 0.9887 | 0.9870 | 0.9878 | 0.9876 | 0.9889 | 0.9885 | 0.9863 | 0.9886 | 0.9875 |
| | LAION AES | 5.1771 | 5.1678 | 5.0666 | 5.1155 | 4.9246 | 5.0426 | 5.0274 | 4.9635 | 4.9558 | 5.0490 |
| | rFID | 4.8658 | 3.7301 | 4.1279 | 2.7334 | 3.6530 | 2.2121 | 2.6575 | 2.8246 | 2.0897 | 3.2105 |
| | LPIPS | 0.0178 | 0.0133 | 0.0153 | 0.0122 | 0.0117 | 0.0095 | 0.0106 | 0.0116 | 0.0102 | 0.0125 |
| transparent | PSNR | 32.7316 | 36.2265 | 30.8402 | 32.1378 | 31.3515 | 32.2155 | 31.4091 | 31.3970 | 31.2557 | 32.1739 |
| | SSIM | 0.8981 | 0.9534 | 0.9147 | 0.9586 | 0.9272 | 0.9531 | 0.9452 | 0.9471 | 0.9596 | 0.9397 |
| | LAION AES | 5.1688 | 4.7869 | 4.9707 | 4.9361 | 4.6774 | 4.6457 | 4.8249 | 4.7332 | 4.5499 | 4.8104 |
| | rFID | 62.1280 | 61.5266 | 63.9293 | 50.4211 | 55.7045 | 48.9617 | 45.0925 | 49.0477 | 46.6854 | 53.7219 |
| | LPIPS | 0.1061 | 0.1107 | 0.1272 | 0.0896 | 0.0947 | 0.0508 | 0.0880 | 0.0582 | 0.0550 | 0.0867 |
| toy | PSNR | 38.5473 | 41.7206 | 39.0384 | 38.9778 | 38.9604 | 39.1150 | 39.0092 | 38.3308 | 38.7905 | 39.1656 |
| | SSIM | 0.9799 | 0.9872 | 0.9841 | 0.9866 | 0.9852 | 0.9875 | 0.9868 | 0.9845 | 0.9875 | 0.9855 |
| | LAION AES | 5.0765 | 5.0098 | 4.9065 | 4.7999 | 4.7441 | 4.6944 | 4.8390 | 4.7310 | 4.6629 | 4.8293 |
| | rFID | 5.4075 | 3.4763 | 4.9800 | 2.8202 | 3.5273 | 2.2839 | 3.1280 | 3.1538 | 2.2533 | 3.4478 |
| | LPIPS | 0.0205 | 0.0155 | 0.0185 | 0.0134 | 0.0129 | 0.0089 | 0.0107 | 0.0121 | 0.0097 | 0.0136 |
| furniture | PSNR | 35.8920 | 39.2213 | 35.6958 | 36.6068 | 36.0804 | 36.4861 | 36.0860 | 35.5971 | 36.2047 | 36.4300 |
| | SSIM | 0.9684 | 0.9771 | 0.9761 | 0.9764 | 0.9769 | 0.9784 | 0.9782 | 0.9741 | 0.9776 | 0.9759 |
| | LAION AES | 5.2283 | 5.1389 | 5.1144 | 4.9328 | 4.8133 | 4.8386 | 4.8877 | 4.7623 | 4.7417 | 4.9398 |
| | rFID | 9.2393 | 7.4296 | 5.5155 | 4.5326 | 4.3792 | 3.0944 | 3.6757 | 5.1087 | 3.7082 | 5.1870 |
| | LPIPS | 0.0293 | 0.0226 | 0.0241 | 0.0172 | 0.0166 | 0.0138 | 0.0155 | 0.0126 | 0.0126 | 0.0182 |
| plant | PSNR | 35.2696 | 39.3673 | 34.8934 | 35.7848 | 35.3838 | 35.8781 | 35.3170 | 34.8419 | 35.2290 | 35.7739 |
| | SSIM | 0.9530 | 0.9806 | 0.9693 | 0.9764 | 0.9739 | 0.9817 | 0.9800 | 0.9680 | 0.9801 | 0.9737 |
| | LAION AES | 5.3542 | 5.1534 | 5.1637 | 5.0315 | 4.9233 | 4.8729 | 4.9753 | 4.8999 | 4.8713 | 5.0273 |
| | rFID | 25.7136 | 14.6049 | 16.4557 | 13.3439 | 13.5467 | 10.4892 | 9.3926 | 11.6004 | 9.7352 | 13.8758 |
| | LPIPS | 0.0478 | 0.0332 | 0.0428 | 0.0335 | 0.0304 | 0.0183 | 0.0243 | 0.0288 | 0.0231 | 0.0314 |
| fruit | PSNR | 37.3497 | 41.3973 | 37.7726 | 38.5253 | 38.5838 | 38.2550 | 37.4183 | 36.8285 | 37.7129 | 38.2048 |
| | SSIM | 0.9833 | 0.9895 | 0.9880 | 0.9885 | 0.9887 | 0.9901 | 0.9897 | 0.9866 | 0.9896 | 0.9882 |
| | LAION AES | 5.2642 | 5.0681 | 5.0751 | 4.9006 | 4.8773 | 4.7383 | 4.8761 | 4.7219 | 4.6551 | 4.9085 |
| | rFID | 3.8448 | 4.3878 | 4.4777 | 3.6509 | 3.8026 | 3.4350 | 3.2648 | 3.1482 | 3.5220 | 3.7260 |
| | LPIPS | 0.0168 | 0.0132 | 0.0144 | 0.0119 | 0.0113 | 0.0089 | 0.0105 | 0.0116 | 0.0094 | 0.0120 |
| animal | PSNR | 36.3091 | 37.7983 | 36.1940 | 36.5812 | 36.3609 | 36.6225 | 36.4130 | 36.1365 | 36.3780 | 36.5326 |
| | SSIM | 0.9639 | 0.9676 | 0.9650 | 0.9672 | 0.9657 | 0.9678 | 0.9670 | 0.9659 | 0.9679 | 0.9664 |
| | LAION AES | 5.4965 | 5.3877 | 5.2624 | 5.1485 | 4.9439 | 4.9376 | 5.0422 | 5.0175 | 4.8832 | 5.1244 |
| | rFID | 1.9569 | 1.6617 | 2.0886 | 1.1434 | 1.6452 | 1.1288 | 1.3345 | 1.1932 | 1.1736 | 1.4807 |
| | LPIPS | 0.0326 | 0.0281 | 0.0328 | 0.0254 | 0.0259 | 0.0220 | 0.0240 | 0.0244 | 0.0224 | 0.0264 |
| overall | PSNR | 36.6044 | 39.7709 | 36.3363 | 36.9999 | 36.6834 | 36.9860 | 36.5232 | 36.1135 | 36.5081 | 36.9473 |
| | SSIM | 0.9615 | 0.9777 | 0.9692 | 0.9774 | 0.9722 | 0.9782 | 0.9765 | 0.9732 | 0.9787 | 0.9738 |
| | LAION AES | 5.2522 | 5.1018 | 5.0799 | 4.9807 | 4.8434 | 4.8243 | 4.9247 | 4.8328 | 4.7600 | 4.9555 |
| | rFID | 16.1651 | 13.8310 | 14.5107 | 11.2351 | 12.3226 | 10.2293 | 9.7922 | 10.8681 | 9.8811 | 12.0928 |
| | LPIPS | 0.0387 | 0.0338 | 0.0393 | 0.0290 | 0.0291 | 0.0186 | 0.0260 | 0.0232 | 0.0203 | 0.0287 |

Table 23: Metric scores across subtypes of AIM and color categories on VAE of FLUX (w/o GAN).

| Subtype | Metric | Colors | | | | | | | | | |
|---------|--------|--------|------|-------|-----|--------|-------|------|------|---------|---------|
| | | black | gray | white | red | yellow | green | cyan | blue | magenta | overall |
| portrait | PSNR | 40.3127 | 43.1050 | 40.1008 | 40.6687 | 40.3176 | 40.4744 | 40.0873 | 39.7835 | 40.1739 | 40.5582 |
| | SSIM | 0.9851 | 0.9898 | 0.9881 | 0.9891 | 0.9887 | 0.9900 | 0.9897 | 0.9876 | 0.9898 | 0.9887 |
| | LAION AES | 5.1579 | 5.1723 | 5.0656 | 5.1126 | 4.9149 | 5.0332 | 5.0337 | 4.9590 | 4.9445 | 5.0437 |
| | rFID | 4.7177 | 4.0622 | 4.2203 | 2.6370 | 3.5575 | 2.3273 | 2.9697 | 2.8760 | 2.1901 | 3.2842 |
| | LPIPS | 0.0163 | 0.0122 | 0.0146 | 0.0111 | 0.0111 | 0.0088 | 0.0100 | 0.0105 | 0.0091 | 0.0115 |
| transparent | PSNR | 31.6978 | 35.9299 | 30.0637 | 31.4137 | 30.6422 | 31.3051 | 30.4507 | 30.4539 | 30.4794 | 31.3818 |
| | SSIM | 0.8302 | 0.9563 | 0.9141 | 0.9593 | 0.9273 | 0.9540 | 0.9457 | 0.9451 | 0.9604 | 0.9325 |
| | LAION AES | 5.0865 | 4.8040 | 4.9750 | 4.9098 | 4.7436 | 4.6837 | 4.8397 | 4.7517 | 4.5447 | 4.8154 |
| | rFID | 67.8687 | 64.0847 | 62.3713 | 54.5902 | 54.9412 | 55.2797 | 47.2112 | 44.7164 | 48.0989 | 55.4625 |
| | LPIPS | 0.1181 | 0.1063 | 0.1259 | 0.0953 | 0.0974 | 0.0550 | 0.0934 | 0.0652 | 0.0587 | 0.0906 |
| toy | PSNR | 38.2779 | 42.0241 | 39.0216 | 38.9291 | 38.9816 | 38.9048 | 38.7893 | 38.0785 | 38.7019 | 39.0788 |
| | SSIM | 0.9795 | 0.9883 | 0.9848 | 0.9873 | 0.9860 | 0.9885 | 0.9878 | 0.9850 | 0.9884 | 0.9862 |
| | LAION AES | 5.0568 | 5.0174 | 4.9094 | 4.7682 | 4.7291 | 4.6713 | 4.8324 | 4.7097 | 4.6481 | 4.8158 |
| | rFID | 5.6927 | 4.9151 | 5.6773 | 3.0320 | 3.9968 | 2.6756 | 3.6771 | 3.7614 | 2.5111 | 3.9932 |
| | LPIPS | 0.0212 | 0.0143 | 0.0188 | 0.0138 | 0.0134 | 0.0091 | 0.0115 | 0.0126 | 0.0098 | 0.0138 |
| furniture | PSNR | 35.7641 | 39.4242 | 35.6760 | 36.5599 | 36.0876 | 36.3383 | 35.9038 | 35.4099 | 36.0903 | 36.3616 |
| | SSIM | 0.9674 | 0.9776 | 0.9763 | 0.9766 | 0.9773 | 0.9787 | 0.9786 | 0.9742 | 0.9779 | 0.9761 |
| | LAION AES | 5.2140 | 5.1324 | 5.0774 | 4.9276 | 4.7818 | 4.8272 | 4.8681 | 4.7582 | 4.7255 | 4.9236 |
| | rFID | 10.7373 | 9.1049 | 6.7307 | 4.2414 | 5.1600 | 3.5609 | 4.9260 | 5.7686 | 3.5526 | 5.9758 |
| | LPIPS | 0.0307 | 0.0232 | 0.0266 | 0.0181 | 0.0185 | 0.0155 | 0.0170 | 0.0170 | 0.0136 | 0.0196 |
| plant | PSNR | 35.0361 | 39.5070 | 34.7445 | 35.6682 | 35.3230 | 35.6192 | 34.9934 | 34.5397 | 35.0301 | 35.6068 |
| | SSIM | 0.9499 | 0.9817 | 0.9697 | 0.9765 | 0.9747 | 0.9827 | 0.9811 | 0.9673 | 0.9806 | 0.9738 |
| | LAION AES | 5.3513 | 5.1623 | 5.1518 | 5.0218 | 4.8957 | 4.8691 | 4.9754 | 4.8932 | 4.8732 | 5.0215 |
| | rFID | 27.0960 | 13.9108 | 17.4389 | 14.3200 | 13.9386 | 12.2158 | 10.4104 | 13.2422 | 9.8923 | 14.7183 |
| | LPIPS | 0.0504 | 0.0318 | 0.0436 | 0.0360 | 0.0323 | 0.0203 | 0.0273 | 0.0318 | 0.0252 | 0.0332 |
| fruit | PSNR | 37.5714 | 41.5986 | 37.8741 | 38.8135 | 38.7407 | 38.3088 | 37.4106 | 36.9225 | 37.8578 | 38.3442 |
| | SSIM | 0.9841 | 0.9903 | 0.9888 | 0.9894 | 0.9895 | 0.9909 | 0.9905 | 0.9876 | 0.9905 | 0.9891 |
| | LAION AES | 5.2547 | 5.0525 | 5.0826 | 4.8519 | 4.8712 | 4.7276 | 4.8842 | 4.7136 | 4.6491 | 4.8986 |
| | rFID | 3.9267 | 8.1560 | 7.5610 | 3.6152 | 5.7479 | 4.5468 | 3.7096 | 3.6284 | 3.7382 | 4.9589 |
| | LPIPS | 0.0169 | 0.0136 | 0.0153 | 0.0121 | 0.0119 | 0.0091 | 0.0112 | 0.0119 | 0.0098 | 0.0124 |
| animal | PSNR | 36.4266 | 38.0187 | 36.3146 | 36.7724 | 36.5454 | 36.7522 | 36.4851 | 36.2234 | 36.5210 | 36.6733 |
| | SSIM | 0.9648 | 0.9686 | 0.9659 | 0.9683 | 0.9667 | 0.9688 | 0.9681 | 0.9670 | 0.9689 | 0.9675 |
| | LAION AES | 5.4833 | 5.4006 | 5.2633 | 5.1316 | 4.9413 | 4.9376 | 5.0491 | 5.0050 | 4.8667 | 5.1198 |
| | rFID | 1.9154 | 2.5115 | 2.4260 | 1.0960 | 1.9794 | 1.2870 | 1.5935 | 1.1721 | 1.1295 | 1.6789 |
| | LPIPS | 0.0348 | 0.0305 | 0.0346 | 0.0271 | 0.0276 | 0.0236 | 0.0257 | 0.0263 | 0.0240 | 0.0282 |
| overall | PSNR | 36.4409 | 39.9439 | 36.2565 | 36.9751 | 36.6626 | 36.8147 | 36.3029 | 35.9159 | 36.4078 | 36.8578 |
| | SSIM | 0.9516 | 0.9789 | 0.9697 | 0.9781 | 0.9729 | 0.9791 | 0.9774 | 0.9734 | 0.9795 | 0.9734 |
| | LAION AES | 5.2292 | 5.1059 | 5.0750 | 4.9605 | 4.8397 | 4.8214 | 4.9261 | 4.8272 | 4.7503 | 4.9484 |
| | rFID | 17.4221 | 15.2493 | 15.2036 | 11.9331 | 12.7602 | 11.6990 | 10.6425 | 10.7379 | 10.1590 | 12.8674 |
| | LPIPS | 0.0412 | 0.0331 | 0.0399 | 0.0305 | 0.0303 | 0.0198 | 0.0278 | 0.0250 | 0.0215 | 0.0299 |

Table 24: Metric scores across subtypes of AIM and color categories on VAE of FLUX (w/o LPIPS).

| Subtype | Metric | Colors | | | | | | | | | |
|---|---|---|---|---|---|---|---|---|---|---|---|
| | | black | gray | white | red | yellow | green | cyan | blue | magenta | overall |
| portrait | PSNR | 36.5699 | 40.8603 | 36.3526 | 37.0664 | 36.7739 | 36.7335 | 36.2499 | 36.0746 | 36.5464 | 37.0253 |
| | SSIM | 0.9786 | 0.9865 | 0.9831 | 0.9852 | 0.9843 | 0.9868 | 0.9861 | 0.9825 | 0.9864 | 0.9844 |
| | LAION AES | 5.1506 | 5.1543 | 5.0381 | 5.1037 | 4.9200 | 5.0348 | 5.0274 | 4.9431 | 4.9473 | 5.0355 |
| | rFID | 8.3673 | 6.8302 | 7.2476 | 5.4567 | 7.2509 | 5.0560 | 6.1597 | 5.7517 | 4.6405 | 6.3067 |
| | LPIPS | 0.0226 | 0.0169 | 0.0198 | 0.0173 | 0.0168 | 0.0145 | 0.0159 | 0.0164 | 0.0149 | 0.0172 |
| transparent | PSNR | 24.6968 | 28.3907 | 22.5216 | 23.2511 | 22.6062 | 22.9936 | 22.2289 | 22.5321 | 22.2666 | 23.4986 |
| | SSIM | 0.8408 | 0.9157 | 0.8687 | 0.9254 | 0.8851 | 0.9144 | 0.9058 | 0.9120 | 0.9239 | 0.8991 |
| | LAION AES | 5.2292 | 4.9488 | 5.0502 | 5.0339 | 4.8193 | 4.7780 | 4.9227 | 4.8435 | 4.6915 | 4.9241 |
| | rFID | 92.6935 | 96.6403 | 88.4724 | 106.9512 | 90.2430 | 108.7483 | 85.6653 | 97.7950 | 94.9947 | 95.8004 |
| | LPIPS | 0.1483 | 0.1550 | 0.1722 | 0.1672 | 0.1658 | 0.1282 | 0.1595 | 0.1338 | 0.1323 | 0.1514 |
| toy | PSNR | 34.3559 | 39.4420 | 35.3949 | 35.0692 | 35.1943 | 34.8597 | 34.8235 | 34.2477 | 34.9432 | 35.3700 |
| | SSIM | 0.9716 | 0.9839 | 0.9783 | 0.9826 | 0.9804 | 0.9843 | 0.9832 | 0.9787 | 0.9842 | 0.9808 |
| | LAION AES | 5.0525 | 4.9901 | 4.8852 | 4.7702 | 4.7363 | 4.6724 | 4.8369 | 4.7136 | 4.6383 | 4.8106 |
| | rFID | 8.7003 | 6.4695 | 9.6447 | 7.7383 | 8.0421 | 7.2861 | 7.7710 | 6.7193 | 6.8511 | 7.6914 |
| | LPIPS | 0.0290 | 0.0218 | 0.0259 | 0.0226 | 0.0215 | 0.0167 | 0.0196 | 0.0203 | 0.0176 | 0.0217 |
| furniture | PSNR | 32.7589 | 37.0403 | 33.0481 | 33.5777 | 33.2780 | 33.3160 | 32.9708 | 32.5393 | 33.2924 | 33.5357 |
| | SSIM | 0.9595 | 0.9719 | 0.9701 | 0.9709 | 0.9716 | 0.9734 | 0.9733 | 0.9672 | 0.9725 | 0.9700 |
| | LAION AES | 5.1759 | 5.1106 | 5.0695 | 4.9124 | 4.7914 | 4.8271 | 4.8798 | 4.7324 | 4.7079 | 4.9119 |
| | rFID | 18.7107 | 11.5729 | 10.3997 | 9.2245 | 10.3477 | 7.1648 | 8.3467 | 11.4726 | 8.0068 | 10.5829 |
| | LPIPS | 0.0394 | 0.0303 | 0.0314 | 0.0267 | 0.0249 | 0.0194 | 0.0222 | 0.0246 | 0.0208 | 0.0266 |
| plant | PSNR | 31.3221 | 36.5534 | 30.9424 | 31.7745 | 31.4213 | 31.5552 | 31.0367 | 30.7780 | 31.1777 | 31.8401 |
| | SSIM | 0.9365 | 0.9737 | 0.9546 | 0.9676 | 0.9620 | 0.9749 | 0.9720 | 0.9552 | 0.9729 | 0.9633 |
| | LAION AES | 5.3261 | 5.1218 | 5.1306 | 4.9927 | 4.8933 | 4.8455 | 4.9676 | 4.8510 | 4.8358 | 4.9960 |
| | rFID | 38.2853 | 24.2539 | 27.7379 | 26.0790 | 27.1514 | 24.1899 | 21.9179 | 23.2332 | 21.9544 | 26.0892 |
| | LPIPS | 0.0627 | 0.0464 | 0.0587 | 0.0586 | 0.0549 | 0.0426 | 0.0491 | 0.0512 | 0.0481 | 0.0525 |
| fruit | PSNR | 34.7017 | 39.2796 | 34.7033 | 35.7306 | 35.5984 | 35.0510 | 34.1333 | 33.8906 | 34.7829 | 35.3190 |
| | SSIM | 0.9795 | 0.9876 | 0.9838 | 0.9868 | 0.9851 | 0.9882 | 0.9874 | 0.9841 | 0.9880 | 0.9856 |
| | LAION AES | 5.2429 | 5.0035 | 5.0539 | 4.8877 | 4.8385 | 4.7300 | 4.8701 | 4.7303 | 4.6685 | 4.8917 |
| | rFID | 5.9514 | 7.6350 | 8.1947 | 7.6183 | 7.3796 | 6.7584 | 6.6411 | 6.1484 | 5.9167 | 6.9160 |
| | LPIPS | 0.0234 | 0.0179 | 0.0204 | 0.0175 | 0.0166 | 0.0146 | 0.0170 | 0.0175 | 0.0146 | 0.0177 |
| animal | PSNR | 33.9218 | 36.9023 | 33.6300 | 34.3093 | 34.0523 | 34.2028 | 33.7999 | 33.6530 | 33.9459 | 34.2686 |
| | SSIM | 0.9580 | 0.9644 | 0.9591 | 0.9638 | 0.9606 | 0.9644 | 0.9632 | 0.9613 | 0.9647 | 0.9622 |
| | LAION AES | 5.4859 | 5.3831 | 5.2456 | 5.1246 | 4.9470 | 4.9229 | 5.0481 | 4.9969 | 4.8607 | 5.1128 |
| | rFID | 3.1328 | 2.7494 | 3.6120 | 2.3258 | 3.3380 | 2.4305 | 2.8961 | 2.3582 | 2.4087 | 2.8057 |
| | LPIPS | 0.0390 | 0.0332 | 0.0392 | 0.0326 | 0.0329 | 0.0287 | 0.0312 | 0.0313 | 0.0291 | 0.0330 |
| overall | PSNR | 32.6182 | 36.9241 | 32.3704 | 32.9684 | 32.7035 | 32.6731 | 32.1776 | 31.9593 | 32.4222 | 32.9796 |
| | SSIM | 0.9464 | 0.9691 | 0.9568 | 0.9689 | 0.9613 | 0.9695 | 0.9673 | 0.9630 | 0.9704 | 0.9636 |
| | LAION AES | 5.2376 | 5.1017 | 5.0676 | 4.9750 | 4.8494 | 4.8301 | 4.9361 | 4.8301 | 4.7643 | 4.9547 |
| | rFID | 25.1202 | 22.3073 | 22.1870 | 23.6277 | 21.9647 | 23.0906 | 19.9140 | 21.9255 | 20.6818 | 22.3132 |
| | LPIPS | 0.0521 | 0.0459 | 0.0525 | 0.0489 | 0.0476 | 0.0378 | 0.0449 | 0.0422 | 0.0396 | 0.0457 |

Table 25: Metric scores across subtypes of AIM and color categories on VAE of FLUX (w/o ABMSE).

| Subtype | Metric | Colors | | | | | | | | | |
|---|---|---|---|---|---|---|---|---|---|---|---|
| | | black | gray | white | red | yellow | green | cyan | blue | magenta | overall |
| portrait | PSNR | 39.8594 | 42.4175 | 39.6565 | 40.2818 | 39.7875 | 40.0540 | 39.7738 | 39.4529 | 39.8838 | 40.1297 |
| | SSIM | 0.9821 | 0.9875 | 0.9858 | 0.9869 | 0.9866 | 0.9881 | 0.9877 | 0.9851 | 0.9878 | 0.9864 |
| | LAION AES | 5.1740 | 5.1882 | 5.0794 | 5.1274 | 4.9326 | 5.0511 | 5.0424 | 4.9615 | 4.9475 | 5.0560 |
| | rFID | 5.6228 | 6.8072 | 5.5474 | 2.6233 | 4.8984 | 2.9176 | 3.7935 | 3.1263 | 2.3099 | 4.1829 |
| | LPIPS | 0.0198 | 0.0160 | 0.0162 | 0.0122 | 0.0118 | 0.0090 | 0.0103 | 0.0112 | 0.0094 | 0.0129 |
| transparent | PSNR | 32.3013 | 35.3760 | 30.1507 | 31.6711 | 30.6555 | 31.5462 | 30.7704 | 30.9122 | 30.7468 | 31.5700 |
| | SSIM | 0.8776 | 0.9393 | 0.9032 | 0.9486 | 0.9142 | 0.9389 | 0.9307 | 0.9387 | 0.9472 | 0.9265 |
| | LAION AES | 5.1273 | 4.7623 | 4.9560 | 4.8911 | 4.6523 | 4.6665 | 4.8095 | 4.7096 | 4.5059 | 4.7867 |
| | rFID | 71.8542 | 64.1741 | 67.3221 | 56.9083 | 58.4857 | 51.6231 | 47.8705 | 52.6748 | 53.9203 | 58.3148 |
| | LPIPS | 0.1236 | 0.1245 | 0.1396 | 0.1048 | 0.1085 | 0.0637 | 0.0997 | 0.0716 | 0.0682 | 0.1005 |
| toy | PSNR | 38.2553 | 41.5137 | 38.9475 | 38.8648 | 38.8014 | 38.8767 | 38.8452 | 38.1234 | 38.7363 | 38.9960 |
| | SSIM | 0.9775 | 0.9860 | 0.9832 | 0.9854 | 0.9844 | 0.9867 | 0.9860 | 0.9830 | 0.9866 | 0.9843 |
| | LAION AES | 5.0622 | 5.0033 | 4.9130 | 4.7785 | 4.7467 | 4.6915 | 4.8485 | 4.7229 | 4.6544 | 4.8246 |
| | rFID | 5.7848 | 4.3267 | 5.4561 | 3.0303 | 3.8870 | 2.6298 | 3.6813 | 3.2973 | 2.4705 | 3.8404 |
| | LPIPS | 0.0233 | 0.0183 | 0.0201 | 0.0143 | 0.0137 | 0.0090 | 0.0111 | 0.0126 | 0.0098 | 0.0147 |
| furniture | PSNR | 35.6978 | 38.9967 | 35.6743 | 36.4394 | 35.9894 | 36.3411 | 36.0080 | 35.4468 | 36.0992 | 36.2992 |
| | SSIM | 0.9666 | 0.9757 | 0.9749 | 0.9752 | 0.9758 | 0.9773 | 0.9771 | 0.9727 | 0.9765 | 0.9746 |
| | LAION AES | 5.1999 | 5.1379 | 5.0773 | 4.9283 | 4.7900 | 4.8314 | 4.8730 | 4.7688 | 4.7382 | 4.9272 |
| | rFID | 11.1104 | 9.3358 | 7.0222 | 5.4066 | 5.1184 | 3.6534 | 4.7019 | 6.4608 | 4.5802 | 6.3766 |
| | LPIPS | 0.0323 | 0.0264 | 0.0251 | 0.0195 | 0.0175 | 0.0122 | 0.0145 | 0.0168 | 0.0134 | 0.0197 |
| plant | PSNR | 35.4279 | 39.2807 | 35.2373 | 36.0428 | 35.5821 | 35.9665 | 35.5285 | 35.0505 | 35.5541 | 35.9634 |
| | SSIM | 0.9506 | 0.9790 | 0.9696 | 0.9747 | 0.9738 | 0.9807 | 0.9793 | 0.9661 | 0.9787 | 0.9725 |
| | LAION AES | 5.3393 | 5.1485 | 5.1641 | 5.0203 | 4.9120 | 4.8701 | 4.9816 | 4.8746 | 4.8568 | 5.0186 |
| | rFID | 25.9331 | 17.9372 | 19.8829 | 13.1940 | 15.7381 | 10.7544 | 11.0308 | 12.3250 | 10.1484 | 15.2160 |
| | LPIPS | 0.0501 | 0.0370 | 0.0425 | 0.0347 | 0.0301 | 0.0184 | 0.0245 | 0.0293 | 0.0229 | 0.0322 |
| fruit | PSNR | 37.6748 | 41.2544 | 38.0453 | 38.9777 | 38.6990 | 38.3644 | 37.6477 | 37.1775 | 38.1958 | 38.4485 |
| | SSIM | 0.9836 | 0.9888 | 0.9870 | 0.9885 | 0.9876 | 0.9895 | 0.9888 | 0.9868 | 0.9895 | 0.9878 |
| | LAION AES | 5.2485 | 5.0089 | 5.0915 | 4.8455 | 4.8225 | 4.7314 | 4.8501 | 4.7143 | 4.6327 | 4.8828 |
| | rFID | 3.5917 | 7.8556 | 6.3795 | 2.8364 | 5.3965 | 4.0415 | 4.6720 | 3.2514 | 3.0356 | 4.5622 |
| | LPIPS | 0.0159 | 0.0147 | 0.0153 | 0.0111 | 0.0117 | 0.0087 | 0.0104 | 0.0107 | 0.0089 | 0.0119 |
| animal | PSNR | 35.9570 | 37.5422 | 35.8278 | 36.3250 | 36.0280 | 36.2946 | 36.0784 | 35.8128 | 36.0990 | 36.2183 |
| | SSIM | 0.9618 | 0.9656 | 0.9622 | 0.9656 | 0.9631 | 0.9659 | 0.9649 | 0.9641 | 0.9663 | 0.9644 |
| | LAION AES | 5.4873 | 5.4025 | 5.2781 | 5.1442 | 4.9510 | 4.9410 | 5.0528 | 5.0029 | 4.8759 | 5.1262 |
| | rFID | 1.8621 | 2.6713 | 2.7684 | 1.0969 | 2.1945 | 1.3686 | 1.8192 | 1.1151 | 1.1463 | 1.7825 |
| | LPIPS | 0.0327 | 0.0294 | 0.0335 | 0.0251 | 0.0262 | 0.0218 | 0.0240 | 0.0243 | 0.0221 | 0.0266 |
| overall | PSNR | 36.4534 | 39.4830 | 36.2199 | 36.9432 | 36.5061 | 36.7776 | 36.3789 | 35.9966 | 36.4736 | 36.8036 |
| | SSIM | 0.9571 | 0.9746 | 0.9666 | 0.9750 | 0.9694 | 0.9753 | 0.9735 | 0.9709 | 0.9761 | 0.9709 |
| | LAION AES | 5.2341 | 5.0931 | 5.0799 | 4.9622 | 4.8296 | 4.8261 | 4.9226 | 4.8221 | 4.7445 | 4.9460 |
| | rFID | 17.9656 | 16.1583 | 16.3398 | 12.1565 | 13.6741 | 10.9983 | 11.0813 | 11.7501 | 11.0873 | 13.4679 |
| | LPIPS | 0.0425 | 0.0380 | 0.0418 | 0.0317 | 0.0314 | 0.0204 | 0.0278 | 0.0252 | 0.0221 | 0.0312 |

Table 26: Metric scores across subtypes of AIM and color categories on VAE of SDXL (ours).

| Subtype | Metric | Colors | | | | | | | | | |
|---|---|---|---|---|---|---|---|---|---|---|---|
| | | black | gray | white | red | yellow | green | cyan | blue | magenta | overall |
| portrait | PSNR | 37.6675 | 38.7091 | 37.0896 | 37.6714 | 37.2382 | 37.5797 | 37.3591 | 37.3685 | 37.4252 | 37.5676 |
| | SSIM | 0.9671 | 0.9708 | 0.9694 | 0.9704 | 0.9699 | 0.9710 | 0.9707 | 0.9693 | 0.9709 | 0.9700 |
| | LAION AES | 5.1871 | 5.1979 | 5.1041 | 5.1119 | 4.9399 | 5.0166 | 5.0471 | 4.9506 | 4.9370 | 5.0547 |
| | rFID | 8.6374 | 10.1861 | 9.1757 | 6.0759 | 7.9020 | 5.9156 | 6.5183 | 5.7959 | 5.1613 | 7.2631 |
| | LPIPS | 0.0430 | 0.0423 | 0.0421 | 0.0358 | 0.0361 | 0.0324 | 0.0338 | 0.0334 | 0.0326 | 0.0368 |
| transparent | PSNR | 34.0029 | 37.1520 | 33.0656 | 33.9251 | 33.3083 | 34.1245 | 33.7798 | 33.5730 | 33.5541 | 34.0539 |
| | SSIM | 0.9204 | 0.9635 | 0.9465 | 0.9626 | 0.9529 | 0.9648 | 0.9617 | 0.9540 | 0.9655 | 0.9547 |
| | LAION AES | 5.1802 | 4.8084 | 4.9807 | 4.8727 | 4.6822 | 4.6363 | 4.7891 | 4.7456 | 4.5068 | 4.8002 |
| | rFID | 36.9315 | 43.9862 | 47.1936 | 28.4102 | 34.1386 | 27.4021 | 28.1313 | 25.5091 | 27.1281 | 33.2034 |
| | LPIPS | 0.0869 | 0.0825 | 0.0886 | 0.0502 | 0.0588 | 0.0305 | 0.0472 | 0.0354 | 0.0306 | 0.0567 |
| toy | PSNR | 36.4159 | 37.8749 | 36.4495 | 36.5654 | 36.4125 | 36.6310 | 36.6108 | 36.3366 | 36.5080 | 36.6450 |
| | SSIM | 0.9620 | 0.9673 | 0.9657 | 0.9669 | 0.9663 | 0.9677 | 0.9673 | 0.9656 | 0.9676 | 0.9663 |
| | LAION AES | 5.0432 | 4.9998 | 4.9018 | 4.7618 | 4.7168 | 4.6712 | 4.8262 | 4.7146 | 4.6545 | 4.8100 |
| | rFID | 6.8024 | 6.0767 | 6.9535 | 5.1010 | 6.1531 | 4.9504 | 5.4167 | 5.2777 | 4.4130 | 5.6827 |
| | LPIPS | 0.0437 | 0.0412 | 0.0424 | 0.0346 | 0.0347 | 0.0305 | 0.0323 | 0.0327 | 0.0304 | 0.0358 |
| furniture | PSNR | 35.4915 | 37.6258 | 35.1162 | 35.9150 | 35.3368 | 35.8346 | 35.6057 | 35.2941 | 35.6632 | 35.7648 |
| | SSIM | 0.9573 | 0.9612 | 0.9612 | 0.9612 | 0.9615 | 0.9622 | 0.9620 | 0.9603 | 0.9619 | 0.9610 |
| | LAION AES | 5.2346 | 5.1203 | 5.0701 | 4.9287 | 4.7926 | 4.8239 | 4.8630 | 4.7768 | 4.7246 | 4.9261 |
| | rFID | 8.4432 | 9.8343 | 6.5405 | 5.5881 | 5.9534 | 4.8536 | 5.5054 | 6.5395 | 5.4140 | 6.5191 |
| | LPIPS | 0.0559 | 0.0528 | 0.0514 | 0.0401 | 0.0404 | 0.0364 | 0.0362 | 0.0362 | 0.0337 | 0.0423 |
| plant | PSNR | 35.2236 | 37.1924 | 34.7983 | 35.3626 | 34.9670 | 35.4417 | 35.1382 | 34.8529 | 35.0376 | 35.3349 |
| | SSIM | 0.9462 | 0.9626 | 0.9583 | 0.9609 | 0.9603 | 0.9634 | 0.9627 | 0.9572 | 0.9627 | 0.9594 |
| | LAION AES | 5.3706 | 5.1580 | 5.1856 | 5.0218 | 4.9258 | 4.8721 | 5.0002 | 4.8934 | 4.8644 | 5.0324 |
| | rFID | 20.5674 | 16.1455 | 16.8867 | 14.0630 | 14.4794 | 11.7910 | 10.4004 | 11.2129 | 10.4149 | 13.9957 |
| | LPIPS | 0.0612 | 0.0535 | 0.0568 | 0.0437 | 0.0419 | 0.0320 | 0.0366 | 0.0373 | 0.0343 | 0.0441 |
| fruit | PSNR | 37.3802 | 38.7392 | 37.4589 | 37.7600 | 37.6816 | 37.6517 | 37.2668 | 36.9641 | 37.3468 | 37.5833 |
| | SSIM | 0.9763 | 0.9784 | 0.9783 | 0.9784 | 0.9785 | 0.9789 | 0.9787 | 0.9779 | 0.9788 | 0.9782 |
| | LAION AES | 5.2286 | 5.0252 | 5.0940 | 4.8594 | 4.8842 | 4.7239 | 4.9024 | 4.7465 | 4.6542 | 4.9020 |
| | rFID | 5.3530 | 5.4807 | 4.9508 | 4.8144 | 5.7877 | 5.0477 | 4.9528 | 4.2923 | 4.3887 | 5.0076 |
| | LPIPS | 0.0317 | 0.0323 | 0.0306 | 0.0263 | 0.0262 | 0.0231 | 0.0241 | 0.0242 | 0.0233 | 0.0269 |
| animal | PSNR | 33.1621 | 33.7741 | 33.0834 | 33.2857 | 33.1532 | 33.3190 | 33.2449 | 33.1161 | 33.2264 | 33.2628 |
| | SSIM | 0.9116 | 0.9141 | 0.9130 | 0.9139 | 0.9134 | 0.9144 | 0.9141 | 0.9131 | 0.9144 | 0.9136 |
| | LAION AES | 5.4537 | 5.3677 | 5.2616 | 5.1041 | 4.9226 | 4.8961 | 5.0294 | 4.9775 | 4.8307 | 5.0937 |
| | rFID | 4.8643 | 5.5086 | 5.2704 | 4.3425 | 5.2651 | 4.5653 | 4.3445 | 4.3220 | 4.2939 | 4.7530 |
| | LPIPS | 0.1159 | 0.1149 | 0.1148 | 0.1016 | 0.1016 | 0.0983 | 0.0975 | 0.0946 | 0.1038 | 0.1038 |
| overall | PSNR | 35.6205 | 37.2953 | 35.2945 | 35.7836 | 35.4425 | 35.7975 | 35.5722 | 35.3579 | 35.5373 | 35.7446 |
| | SSIM | 0.9487 | 0.9597 | 0.9561 | 0.9592 | 0.9576 | 0.9604 | 0.9596 | 0.9568 | 0.9603 | 0.9576 |
| | LAION AES | 5.2426 | 5.0967 | 5.0854 | 4.9515 | 4.8378 | 4.8057 | 4.9225 | 4.8293 | 4.7389 | 4.9456 |
| | rFID | 13.0856 | 13.8883 | 13.8530 | 9.7707 | 11.3827 | 9.2180 | 9.3242 | 8.9928 | 8.7448 | 10.9178 |
| | LPIPS | 0.0626 | 0.0599 | 0.0609 | 0.0475 | 0.0485 | 0.0396 | 0.0441 | 0.0424 | 0.0399 | 0.0495 |

Table 27: Metric scores across subtypes of AIM and color categories on VAE of SDXL (w/o Norm KL).

| Subtype | Metric | Colors | | | | | | | | | |
|---|---|---|---|---|---|---|---|---|---|---|---|
| | | black | gray | white | red | yellow | green | cyan | blue | magenta | overall |
| portrait | PSNR | 36.4746 | 38.3820 | 35.9757 | 36.7160 | 36.3575 | 36.5646 | 36.1745 | 36.1212 | 36.3180 | 36.5649 |
| | SSIM | 0.9641 | 0.9698 | 0.9675 | 0.9689 | 0.9683 | 0.9701 | 0.9696 | 0.9671 | 0.9698 | 0.9684 |
| | LAION AES | 5.1593 | 5.1762 | 5.0724 | 5.0840 | 4.8959 | 5.0014 | 5.0296 | 4.9389 | 4.9164 | 5.0305 |
| | rFID | 11.4563 | 11.0646 | 10.9848 | 7.5095 | 9.6601 | 7.1380 | 7.7696 | 7.6093 | 6.9458 | 8.9042 |
| | LPIPS | 0.0462 | 0.0436 | 0.0453 | 0.0381 | 0.0385 | 0.0338 | 0.0361 | 0.0356 | 0.0341 | 0.0390 |
| transparent | PSNR | 29.2929 | 33.3247 | 27.8862 | 29.1718 | 28.7140 | 29.2333 | 28.3210 | 28.3100 | 28.2088 | 29.1625 |
| | SSIM | 0.8635 | 0.9420 | 0.9055 | 0.9410 | 0.9186 | 0.9423 | 0.9358 | 0.9244 | 0.9452 | 0.9243 |
| | LAION AES | 5.1107 | 4.7450 | 4.8962 | 4.9054 | 4.6995 | 4.6404 | 4.8111 | 4.7119 | 4.5554 | 4.7862 |
| | rFID | 83.7437 | 75.8321 | 81.9589 | 71.7247 | 73.8008 | 67.8855 | 62.5874 | 63.8879 | 66.9673 | 72.0431 |
| | LPIPS | 0.1363 | 0.1357 | 0.1500 | 0.1150 | 0.1149 | 0.0641 | 0.1078 | 0.0804 | 0.0711 | 0.1084 |
| toy | PSNR | 34.7239 | 37.3978 | 35.1756 | 35.2084 | 35.3314 | 35.2815 | 35.1367 | 34.6554 | 35.0577 | 35.3298 |
| | SSIM | 0.9561 | 0.9656 | 0.9628 | 0.9645 | 0.9641 | 0.9662 | 0.9656 | 0.9619 | 0.9658 | 0.9636 |
| | LAION AES | 5.0141 | 4.9707 | 4.9152 | 4.7470 | 4.7020 | 4.6439 | 4.8172 | 4.6924 | 4.6330 | 4.7928 |
| | rFID | 9.3693 | 8.0077 | 9.9768 | 7.6595 | 9.1844 | 7.1457 | 7.9718 | 7.1072 | 7.0030 | 8.1584 |
| | LPIPS | 0.0495 | 0.0451 | 0.0471 | 0.0401 | 0.0385 | 0.0331 | 0.0363 | 0.0374 | 0.0340 | 0.0401 |
| furniture | PSNR | 33.7311 | 36.7496 | 33.5228 | 34.3531 | 33.9765 | 34.2518 | 33.8279 | 33.4597 | 33.9589 | 34.2035 |
| | SSIM | 0.9503 | 0.9584 | 0.9579 | 0.9577 | 0.9586 | 0.9597 | 0.9596 | 0.9554 | 0.9589 | 0.9574 |
| | LAION AES | 5.1793 | 5.1193 | 5.0612 | 4.9008 | 4.7511 | 4.8157 | 4.8470 | 4.7555 | 4.7045 | 4.9038 |
| | rFID | 14.4978 | 11.9041 | 9.5858 | 8.2464 | 8.0627 | 6.0670 | 7.5150 | 9.7740 | 7.2797 | 9.2147 |
| | LPIPS | 0.0618 | 0.0546 | 0.0538 | 0.0429 | 0.0417 | 0.0343 | 0.0376 | 0.0388 | 0.0349 | 0.0445 |
| plant | PSNR | 32.6456 | 36.0937 | 32.2240 | 33.0099 | 32.8091 | 33.1223 | 32.5393 | 32.2122 | 32.4657 | 33.0135 |
| | SSIM | 0.9318 | 0.9585 | 0.9489 | 0.9544 | 0.9531 | 0.9584 | 0.9466 | 0.9580 | 0.9580 | 0.9522 |
| | LAION AES | 5.3206 | 5.1469 | 5.1530 | 4.9877 | 4.9164 | 4.8595 | 4.9772 | 4.8472 | 4.8576 | 5.0073 |
| | rFID | 30.2081 | 22.2946 | 24.4098 | 22.9032 | 22.7914 | 19.3908 | 16.0152 | 18.5697 | 17.8923 | 21.6083 |
| | LPIPS | 0.0751 | 0.0640 | 0.0701 | 0.0608 | 0.0556 | 0.0424 | 0.0497 | 0.0528 | 0.0487 | 0.0577 |
| fruit | PSNR | 35.1348 | 38.0898 | 35.6512 | 36.1435 | 36.3533 | 35.8992 | 35.2545 | 34.7241 | 35.4987 | 35.8610 |
| | SSIM | 0.9720 | 0.9769 | 0.9761 | 0.9762 | 0.9766 | 0.9776 | 0.9772 | 0.9748 | 0.9771 | 0.9761 |
| | LAION AES | 5.2276 | 4.9930 | 5.0883 | 4.8680 | 4.8560 | 4.7150 | 4.8682 | 4.7374 | 4.6646 | 4.8909 |
| | rFID | 7.0824 | 7.3004 | 6.9439 | 6.6512 | 6.6461 | 6.4959 | 7.1161 | 5.7505 | 6.3623 | 6.7025 |
| | LPIPS | 0.0356 | 0.0335 | 0.0334 | 0.0290 | 0.0282 | 0.0253 | 0.0268 | 0.0275 | 0.0259 | 0.0295 |
| animal | PSNR | 32.5646 | 33.5887 | 32.4113 | 32.7700 | 32.6374 | 32.8030 | 32.6049 | 32.4671 | 32.6082 | 32.7172 |
| | SSIM | 0.9079 | 0.9121 | 0.9094 | 0.9117 | 0.9102 | 0.9123 | 0.9116 | 0.9103 | 0.9124 | 0.9109 |
| | LAION AES | 5.4384 | 5.3602 | 5.2445 | 5.0956 | 4.9037 | 4.9012 | 5.0182 | 4.9666 | 4.8170 | 5.0828 |
| | rFID | 6.1956 | 6.4448 | 6.6242 | 5.3189 | 6.3728 | 5.5886 | 5.5038 | 5.1489 | 5.3434 | 5.8379 |
| | LPIPS | 0.1190 | 0.1156 | 0.1183 | 0.1035 | 0.1035 | 0.0952 | 0.0998 | 0.0995 | 0.0954 | 0.1055 |
| overall | PSNR | 33.5096 | 36.2323 | 33.2638 | 33.9104 | 33.7399 | 33.8794 | 33.4084 | 33.1357 | 33.4451 | 33.8361 |
| | SSIM | 0.9351 | 0.9548 | 0.9469 | 0.9535 | 0.9499 | 0.9554 | 0.9486 | 0.9540 | 0.9553 | 0.9504 |
| | LAION AES | 5.2071 | 5.0730 | 5.0615 | 4.9412 | 4.8178 | 4.7967 | 4.9098 | 4.8071 | 4.7355 | 4.9277 |
| | rFID | 23.2219 | 20.4069 | 21.4977 | 18.5696 | 19.5026 | 17.1016 | 16.3541 | 16.8354 | 16.8277 | 18.9242 |
| | LPIPS | 0.0748 | 0.0703 | 0.0740 | 0.0613 | 0.0601 | 0.0469 | 0.0563 | 0.0531 | 0.0492 | 0.0607 |

Table 28: Metric scores across subtypes of AIM and color categories on VAE of SDXL (w/o Ref KL).

| Subtype | Metric | Colors | | | | | | | | | |
|---|---|---|---|---|---|---|---|---|---|---|---|
| | | black | gray | white | red | yellow | green | cyan | blue | magenta | overall |
| portrait | PSNR | 36.5102 | 38.3645 | 36.0600 | 36.7652 | 36.4221 | 36.6211 | 36.2449 | 36.1715 | 36.3789 | 36.6154 |
| | SSIM | 0.9641 | 0.9697 | 0.9677 | 0.9689 | 0.9685 | 0.9700 | 0.9696 | 0.9670 | 0.9698 | 0.9684 |
| | LAION AES | 5.1657 | 5.1846 | 5.0962 | 5.0851 | 4.9031 | 5.0043 | 5.0288 | 4.9429 | 4.9228 | 5.0371 |
| | rFID | 11.6975 | 10.3056 | 9.7826 | 7.1901 | 9.0144 | 6.5155 | 7.2468 | 7.4109 | 6.7683 | 8.4369 |
| | LPIPS | 0.0455 | 0.0426 | 0.0439 | 0.0372 | 0.0373 | 0.0329 | 0.0351 | 0.0349 | 0.0334 | 0.0381 |
| transparent | PSNR | 29.9848 | 33.8929 | 28.4385 | 29.8346 | 29.2755 | 29.8737 | 28.9514 | 28.9870 | 28.8585 | 29.7885 |
| | SSIM | 0.8655 | 0.9449 | 0.9088 | 0.9446 | 0.9212 | 0.9450 | 0.9383 | 0.9286 | 0.9484 | 0.9273 |
| | LAION AES | 5.1389 | 4.7792 | 5.0217 | 4.9048 | 4.6819 | 4.6224 | 4.7688 | 4.7405 | 4.5380 | 4.7996 |
| | rFID | 82.9335 | 75.8111 | 81.5759 | 66.8182 | 67.3492 | 63.0626 | 54.0404 | 58.7423 | 60.8001 | 67.9037 |
| | LPIPS | 0.1331 | 0.1322 | 0.1445 | 0.1093 | 0.1100 | 0.0622 | 0.1045 | 0.0742 | 0.0671 | 0.1041 |
| toy | PSNR | 34.8420 | 37.4289 | 35.2512 | 35.3170 | 35.4120 | 35.4070 | 35.2391 | 34.7505 | 35.1522 | 35.4222 |
| | SSIM | 0.9563 | 0.9655 | 0.9627 | 0.9645 | 0.9639 | 0.9661 | 0.9619 | 0.9619 | 0.9657 | 0.9636 |
| | LAION AES | 5.0301 | 4.9677 | 4.9087 | 4.7619 | 4.7183 | 4.6655 | 4.8353 | 4.7061 | 4.6462 | 4.8044 |
| | rFID | 9.3078 | 7.6350 | 9.6481 | 7.5463 | 9.1604 | 6.6476 | 7.0186 | 7.1434 | 6.4199 | 7.8363 |
| | LPIPS | 0.0487 | 0.0449 | 0.0466 | 0.0395 | 0.0379 | 0.0326 | 0.0357 | 0.0367 | 0.0335 | 0.0396 |
| furniture | PSNR | 33.8852 | 36.7363 | 33.5582 | 34.4618 | 34.0315 | 34.3705 | 33.9374 | 33.6103 | 34.0572 | 34.2943 |
| | SSIM | 0.9507 | 0.9584 | 0.9578 | 0.9580 | 0.9584 | 0.9596 | 0.9595 | 0.9560 | 0.9590 | 0.9575 |
| | LAION AES | 5.1828 | 5.1056 | 5.0628 | 4.9264 | 4.7675 | 4.8128 | 4.8562 | 4.7774 | 4.7194 | 4.9123 |
| | rFID | 14.7214 | 13.1675 | 10.3931 | 8.1321 | 9.0268 | 6.1002 | 8.1966 | 9.8434 | 7.1342 | 9.6350 |
| | LPIPS | 0.0574 | 0.0522 | 0.0512 | 0.0402 | 0.0394 | 0.0354 | 0.0361 | 0.0394 | 0.0329 | 0.0419 |
| plant | PSNR | 32.8763 | 36.1623 | 32.4867 | 33.2636 | 33.0588 | 33.3588 | 32.7822 | 32.4405 | 32.7222 | 33.2390 |
| | SSIM | 0.9339 | 0.9589 | 0.9498 | 0.9551 | 0.9537 | 0.9601 | 0.9587 | 0.9479 | 0.9584 | 0.9529 |
| | LAION AES | 5.3277 | 5.1496 | 5.1608 | 4.9865 | 4.9031 | 4.8378 | 4.9762 | 4.8447 | 4.8448 | 5.0035 |
| | rFID | 30.4734 | 22.7255 | 24.0956 | 22.1485 | 22.5562 | 18.2961 | 16.4897 | 17.4623 | 17.5252 | 21.3081 |
| | LPIPS | 0.0719 | 0.0621 | 0.0677 | 0.0575 | 0.0527 | 0.0401 | 0.0471 | 0.0500 | 0.0462 | 0.0550 |
| fruit | PSNR | 35.0032 | 37.9632 | 35.5167 | 36.0654 | 36.2604 | 35.8223 | 35.1289 | 34.5998 | 35.3952 | 35.7506 |
| | SSIM | 0.9715 | 0.9765 | 0.9757 | 0.9758 | 0.9762 | 0.9772 | 0.9769 | 0.9742 | 0.9767 | 0.9756 |
| | LAION AES | 5.2299 | 5.0317 | 5.1168 | 4.8882 | 4.9300 | 4.7544 | 4.9370 | 4.7473 | 4.6697 | 4.9228 |
| | rFID | 7.5752 | 6.6650 | 7.2798 | 6.9078 | 7.0217 | 6.5673 | 6.2841 | 5.9389 | 7.0940 | 6.8149 |
| | LPIPS | 0.0343 | 0.0330 | 0.0328 | 0.0283 | 0.0278 | 0.0249 | 0.0264 | 0.0269 | 0.0253 | 0.0289 |
| animal | PSNR | 32.5627 | 33.5600 | 32.4198 | 32.7692 | 32.6389 | 32.8029 | 32.6095 | 32.4679 | 32.6107 | 32.7157 |
| | SSIM | 0.9081 | 0.9121 | 0.9095 | 0.9118 | 0.9103 | 0.9123 | 0.9116 | 0.9104 | 0.9125 | 0.9110 |
| | LAION AES | 5.4446 | 5.3727 | 5.2553 | 5.1006 | 4.9137 | 4.8993 | 5.0197 | 4.9711 | 4.8224 | 5.0888 |
| | rFID | 6.1257 | 6.8085 | 6.7694 | 5.3915 | 6.5794 | 5.5923 | 5.5619 | 5.2625 | 5.4051 | 5.9440 |
| | LPIPS | 0.1162 | 0.1143 | 0.1162 | 0.1013 | 0.1017 | 0.0981 | 0.0973 | 0.0937 | 0.0937 | 0.1036 |
| overall | PSNR | 33.6663 | 36.3012 | 33.3902 | 34.0681 | 33.8713 | 34.0366 | 33.5562 | 33.2896 | 33.5964 | 33.9751 |
| | SSIM | 0.9357 | 0.9551 | 0.9474 | 0.9541 | 0.9503 | 0.9558 | 0.9543 | 0.9494 | 0.9558 | 0.9509 |
| | LAION AES | 5.2171 | 5.0844 | 5.0889 | 4.9505 | 4.8311 | 4.7995 | 4.9174 | 4.8186 | 4.7376 | 4.9383 |
| | rFID | 23.2621 | 20.4455 | 21.3635 | 17.7335 | 18.6726 | 16.1117 | 14.9769 | 15.9720 | 15.8781 | 18.2684 |
| | LPIPS | 0.0724 | 0.0688 | 0.0718 | 0.0590 | 0.0581 | 0.0455 | 0.0546 | 0.0509 | 0.0474 | 0.0587 |

2700
2701
2702
2703
2704
2705
2706
2707
2708
2709
2710
2711
2712
2713

Table 29: Metric scores across subtypes of AIM and color categories on VAE of SDXL (w/o GAN).

| Subtype | Metric | Colors | | | | | | | | | |
|---------|--------|--------|------|-------|-----|--------|-------|------|------|---------|---------|
| | | black | gray | white | red | yellow | green | cyan | blue | magenta | overall |
| portrait | PSNR | 36.7320 | 38.5246 | 36.2216 | 36.9466 | 36.5775 | 36.8067 | 36.4386 | 36.3937 | 36.5608 | 36.8002 |
| | SSIM | 0.9647 | 0.9704 | 0.9683 | 0.9696 | 0.9691 | 0.9707 | 0.9702 | 0.9678 | 0.9704 | 0.9690 |
| | LAION AES | 5.1517 | 5.1743 | 5.0571 | 5.0904 | 4.8949 | 5.0084 | 5.0292 | 4.9452 | 4.9174 | 5.0298 |
| | rFID | 11.3116 | 11.4391 | 10.1436 | 7.1283 | 9.4828 | 6.6206 | 7.6032 | 7.1238 | 6.7191 | 8.6191 |
| | LPIPS | 0.0457 | 0.0429 | 0.0449 | 0.0375 | 0.0379 | 0.0332 | 0.0354 | 0.0350 | 0.0336 | 0.0385 |
| transparent | PSNR | 29.5129 | 33.7601 | 27.9406 | 29.3151 | 28.7980 | 29.4456 | 28.4825 | 28.5114 | 28.3432 | 29.3455 |
| | SSIM | 0.8120 | 0.9469 | 0.9099 | 0.9442 | 0.9233 | 0.9461 | 0.9401 | 0.9259 | 0.9486 | 0.9219 |
| | LAION AES | 5.1378 | 4.7396 | 4.9448 | 4.9050 | 4.6889 | 4.6468 | 4.7928 | 4.7330 | 4.5242 | 4.7903 |
| | rFID | 90.2189 | 71.4112 | 78.7081 | 72.1483 | 71.7091 | 64.5011 | 57.2832 | 59.7343 | 66.3588 | 70.2303 |
| | LPIPS | 0.1409 | 0.1296 | 0.1455 | 0.1116 | 0.1105 | 0.0615 | 0.1067 | 0.0776 | 0.0680 | 0.1058 |
| toy | PSNR | 35.0278 | 37.6045 | 35.4149 | 35.4882 | 35.5817 | 35.5749 | 35.4023 | 34.9298 | 35.3101 | 35.5927 |
| | SSIM | 0.9559 | 0.9664 | 0.9636 | 0.9651 | 0.9648 | 0.9669 | 0.9664 | 0.9625 | 0.9664 | 0.9642 |
| | LAION AES | 5.0197 | 4.9713 | 4.9149 | 4.7523 | 4.7135 | 4.6555 | 4.8358 | 4.6951 | 4.6363 | 4.7994 |
| | rFID | 9.2848 | 7.9326 | 9.8077 | 7.3926 | 8.6993 | 6.2666 | 7.3856 | 6.8320 | 6.3730 | 7.7749 |
| | LPIPS | 0.0486 | 0.0433 | 0.0463 | 0.0393 | 0.0378 | 0.0323 | 0.0355 | 0.0365 | 0.0332 | 0.0392 |
| furniture | PSNR | 33.9997 | 37.0421 | 33.6614 | 34.5392 | 34.0922 | 34.4621 | 34.0329 | 33.7113 | 34.1270 | 34.4075 |
| | SSIM | 0.9506 | 0.9598 | 0.9585 | 0.9591 | 0.9593 | 0.9607 | 0.9605 | 0.9570 | 0.9601 | 0.9584 |
| | LAION AES | 5.1857 | 5.1232 | 5.0548 | 4.9165 | 4.7868 | 4.8183 | 4.8588 | 4.7536 | 4.7142 | 4.9124 |
| | rFID | 14.0972 | 11.2461 | 9.6543 | 7.9404 | 7.9274 | 6.1824 | 7.4099 | 8.9573 | 7.0943 | 8.9455 |
| | LPIPS | 0.0604 | 0.0540 | 0.0554 | 0.0429 | 0.0427 | 0.0348 | 0.0382 | 0.0386 | 0.0352 | 0.0447 |
| plant | PSNR | 32.9529 | 36.3510 | 32.5872 | 33.2800 | 33.1096 | 33.4526 | 32.9221 | 32.5458 | 32.7738 | 33.3306 |
| | SSIM | 0.9302 | 0.9601 | 0.9511 | 0.9558 | 0.9552 | 0.9612 | 0.9601 | 0.9482 | 0.9593 | 0.9535 |
| | LAION AES | 5.3265 | 5.1676 | 5.1845 | 4.9799 | 4.9262 | 4.8513 | 4.9897 | 4.8501 | 4.8479 | 5.0137 |
| | rFID | 30.7646 | 21.2863 | 23.9412 | 21.8784 | 22.9284 | 18.0878 | 16.1799 | 17.1372 | 17.1516 | 21.0395 |
| | LPIPS | 0.0732 | 0.0601 | 0.0682 | 0.0584 | 0.0535 | 0.0405 | 0.0480 | 0.0505 | 0.0462 | 0.0554 |
| fruit | PSNR | 35.2374 | 38.2192 | 35.7600 | 36.2187 | 36.4331 | 36.0560 | 35.4204 | 34.8591 | 35.5977 | 35.9780 |
| | SSIM | 0.9717 | 0.9776 | 0.9767 | 0.9767 | 0.9772 | 0.9782 | 0.9780 | 0.9751 | 0.9776 | 0.9765 |
| | LAION AES | 5.2389 | 5.0181 | 5.0921 | 4.8644 | 4.8912 | 4.7368 | 4.7578 | 4.6620 | 4.6620 | 4.9069 |
| | rFID | 7.4422 | 9.2016 | 8.0375 | 6.6245 | 8.6047 | 6.9794 | 7.6668 | 5.9018 | 6.5614 | 7.4467 |
| | LPIPS | 0.0353 | 0.0340 | 0.0341 | 0.0293 | 0.0287 | 0.0253 | 0.0270 | 0.0274 | 0.0260 | 0.0297 |
| animal | PSNR | 32.6954 | 33.6484 | 32.5677 | 32.8898 | 32.7663 | 32.9277 | 32.7527 | 32.6096 | 32.7465 | 32.8449 |
| | SSIM | 0.9083 | 0.9126 | 0.9101 | 0.9121 | 0.9109 | 0.9128 | 0.9122 | 0.9108 | 0.9128 | 0.9114 |
| | LAION AES | 5.4412 | 5.3642 | 5.2427 | 5.0998 | 4.8958 | 4.8931 | 5.0124 | 4.9792 | 4.8283 | 5.0841 |
| | rFID | 5.8278 | 6.4157 | 6.3684 | 5.1010 | 6.0304 | 5.2818 | 5.2617 | 5.0299 | 5.0766 | 5.5993 |
| | LPIPS | 0.1191 | 0.1159 | 0.1186 | 0.1035 | 0.1036 | 0.0954 | 0.0998 | 0.0994 | 0.0955 | 0.1056 |
| overall | PSNR | 33.7369 | 36.4500 | 33.4505 | 34.0968 | 33.9083 | 34.1037 | 33.6359 | 33.3658 | 33.6370 | 34.0428 |
| | SSIM | 0.9276 | 0.9563 | 0.9483 | 0.9547 | 0.9514 | 0.9567 | 0.9554 | 0.9496 | 0.9565 | 0.9507 |
| | LAION AES | 5.2145 | 5.0798 | 5.0701 | 4.9440 | 4.8282 | 4.8015 | 4.9171 | 4.8163 | 4.7329 | 4.9338 |
| | rFID | 24.1353 | 19.8475 | 20.9515 | 18.3162 | 19.3403 | 16.2742 | 15.5415 | 15.8166 | 16.4764 | 18.5222 |
| | LPIPS | 0.0747 | 0.0685 | 0.0733 | 0.0604 | 0.0592 | 0.0461 | 0.0558 | 0.0521 | 0.0482 | 0.0598 |

Table 30: Metric scores across subtypes of AIM and color categories on VAE of SDXL (w/o LPIPS).

| Subtype | Metric | Colors | | | | | | | | | |
|---|---|---|---|---|---|---|---|---|---|---|---|
| | | black | gray | white | red | yellow | green | cyan | blue | magenta | overall |
| portrait | PSNR | 36.3674 | 38.3427 | 35.8862 | 36.7687 | 36.2056 | 36.2692 | 35.9600 | 36.0784 | 36.4217 | 36.4778 |
| | SSIM | 0.9641 | 0.9697 | 0.9674 | 0.9689 | 0.9682 | 0.9700 | 0.9695 | 0.9671 | 0.9698 | 0.9683 |
| | LAION AES | 5.1359 | 5.1682 | 5.0700 | 5.0833 | 4.9013 | 4.9964 | 5.0246 | 4.9396 | 4.9223 | 5.0268 |
| | rFID | 10.8159 | 10.3261 | 10.1067 | 7.1798 | 9.1609 | 6.8984 | 8.2479 | 7.2155 | 6.2245 | 8.4640 |
| | LPIPS | 0.0457 | 0.0433 | 0.0446 | 0.0370 | 0.0382 | 0.0339 | 0.0360 | 0.0351 | 0.0332 | 0.0386 |
| transparent | PSNR | 29.8215 | 33.6546 | 28.3403 | 29.9983 | 28.8181 | 29.1258 | 28.6430 | 29.0760 | 29.2703 | 29.6387 |
| | SSIM | 0.8789 | 0.9433 | 0.9044 | 0.9449 | 0.9176 | 0.9434 | 0.9361 | 0.9308 | 0.9482 | 0.9275 |
| | LAION AES | 5.1506 | 4.7543 | 4.9682 | 4.8956 | 4.6648 | 4.6381 | 4.8149 | 4.7318 | 4.5253 | 4.7937 |
| | rFID | 82.1560 | 72.0828 | 78.3937 | 64.3743 | 73.5320 | 69.7988 | 57.3710 | 56.0134 | 62.7314 | 68.4948 |
| | LPIPS | 0.1279 | 0.1351 | 0.1461 | 0.1053 | 0.1139 | 0.0672 | 0.1067 | 0.0728 | 0.0623 | 0.1041 |
| toy | PSNR | 34.7257 | 37.4039 | 35.2196 | 35.3687 | 35.2077 | 35.0073 | 35.0335 | 34.7860 | 35.3667 | 35.3466 |
| | SSIM | 0.9567 | 0.9653 | 0.9625 | 0.9645 | 0.9638 | 0.9659 | 0.9653 | 0.9620 | 0.9657 | 0.9635 |
| | LAION AES | 5.0256 | 4.9785 | 4.9219 | 4.7594 | 4.7233 | 4.6627 | 4.8177 | 4.7081 | 4.6457 | 4.8048 |
| | rFID | 9.4547 | 9.7439 | 10.0182 | 6.5025 | 10.4319 | 6.9818 | 7.9857 | 6.8161 | 5.7989 | 8.1926 |
| | LPIPS | 0.0485 | 0.0447 | 0.0462 | 0.0384 | 0.0383 | 0.0336 | 0.0362 | 0.0361 | 0.0323 | 0.0394 |
| furniture | PSNR | 33.7006 | 36.6126 | 33.2743 | 34.4088 | 33.6803 | 33.8824 | 33.5253 | 33.4577 | 34.0421 | 34.0649 |
| | SSIM | 0.9516 | 0.9585 | 0.9576 | 0.9583 | 0.9583 | 0.9596 | 0.9594 | 0.9563 | 0.9593 | 0.9577 |
| | LAION AES | 5.1837 | 5.1192 | 5.0696 | 4.9110 | 4.7930 | 4.8190 | 4.8684 | 4.7616 | 4.7129 | 4.9154 |
| | rFID | 13.7325 | 13.3523 | 9.6613 | 8.2026 | 9.9544 | 7.0330 | 8.1623 | 8.8281 | 7.2228 | 9.5721 |
| | LPIPS | 0.0581 | 0.0537 | 0.0519 | 0.0401 | 0.0405 | 0.0366 | 0.0336 | 0.0367 | 0.0329 | 0.0427 |
| plant | PSNR | 32.8920 | 36.2432 | 32.6305 | 33.4965 | 32.8779 | 32.9688 | 32.7490 | 32.7491 | 33.2666 | 33.3193 |
| | SSIM | 0.9356 | 0.9591 | 0.9497 | 0.9557 | 0.9537 | 0.9604 | 0.9589 | 0.9489 | 0.9588 | 0.9534 |
| | LAION AES | 5.3360 | 5.1277 | 5.1541 | 4.9762 | 4.9097 | 4.8351 | 4.9661 | 4.8495 | 4.8415 | 4.9995 |
| | rFID | 31.5805 | 26.0046 | 24.4139 | 21.3878 | 23.2555 | 21.8749 | 18.3493 | 18.1195 | 16.9483 | 22.4371 |
| | LPIPS | 0.0715 | 0.0632 | 0.0690 | 0.0556 | 0.0559 | 0.0437 | 0.0487 | 0.0485 | 0.0430 | 0.0555 |
| fruit | PSNR | 35.3122 | 37.9947 | 35.6403 | 36.4715 | 36.0864 | 35.5568 | 35.1326 | 35.0429 | 35.9749 | 35.9125 |
| | SSIM | 0.9732 | 0.9772 | 0.9754 | 0.9769 | 0.9760 | 0.9777 | 0.9770 | 0.9757 | 0.9777 | 0.9763 |
| | LAION AES | 5.2457 | 5.0548 | 5.0950 | 4.8876 | 4.9055 | 4.7401 | 4.8920 | 4.7345 | 4.6761 | 4.9146 |
| | rFID | 6.1697 | 7.1512 | 7.9268 | 5.8491 | 9.3041 | 8.4195 | 8.0415 | 6.3779 | 5.7182 | 7.2176 |
| | LPIPS | 0.0350 | 0.0346 | 0.0345 | 0.0273 | 0.0285 | 0.0249 | 0.0266 | 0.0262 | 0.0239 | 0.0291 |
| animal | PSNR | 32.5041 | 33.5622 | 32.3211 | 32.7738 | 32.5149 | 32.6105 | 32.4568 | 32.4371 | 32.6487 | 32.6477 |
| | SSIM | 0.9078 | 0.9117 | 0.9089 | 0.9115 | 0.9097 | 0.9119 | 0.9112 | 0.9101 | 0.9122 | 0.9106 |
| | LAION AES | 5.4546 | 5.3724 | 5.2595 | 5.1001 | 4.9041 | 4.8862 | 5.0174 | 4.9748 | 4.8283 | 5.0886 |
| | rFID | 6.1345 | 6.4167 | 6.4351 | 5.3497 | 6.4877 | 5.7182 | 5.5588 | 5.1923 | 5.3278 | 5.8468 |
| | LPIPS | 0.1177 | 0.1150 | 0.1175 | 0.1025 | 0.1033 | 0.0955 | 0.0999 | 0.0986 | 0.0947 | 0.1050 |
| overall | PSNR | 33.6176 | 36.2591 | 33.3303 | 34.1838 | 33.6273 | 33.6315 | 33.3572 | 33.3753 | 33.8559 | 33.9153 |
| | SSIM | 0.9383 | 0.9550 | 0.9466 | 0.9544 | 0.9496 | 0.9556 | 0.9539 | 0.9501 | 0.9560 | 0.9511 |
| | LAION AES | 5.2189 | 5.0822 | 5.0769 | 4.9447 | 4.8288 | 4.7968 | 4.9144 | 4.8143 | 4.7360 | 4.9348 |
| | rFID | 22.8634 | 20.7254 | 20.9937 | 16.9780 | 20.3038 | 18.1035 | 16.2452 | 15.5090 | 15.7103 | 18.6036 |
| | LPIPS | 0.0721 | 0.0699 | 0.0728 | 0.0580 | 0.0598 | 0.0475 | 0.0558 | 0.0506 | 0.0460 | 0.0592 |

Table 31: Metric scores across subtypes of AIM and color categories on VAE of SDXL (w/o ABMSE).

| Subtype | Metric | Colors | | | | | | | | | |
|---------|--------|--------|------|-------|------|--------|-------|------|------|---------|---------|
| | | black | gray | white | red | yellow | green | cyan | blue | magenta | overall |
| portrait | PSNR | 36.7638 | 38.0540 | 36.1074 | 36.9684 | 36.4335 | 36.7649 | 36.4643 | 36.4923 | 36.6376 | 36.7429 |
| | SSIM | 0.9628 | 0.9666 | 0.9624 | 0.9675 | 0.9641 | 0.9674 | 0.9664 | 0.9659 | 0.9681 | 0.9657 |
| | LAION AES | 5.1830 | 5.2021 | 5.0998 | 5.1076 | 4.9582 | 5.0416 | 5.0637 | 4.9568 | 4.9539 | 5.0630 |
| | rFID | 10.4060 | 14.1848 | 12.3171 | 7.6433 | 11.2678 | 7.4332 | 8.7145 | 7.3389 | 7.1011 | 9.6007 |
| | LPIPS | 0.0463 | 0.0519 | 0.0547 | 0.0368 | 0.0400 | 0.0327 | 0.0353 | 0.0341 | 0.0329 | 0.0405 |
| transparent | PSNR | 31.6651 | 35.3412 | 30.7818 | 31.7570 | 31.3339 | 32.0047 | 31.4061 | 31.1377 | 31.1593 | 31.8430 |
| | SSIM | 0.8937 | 0.9514 | 0.9235 | 0.9521 | 0.9338 | 0.9535 | 0.9481 | 0.9391 | 0.9559 | 0.9390 |
| | LAION AES | 5.1738 | 4.7540 | 4.9455 | 4.8714 | 4.6410 | 4.6142 | 4.7487 | 4.7358 | 4.5102 | 4.7772 |
| | rFID | 64.5636 | 59.6612 | 67.7993 | 46.9943 | 56.6676 | 42.4200 | 46.7917 | 45.6588 | 43.8324 | 52.7099 |
| | LPIPS | 0.1193 | 0.1266 | 0.1316 | 0.0894 | 0.0940 | 0.0479 | 0.0814 | 0.0590 | 0.0508 | 0.0889 |
| toy | PSNR | 35.2063 | 37.2530 | 35.5171 | 35.6248 | 35.5931 | 35.7167 | 35.6567 | 35.2110 | 35.5724 | 35.7057 |
| | SSIM | 0.9554 | 0.9627 | 0.9583 | 0.9631 | 0.9603 | 0.9637 | 0.9628 | 0.9608 | 0.9642 | 0.9613 |
| | LAION AES | 5.0388 | 4.9659 | 4.8571 | 4.7684 | 4.7347 | 4.6770 | 4.8403 | 4.7163 | 4.6615 | 4.8067 |
| | rFID | 10.1643 | 9.3191 | 11.5638 | 7.3923 | 9.1299 | 6.5489 | 7.8263 | 7.0808 | 6.1367 | 8.3513 |
| | LPIPS | 0.0508 | 0.0520 | 0.0561 | 0.0389 | 0.0400 | 0.0321 | 0.0353 | 0.0361 | 0.0328 | 0.0416 |
| furniture | PSNR | 34.3347 | 36.6940 | 34.1015 | 34.9100 | 34.4811 | 34.8614 | 34.5162 | 34.1372 | 34.5826 | 34.7354 |
| | SSIM | 0.9515 | 0.9571 | 0.9553 | 0.9578 | 0.9566 | 0.9586 | 0.9583 | 0.9562 | 0.9587 | 0.9567 |
| | LAION AES | 5.1950 | 5.1201 | 5.0741 | 4.9125 | 4.8086 | 4.8210 | 4.8715 | 4.7553 | 4.7145 | 4.9192 |
| | rFID | 13.2863 | 12.6409 | 10.0453 | 8.1039 | 8.4774 | 6.0336 | 7.5791 | 9.4041 | 6.9290 | 9.1666 |
| | LPIPS | 0.0514 | 0.0518 | 0.0498 | 0.0360 | 0.0359 | 0.0284 | 0.0313 | 0.0317 | 0.0292 | 0.0359 |
| plant | PSNR | 33.7658 | 36.3762 | 33.4606 | 34.1167 | 33.8582 | 34.2416 | 33.8079 | 33.4284 | 33.7074 | 34.0848 |
| | SSIM | 0.9374 | 0.9576 | 0.9493 | 0.9558 | 0.9532 | 0.9592 | 0.9579 | 0.9499 | 0.9585 | 0.9532 |
| | LAION AES | 5.3573 | 5.1615 | 5.1724 | 5.0271 | 4.9433 | 4.8847 | 5.0038 | 4.8811 | 4.8887 | 5.0355 |
| | rFID | 26.7165 | 24.4358 | 20.8311 | 18.6915 | 19.6878 | 14.9545 | 14.0651 | 15.1785 | 14.9320 | 18.8325 |
| | LPIPS | 0.0685 | 0.0654 | 0.0696 | 0.0509 | 0.0489 | 0.0352 | 0.0419 | 0.0438 | 0.0397 | 0.0515 |
| fruit | PSNR | 35.3166 | 37.9066 | 35.8143 | 36.3363 | 36.4182 | 36.1277 | 35.5804 | 35.0427 | 35.8157 | 36.0398 |
| | SSIM | 0.9710 | 0.9743 | 0.9714 | 0.9749 | 0.9727 | 0.9753 | 0.9745 | 0.9736 | 0.9756 | 0.9737 |
| | LAION AES | 5.2249 | 5.0587 | 5.0752 | 4.8909 | 4.9181 | 4.7495 | 4.8984 | 4.7659 | 4.6892 | 4.9190 |
| | rFID | 6.9056 | 8.4866 | 8.7358 | 5.7101 | 7.7177 | 7.4640 | 6.4369 | 5.9940 | 5.6630 | 7.0126 |
| | LPIPS | 0.0331 | 0.0359 | 0.0377 | 0.0264 | 0.0285 | 0.0231 | 0.0248 | 0.0246 | 0.0236 | 0.0286 |
| animal | PSNR | 32.6555 | 33.3943 | 32.4677 | 32.8334 | 32.6445 | 32.8359 | 32.6955 | 32.5905 | 32.7129 | 32.7589 |
| | SSIM | 0.9074 | 0.9092 | 0.9048 | 0.9105 | 0.9064 | 0.9098 | 0.9087 | 0.9095 | 0.9109 | 0.9086 |
| | LAION AES | 5.4574 | 5.3803 | 5.2339 | 5.1234 | 4.9623 | 4.9110 | 5.0662 | 4.9875 | 4.8588 | 5.1090 |
| | rFID | 6.1759 | 7.7162 | 7.7732 | 5.7749 | 7.2939 | 6.0623 | 6.0354 | 5.3595 | 5.7247 | 6.4351 |
| | LPIPS | 0.1112 | 0.1162 | 0.1188 | 0.0977 | 0.0999 | 0.0906 | 0.0949 | 0.0936 | 0.0907 | 0.1015 |
| overall | PSNR | 34.2440 | 36.4313 | 34.0358 | 34.6495 | 34.3946 | 34.6504 | 34.3039 | 34.0057 | 34.3126 | 34.5586 |
| | SSIM | 0.9399 | 0.9541 | 0.9464 | 0.9545 | 0.9496 | 0.9554 | 0.9538 | 0.9507 | 0.9560 | 0.9512 |
| | LAION AES | 5.2329 | 5.0918 | 5.0654 | 4.9573 | 4.8523 | 4.8141 | 4.9275 | 4.8284 | 4.7538 | 4.9471 |
| | rFID | 19.7455 | 19.4921 | 19.8665 | 14.3300 | 17.1774 | 12.9881 | 13.9213 | 13.7164 | 12.9027 | 16.0156 |
| | LPIPS | 0.0687 | 0.0714 | 0.0740 | 0.0537 | 0.0553 | 0.0414 | 0.0493 | 0.0461 | 0.0428 | 0.0559 |

