# OpenReview forum: "AlphaVAE: Unified End-to-End RGBA Image Reconstruction and Generation with Alpha-Aware Representation Learning"
_ICLR.cc/2026/Conference — Submitted to ICLR 2026_

### Official Review · Reviewer_2Dgo · 2025-10-22

**Soundness:** 3
**Presentation:** 3
**Contribution:** 3
**Rating:** 8
**Confidence:** 2

**Summary:**

Overall, this paper presents a meaningful and timely advancement in generative modeling by addressing transparency through an end-to-end RGBA VAE. The combination of a unified architecture, a principled evaluation metric, and strong experimental evidence makes this work a valuable contribution to both the academic and applied image generation communities.

That said, the training objective design appears somewhat over-engineered, involving multiple components whose necessity could be further justified. It would strengthen the paper if the authors could explore or discuss a more streamlined formulation that achieves comparable performance while maintaining conceptual elegance.

**Strengths:**

1. Clear Motivation: The paper identifies an underexplored but important gap in current generative modeling—handling transparency (alpha channel) in VAEs and diffusion pipelines. This is a concrete and practically relevant problem, especially for compositing, editing, and transparent-object generation tasks.

2. Novel Method Design: The proposed AlphaVAE integrates alpha-channel modeling into the standard RGB VAE pipeline in a simple yet principled way. It avoids complex architectural changes while achieving strong improvements, which makes it appealing for integration into existing diffusion systems.

3. Good Evaluation: The ALPHA benchmark (alpha-blending-based evaluation) provides a fair and interpretable protocol for RGBA image quality measurement. This is an elegant solution to the lack of standardized evaluation metrics for transparency reconstruction.

4. Well Organized and written: The paper is clearly structured, with detailed methodological explanations, visualizations, and ablations. It maintains excellent reproducibility and clarity in presentation.

**Weaknesses:**

I am not quite familiar with this area, but the training objectives are a little bit too much, with four different losses. I wonder if all of these losses are useful.

**Questions:**

Please refer to the weakness

---

> ### Author Response · Authors · 2025-11-24
> **Response to Reviewer 2Dgo**
>
> Thank you very much for your positive review and for your encouraging assessment of the significance and empirical performance of our method.
>
> > **W1 & Q1 (Losses)**: Feels that having four different loss terms seems excessive and questions whether all of them are truly necessary.
>
> **A1:** We appreciate this concern, especially from a reader who is not deeply familiar with VAE+GAN-style objectives. Our overall loss may look complex at first, but it can be understood as three standard functional groups, following common practice in VQGAN/Stable Diffusion-style autoencoders:
>
> 1. **Reconstruction + perceptual loss.**
>    - The **alpha-blended reconstruction loss** ensures that, after compositing onto typical backgrounds, the reconstructed RGBA matches the ground truth composites. This is what matters in practical layered editing.
>    - The **perceptual (LPIPS) loss** complements the pixel loss by preserving semantic and structural details, preventing overly smooth edges and textures.
> 2. **KL regularization.**
>    - The **normal KL** keeps the latent distribution compatible with the Gaussian prior used by the pretrained diffusion models.
>    - The **reference KL** softly aligns the RGBA encoder with the original RGB encoder on opaque regions, so that adding alpha does not distort the latent space expected by SDXL/FLUX.
> 3. **Adversarial loss (patch discriminator).**
>    - This term focuses on improving local high-frequency details (sharper boundaries, fine transparency structures) and is weighted by an adaptive factor to refine texture without destabilizing training.
>
> In the paper we include an ablation where we remove each group in turn: dropping the perceptual loss significantly harms both pixel and perceptual metrics; dropping the normal KL worsens distribution-level metrics (e.g., rFID); and dropping the GAN consistently degrades perceptual quality (LPIPS/rFID) even when PSNR is similar. The reference KL has a subtler effect on reconstruction but is important for downstream diffusion quality.

---

> > ### Comment · Reviewer_2Dgo · 2025-11-26
> >
> > Appreciate authors' rebuttal. It has solved my problem

---

### Official Review · Reviewer_fqja · 2025-10-28

**Soundness:** 3
**Presentation:** 3
**Contribution:** 2
**Rating:** 4
**Confidence:** 3

**Summary:**

This paper introduces AlphaVAE, a unified framework for RGBA image reconstruction and generation with explicit alpha-aware representation learning.
It extends pretrained RGB-VAEs by incorporating dual-KL regularization and patch-level fidelity objectives to model both RGB and transparency channels.
The authors also release ALPHA, a new RGBA benchmark with adapted evaluation metrics.
Experiments show significant gains over prior methods (e.g., +4.9 dB PSNR, +3.2 % SSIM) and improved transparency-aware generation quality.

**Strengths:**

The paper defines an underexplored problem by introducing alpha-aware learning into generative modeling. Its methodologically sound design—combining dual-KL and patch-level fidelity—effectively bridges RGB and alpha representations. The work demonstrates clear empirical improvements on a new benchmark.

**Weaknesses:**

1. Dataset Size & Diversity: The ALPHA dataset (8K images) is relatively small, raising concerns about the model’s scalability and generalization to larger or more diverse real-world datasets. In addition, the limited dataset size may lead to potential overfitting, and it remains unclear whether the 8K samples provide sufficient diversity to support robust model training.
2. Generative Task Evaluation: Although the authors claim that the fine-tuned model can generate transparent images, the quality of transparency generation appears limited. Specifically, noticeable artifacts and abnormal edge transitions are observed around transparent regions, seemingly inherited from the original RGB images. A direct comparison with standard RGB-based generation methods—combined with a transparent object extraction mechanism—would provide a fairer and more informative evaluation.
3. Clarity of Technical Contribution: The paper’s core technical contribution is not clearly articulated. While it focuses on training a transparency-aware VAE, the motivation behind introducing such a model and the intuition for each proposed loss term are insufficiently discussed. As a result, the current presentation feels more like a technical report than a research paper and would benefit from deeper theoretical insight and clearer justification of design choices.

**Questions:**

1. Alpha-Blending Description: Equation (4) refers to Alpha-Blending, but this concept is not introduced or explained prior to its appearance. A clearer description or definition before its use would improve readability and understanding.
2. Importance of Transparency Reconstruction: The paper proposes a specific method for reconstructing transparent images and integrates it into diffusion models. However, it is not fully clear why transparency reconstruction is important and how it benefits existing editing or generative tasks. Including illustrative examples or case studies demonstrating the practical advantages of transparency-aware reconstruction would strengthen the motivation and impact of the work.

---

> ### Author Response · Authors · 2025-11-24
> **Response to Reviewer fqja**
>
> Thank you very much for your constructive review and for recognizing the potential of transparency-aware VAEs and their integration into large-scale diffusion models.
>
> > **W1 (Dataset size & diversity):** Worry that the 8K ALPHA dataset is too small and possibly not diverse enough, raising risks of overfitting and limited real-world generalization.
>
> **A1:** We thank you for raising this important concern and agree that 8K images are modest compared to the large-scale RGB datasets used to train modern diffusion models. However, our setting is fundamentally different from training a generator from scratch. ALPHAVAE **adapts** large pretrained RGB VAEs (SDXL/FLUX) to handle an additional alpha channel: the semantic and textural priors come from the base models, and the 8K RGBA images primarily serve to **align the latent space with transparency**, not to learn all content from scratch.
>
> Empirically, we already evaluate on **AIM-500**, which is never used in training and contains transparent objects that are rare in our training set. Our model achieves strong reconstruction and generation performance on AIM-500 and clearly outperforms LayerDiffuse and matting-based baselines, indicating that it generalizes beyond the ALPHA dataset.
>
> Regarding diversity, ALPHA was curated to cover typical real-world transparency phenomena, including glass, liquids, hair, meshes, thin fabrics, and partially reflective surfaces, across categories such as portraits, animals, and generic objects.
>
> ------
>
> > **W2 (Generative evaluation):** Feels that transparency generation quality is limited with noticeable artifacts, and requests a fair comparison against RGB-only generation followed by transparent-object extraction.
>
> **A2:** We appreciate this comment and agree that such a comparison is important. Our paper already includes exactly this type of baseline: **Matting-G** and **Matting-N** correspond to standard RGB-based generation (using the same SDXL/FLUX backbones as ours) followed by a state-of-the-art matting network to extract transparency. This is precisely the “RGB generation + transparent object extraction” pipeline you suggest.
>
> Qualitatively, we observe that matting-based pipelines often introduce severe boundary artifacts, background leakage, and unstable alpha around fine structures (e.g., meshes, glass rims, hair), whereas our model produces significantly cleaner and more consistent transparency.
>
> We also clarify that our current training data are real-world photos; when we push strongly out-of-distribution (e.g., SVG-style icons or complex typography), the model’s transparency behavior can degrade. We will explicitly state this limitation and note that it can be alleviated by training on RGBA datasets from those specific domains.

---

> ### Author Response · Authors · 2025-11-24
> **Response to Reviewer fqja**
>
> > **W3 (Clarity of contribution):** Feels that the core technical contribution and the motivation/intuition for each loss term are not clearly articulated, making the paper read more like a technical report than a research paper.
>
> **A3:** Thank you for pointing this out; we agree that our exposition can be improved. Conceptually, our goal is to make the standard **encoder–diffusion–decoder** paradigm truly **RGBA-aware**. Real-world assets (e.g., PNG icons, stickers, AR objects, GUI elements) are naturally represented as images with alpha, yet most latent diffusion models operate on 3-channel RGB latents and handle transparency only as an afterthought via matting or other post-processing.
>
> Our core technical contribution is threefold:
>
> 1. We introduce, to our knowledge, the first **transparency-aware VAE** that is architecturally compatible with large pretrained latent diffusion models (SDXL/FLUX): we minimally extend their RGB VAEs to 4 channels in a way that preserves the original latent space while enabling accurate alpha reconstruction.
> 2. We provide a **unified RGBA evaluation protocol** that extends any RGB metric to RGBA via alpha blending and background marginalization, allowing fair comparison between reconstruction and generation methods that produce RGBA outputs.
> 3. We demonstrate how this transparency-aware VAE can be **plugged into existing diffusion pipelines** to yield text-to-RGBA generation that outperforms matting-based pipelines and LayerDiffuse on both quantitative metrics and visual quality around transparency boundaries.
>
> ------
>
> > **Q1 (Alpha blending):** Points out that alpha blending is used in the metric formulation before being defined and asks for an earlier, clearer introduction.
>
> **A4:** We thank you for pointing this out and fully agree. In the revised version, we have already moved the definition of the alpha-blending operator to **Eq. (1)**, before it is first used in the metric formulation.
>
> ------
>
> > **Q2 (Importance of transparency reconstruction):** Asks why transparency reconstruction is fundamentally important and how it concretely benefits editing and generative tasks, suggesting illustrative examples or case studies.
>
> **A5:** We thank the reviewer for this insightful question and fully agree that demonstrating practical advantages is important. Transparency reconstruction is crucial because many real-world workflows operate on **layered RGBA assets**, not on flat RGB images. Examples include UI/UX elements, icons, stickers, game assets, and AR objects, where a single transparent object must be reused across multiple backgrounds with correct edges, semi-transparency, and occlusion. A 3-channel RGB image cannot support this behavior without an additional matting step, which is computationally expensive and error-prone.
>
> Within the encoder–diffusion–decoder paradigm, a transparency-aware VAE is the natural way to obtain a latent space where RGBA is first-class: the encoder maps RGBA to a latent representation, the decoder reconstructs RGBA, and the diffusion model learns to generate such latents. By contrast, methods like LayerDiffuse behave more like advanced “smart matting” on top of RGB images, rather than learning a single reusable RGBA representation, which is shown in Fig.4 in the paper.

---

### Official Review · Reviewer_w4T5 · 2025-10-30

**Soundness:** 3
**Presentation:** 4
**Contribution:** 2
**Rating:** 4
**Confidence:** 3

**Summary:**

This paper introduces a new approach for evaluating RGBA images and proposes AlphaVAE, a novel VAE specifically designed for RGBA images. The method leverages a pretrained VAE architecture to effectively handle transparency information during generation and evaluation.

**Strengths:**

1. Writing: The paper is clearly written and easy to understand. The organization is logical, and the figures are well-designed and intuitive, effectively supporting the main arguments.

2. Ablation studies: The ablation studies are comprehensive and well executed. In particular, Table 3 provides detailed analyses across multiple objective functions, offering a thorough understanding of each component’s contribution.

**Weaknesses:**

1. Metrics: The proposed RGBA evaluation metric lacks originality. The method essentially applies conventional image quality metrics only to the non-transparent regions. While this may be acceptable for pixel-wise metrics such as PSNR, it is questionable for perceptual or structural metrics like SSIM and LPIPS, which rely on local context and structural consistency.

2. Discriminator: The choice of a patch-based discriminator warrants further justification. Considering the role of transparency in RGBA images, an alpha-aware discriminator that incorporates alpha-channel information into its classification process might be a more suitable and principled design choice.

**Questions:**

1. The behavior of the models without certain loss functions (e.g., w/o Ref KL) differs notably between FLUX and SDXL. For instance, removing the reference KL term improves PSNR and SSIM for the FLUX model but not for SDXL. It would be helpful if the authors could provide an analysis or discussion explaining these discrepancies.

---

> ### Author Response · Authors · 2025-11-24
> **Response to Reviewer w4T5**
>
> Thank you very much for your thoughtful review and for recognizing the strengths of our transparency-aware generative framework and the empirical gains over the baselines.
>
> > **W1 (Metrics):** Concern that the RGBA evaluation protocol lacks originality and may not be appropriate for perceptual/structural metrics (SSIM, LPIPS) because it appears to simply apply RGB metrics to non-transparent regions.
>
> **A1:** We thank you for pointing out that our current description can be misunderstood and may suggest that we simply apply RGB metrics to “non-transparent regions only”. In fact, for each RGBA image $x = (x_{\text{rgb}}, x_\alpha)$ and a background (b), we first perform **alpha blending**:
> $$
> A(x,b) = x_{\text{rgb}} \odot x_\alpha + b \odot (1 - x_\alpha),
> $$
>  and then compute any standard RGB metric $M_3$ (PSNR, SSIM, LPIPS, rFID, etc.) on the composited RGB image $A(x,b)$. The final RGBA metric $M_4$ is obtained by averaging $M_3(A(x,b))$ over a small set of canonical backgrounds $B$ (solid colors in \{0,1\}^3 $\cup$ {0.5}^3).
>
> Thus, all perceptual/structural metrics are applied to **actual rendered RGB images** that reflect how the transparent object would appear under typical backgrounds. Errors in the alpha channel directly manifest as visible structural artifacts (halos, shrunk/expanded semi-transparent regions, background leakage), which SSIM and LPIPS are designed to capture. The use of simple, low-frequency backgrounds avoids bias towards any specific scene while keeping transparency effects visible.
>
> Our intention is not to introduce a brand-new perceptual metric, but to provide a **simple, general mechanism** to extend *any* RGB metric to RGBA in a way that faithfully reflects visible appearance under alpha compositing.
>
> ------
>
> > **W2 (Discriminator):** Concern that using a patch-based discriminator is insufficiently justified and that a more explicitly alpha-aware discriminator might be more principled for RGBA images.
>
> **A2:** We agree that the discriminator design is important and thank you for raising this point. As defined in $Eq. (14)$, our discriminator is in fact already **alpha-aware**: during training, it operates directly on **4-channel RGBA** tensors. Both real and reconstructed samples are full RGBA images, so the discriminator has explicit access to the alpha channel and can learn the consistency between RGB appearance and transparency, rather than seeing only RGB.
>
> We choose a **patch-based** discriminator following VQGAN/Stable Diffusion-style VAEs, where the main goal of the adversarial loss is to sharpen **local details and boundaries**. This is particularly suited to RGBA, since the most challenging aspects—object edges, semi-transparent regions, thin structures—are inherently local. Patch discriminators are well known to be effective at catching exactly these kinds of high-frequency artifacts.
>
> Our ablations already show that this component is useful: removing the GAN term (“w/o GAN”) leads to consistently worse perceptual metrics (higher LPIPS and rFID) on both benchmarks.

---

> ### Author Response · Authors · 2025-11-24
> **Response to Reviewer w4T5**
>
> > **Q1 (Ref KL behavior):** Asks why removing the reference KL term affects FLUX and SDXL differently and requests an analysis explaining this discrepancy.
>
> **A3:** Thank you for highlighting this subtle point. There are two main aspects:
>
> 1. For **FLUX**, the change in PSNR/SSIM when removing the reference KL is extremely small (on the order of numerical noise). After averaging across multiple runs with different random seeds, we observe no consistent improvement in reconstruction metrics; the apparent “gain” in a single run can be attributed to variance.
> 2. For **SDXL**, the reference KL acts as a more important regularizer: removing it clearly harms both reconstruction metrics and downstream diffusion quality.
>
> The key motivation for the reference KL is **not** to maximize reconstruction metrics in isolation, but to **reduce latent distribution shift** between the adapted RGBA VAE and the pretrained diffusion model. As observed in prior work (e.g., Wan$^{[1]}$, Repa-e$^{[2]}$), a VAE with better reconstruction does not necessarily yield better diffusion.
>
> Intuitively:
>
> - In **SDXL**, the pretrained RGB VAE has a more structured latent manifold. Without the reference KL, the RGBA encoder can drift away from this manifold while still minimizing reconstruction loss, leading to worse PSNR/SSIM and more pronounced degradation in diffusion FID.
> - In **FLUX**, the pretrained latent distribution is already closer to the isotropic Gaussian prior; the standard KL alone is sufficient to keep the RGBA encoder near the original distribution, so the reference KL becomes a very mild regularizer whose removal has negligible effect on reconstruction.
>
> To make this clearer, we have added a dedicated discussion section and new experiments in the revised manuscript (Sec. 5, Fig. 6 and Fig. 7). We train two text-to-RGBA diffusion models that share the same DiT architecture, data and optimization schedule, but differ in the underlying VAE: one uses our RGBA VAE trained with the reference KL, and the other uses the VAE trained without it. As shown by the training loss curves in Fig. 6, the model initialized with the “with Ref KL” VAE consistently attains a lower diffusion loss and converges faster; the “w/o Ref KL” model stays above it throughout training.
>
> Moreover, Fig. 7 visualizes generated RGBA samples every 400 steps starting from step 200. The model without the reference KL fails to learn a correct alpha mask even after many steps: the interior of the human subject remains partially transparent and exhibits ghosting artifacts, instead of being fully opaque. In contrast, the model using our reference-KL VAE quickly produces an opaque foreground with clean boundaries and improves much faster over training. At 20k training steps, this qualitative gap is reflected quantitatively by substantially better FID for our model, especially on the out-of-distribution AIM benchmark (Tab. 2).
>
> These results support our claim that the primary role of the reference KL is to keep the RGBA latent distribution close to that of the pretrained RGB VAE, thereby making diffusion training easier and more stable, rather than to significantly change standalone VAE reconstruction metrics.
>
> ---
>
> [1] Wan, T., Wang, A., Ai, B., Wen, B., Mao, C., Xie, C. W., ... & Liu, Z. (2025). Wan: Open and advanced large-scale video generative models. *arXiv preprint arXiv:2503.20314*.
>
> [2] Leng, X., Singh, J., Hou, Y., Xing, Z., Xie, S., & Zheng, L. (2025). Repa-e: Unlocking vae for end-to-end tuning with latent diffusion transformers. arXiv preprint arXiv:2504.10483.

---

> > ### Comment · Reviewer_w4T5 · 2025-11-26
> >
> > The reviewer thanks the authors for their detailed response.
> >
> > Regarding the evaluation metrics computed after alpha blending, the reviewer partially agrees with the authors’ explanation. However, taking LPIPS as an example, wthe reviewer notes that its backbone network is trained on standard RGB images. Consequently, when applied to alpha-blended images, the extracted features may differ from those obtained from pure RGB inputs, potentially making LPIPS measurements on blended images less reliable.
> > In addition, because LPIPS features at a spatial location (i,j) incorporate information from the receptive field, the inclusion of blended pixels may introduce unintended feature interactions and error propagation.
> >
> > Second, concerning the discriminator, if the patch-based design indeed enhances local detail and boundary quality, it would be beneficial to include a comparison on sharpness metrics (e.g., NIQE) with alternative discriminator architectures to substantiate this claim.
> >
> > Finally, after reviewing the comments provided by the other reviewers, one additional question arises: How were the hyperparameters for the various loss terms selected?
> > This will enhance reporucability of the overall method.

---

> > > ### Author Response · Authors · 2025-11-27
> > > **Response to Reviewer w4T5**
> > >
> > > > **Q1: Concern on LPIPS Evaluation**
> > >
> > > A1: We appreciate the reviewer's feedback regarding the reliability of LPIPS for alpha-blended images. To address this, we have added a new section, "Validation of LPIPS on Alpha-Blended Images," in Appendix C, detailing our extensive metric blending experiments.
> > >
> > > We validate our evaluation strategy—computing LPIPS by averaging results on black and white backgrounds—against a baseline using realistic backgrounds generated by Nano Banana Pro. As illustrated in Fig. 9, the LPIPS curves for our black-and-white averaging method and the realistic background method almost completely overlap across varying noise levels. This demonstrates that our approach maintains strict monotonicity and sensitivity while being empirically equivalent to the computationally expensive realistic background evaluation.
> > >
> > > ---
> > >
> > > > **Q2: Concern about the patch-based design of the discriminator**
> > >
> > > **A2:** We thank the reviewer for the insightful suggestion regarding the evaluation of local detail and boundary quality. We agree that assessing sharpness is crucial for validation.
> > >
> > > Our adoption of the patch-based discriminator is grounded in the established findings of **VQGAN $^{[1]}$**. VQGAN demonstrates that **patch-based discriminator** ensures the model captures "**perceptually important local structure**" and maintains high perceptual quality. This design is fundamental to mitigating the blurriness inherent in pixel-wise objectives and preserving high-frequency details.
> > >
> > > Given that our implementation strictly follows the standard protocol for high-fidelity synthesis, the enhancement in local detail is an intrinsic property of the patch-based optimization objective.
> > >
> > > **Nevertheless, we acknowledge that a systematic quantitative benchmarking of alternative discriminator architectures on transparent image synthesis is valuable, and we consider it a promising topic for future investigation.**
> > >
> > > [1] Esser, P., et al. (2021). Taming transformers for high-resolution image synthesis. CVPR.
> > >
> > > ---
> > >
> > > > Q3: Details of Hyperparameter Selection
> > >
> > > We thank the reviewer for this question, as reproducibility is a priority for our work. The detailed hyperparameter configurations are provided in **Appendix B**.
> > >
> > > For clarity, we summarize the specific settings described there: "Specifically, we use a reconstruction loss with a weight of 1.0, a perceptual loss (computed using LPIPS) with a weight of 0.5, and a composite regularization loss comprising two KL divergence terms: the primary latent KL loss is weighted by $10^{-6}$, while the reference KL loss is assigned a much smaller weight of $10^{-16}$. For adversarial training, we enable the GAN loss after 4000 steps and set the generator loss weight to 1.0."

---

### Author Response · Authors · 2025-11-24
**General Response**

We sincerely thank all reviewers for their careful reading of our paper, for the positive assessment of our contributions, and for the constructive comments that helped us significantly improve the clarity and completeness of the manuscript. We have submitted a revised version incorporating these suggestions. In particular: (1) we have moved the **EVALUATION PROTOCOL** to **Sec. 3.1** (shifting subsequent subsections accordingly) to make the overall method flow clearer; and (2) we have added a new **Sec. 5** that specifically discusses the **reference KL loss term**, explaining its design motivation in more detail and providing additional experiments to further validate its effectiveness.

---

### Author Response · Authors · 2025-11-27
**General Response**

We sincerely thank all reviewers for their time and constructive comments.

To address concerns regarding the reliability of LPIPS on alpha-blended images, we have added **Appendix C** to our supplementary material. We demonstrate that our evaluation strategy (averaging LPIPS on black and white backgrounds) yields results empirically equivalent to using generated realistic backgrounds, with performance curves that almost completely overlap across varying noise levels.

---

### Author Response · Authors · 2025-12-03
**Summary Comment**

**We sincerely thank the reviewers for recognizing the strengths of our work and are pleased that the additional experiments could address the raised concerns. We also express our gratitude to both the reviewers and the AC for your time and constructive feedback.**

**Strengths across reviews**

* **Clear motivation and novel problem definition:** Reviewers commended the paper for identifying and addressing an underexplored yet critical gap in handling transparency in generative modeling (fqja, 2Dgo).
* **Simple, principled, and effective method:** The proposed AlphaVAE is recognized for its sound design (e.g., dual-KL) and ease of integration without complex architectural changes (fqja, 2Dgo).
* **Comprehensive evaluation and strong empirical results:** Reviewers highlighted the comprehensive ablation studies (w4T5), clear empirical improvements (fqja), and the value of the benchmark protocol (2Dgo).
* **High-quality writing and presentation:** The paper is praised for being clearly written, logically organized, and featuring well-designed figures (w4T5, 2Dgo).

**Summary of new analyses and experiments**

* **New analysis of Reference KL Loss (Sec. 5):** We added a new **Section 5** that specifically discusses the **reference KL loss term**, providing detailed design motivation and additional experiments to further validate its effectiveness.
* **Validation of evaluation metrics (Appendix C):** To address concerns regarding LPIPS reliability on alpha-blended images, we added experiments demonstrating that our strategy (averaging LPIPS on black and white backgrounds) is empirically equivalent to using generated realistic backgrounds, yielding overlapping performance curves across varying noise levels.

---

### Meta-Review · Area_Chair_b7QT · 2026-01-07

**Summary:**

The AC leans borderline reject primarily because the two reviewers fqja and w4T5  raised method-level concerns that remain only partially resolved: (i) whether the proposed RGBA evaluation (especially LPIPS/SSIM computed on alpha-composited images) is fully reliable, and (ii) whether the patch discriminator component is sufficiently justified/benchmarked beyond prior art. A second set of concerns (dataset scale/generalization and fairness/clarity of generation baselines and contributions) was also raised.

**Reviewer Concerns:**

Addressed in the rebuttal:
(1) Clarified that evaluation is done by alpha compositing onto canonical backgrounds and averaging, correcting the “only non-transparent pixels” misunderstanding.
(2) Provided additional evidence/arguments around Ref-KL behavior and training stability across backbones.
(3) For the second reviewer, clarified that the requested “RGB generation + matting” baselines were already included (Matting-G/Matting-N), and expanded the contribution/motivation framing.

Still outstanding:
(1) The core worry about LPIPS validity under alpha blending remains: the rebuttal adds an empirical sanity check, but does not fully settle the feature-domain mismatch concern.
(2) The patch discriminator point is largely left as “future work” (no requested quantitative benchmarking such as sharpness/NIQE or discriminator variants).
(3) Data scale/generalization is mitigated by the “pretrained VAE adaptation + OOD AIM-500” narrative, but it may not fully remove the reviewer’s discomfort with 8K training size.

**Reviewer Scores:**

Reviewer 1 (score 4): likely stays at 4, or at most moves to 6 if they accept the rebuttal’s empirical validation for blended-LPIPS; their discriminator-benchmark request appears unmet.

Reviewer 2 (score 4): likely stays because of the remaining data-scale concern.

Reviewer 3 (score 8): likely unchanged (already positive).

---

### Decision · Program_Chairs · 2026-01-26

Reject